# Scaling Up, Speeding Up: A Benchmark of Speculative Decoding for Efficient LLM Test-Time Scaling

**Shengyin Sun**[1*], **Yiming Li**[2*], **Xing Li**[2], **Yingzhao Lian**[2], **Weizhe Lin**[2], **Hui-Ling Zhen**[2],
**Zhiyuan Yang**[2], **Xianzhi Yu**[2], **Chen Chen**[2], **Mingxuan Yuan**[2], **Chen Ma**[1†]
[1]City University of Hong Kong, [2]Huawei Technologies
`shengysun4-c@my.cityu.edu.hk, li.yiming3@huawei.com`
`Yuan.Mingxuan@huawei.com, chenma@cityu.edu.hk`

## Abstract

Test-time scaling has emerged as a powerful paradigm for enhancing the reasoning capabilities of large language models (LLMs) by allocating additional computational resources during inference. However, this paradigm is inherently inefficient due to the generation of different reasoning traces, leading to significant computational overhead. Speculative decoding offers a promising avenue for mitigating this inefficiency, yet its efficacy in the structured and repetition-rich context remains unexplored. To bridge this gap, we introduce the first comprehensive benchmark designed to evaluate speculative decoding methods in LLM test-time scaling. Our benchmark provides consistent experimental protocols across representative test-time scaling paradigms (e.g., Best-of-N sampling and multi-round thinking), enabling a fair comparison of three major categories of speculative decoding: model-based, training-based, and N-gram-based methods. Extensive experiments reveal that simple N-gram-based methods effectively capture repetitive patterns, demonstrating unique potential in accelerating test-time scaling. This phenomenon demonstrates the value of integrating N-gram-based methods with model-based or training-based approaches to benefit both repetitive and diverse reasoning in test-time scaling. We hope this benchmark spurs further research on speculative decoding for test-time scaling, enabling faster and more practical reasoning in LLMs through better handling of repetitive and diverse reasoning paths. Code available at `https://github.com/sunshy-1/SpecTTS-Bench`.

## 1 Introduction

In the pursuit of more capable large language models (LLMs), the test-time scaling (Snell et al., 2025; Wang et al., 2023; Madaan et al., 2023; Tian et al., 2025; Yao et al., 2023) has emerged as a key performance driver on complex tasks. Unlike training-time scaling, which relies on larger models or datasets, test-time scaling offers a complementary strategy that can be applied to already-deployed models without retraining. This approach enhances performance by allocating substantial computational resources during inference, allowing the model to "think" longer and more deeply. Dominant paradigms in test-time scaling include *Best-of-N (BoN) sampling* (Snell et al., 2025) and *multi-round thinking* (Tian et al., 2025). These paradigms have demonstrated remarkable success in complex reasoning (Lightman et al., 2023; Cobbe et al., 2021; Li et al., 2022). However, this enhanced performance comes at a steep price. Generating multiple full-length responses or iterative reasoning chains incurs prohibitive computational overhead, leading to a significant latency bottleneck that severely limits their use in real-time, interactive scenarios.

To unlock the potential of test-time scaling for practical use, it is imperative to mitigate its associated latency. Speculative decoding (Liu et al., 2025; Huang et al., 2025) has emerged as a leading technique to achieve this goal. This approach accelerates autoregressive generation by using a fast

---

* Equal contribution.
† Corresponding author.

Table 1: Illustration of intra-turn and inter-turn redundancy under the test-time scaling paradigm.

| Question | Let $p$ be the least prime number for which there exists a positive integer $n$ such that $n^4 + 1$ is divisible by $p^2$. Find the least positive integer $m$ such that $m^4 + 1$ is divisible by $p^2$. |
|---|---|
| Intra-turn Redundancy | **Turn 1:** Compute $8^3 = 512$. Then 512 mod 17: since $17 \times 30 = 510$, we have $512 \equiv 2 \pmod{17}$. Therefore, $4 \times 8^3 \equiv 4 \times 2 = 8 \pmod{17}$. Thus, $4 \times 8^3 \times 17k \equiv 8 \times 17k \pmod{289}$. So, $n^4 \equiv 50 + 8 \times 17k \pmod{289}$. Hence, $n^4 + 1 \equiv 51 + 136k \pmod{289}$. We need this to be 0 (mod 289), so $51 + 136k \equiv 0 \pmod{289}$. Dividing both sides by 17, we get $3 + 8k \equiv 0 \pmod{17}$, i.e., $\cdots$ Wait, earlier I said that $4 \times 8^3 \times 17k \equiv 8 \times 17k$ mod 289. Let me check that again. $4 \times 8^3$. Compute $8^3 = 512$. Then 512 mod 17: since $17 \times 30 = 510$, we have $512 \equiv 2 \pmod{17}$. |
| Inter-turn Redundancy | **Turn 1:** `<think>\n` Okay, so I need to find the least prime number $p$ such that there's a positive integer n where $p^2$ divides $n^4 + 1$. Then, once I find that p, I have to find the smallest positive integer m such that $p^2$ divides $m^4 + 1$. Alright, let me start by understanding the problem $\cdots$ 
 **Turn 2:** `<think>\n` Okay, so I need to find the least prime p such that there's a positive integer $n$ with $p^2$ dividing $n^4 + 1$. Then, once I find that $p$, I need to find the smallest m such that $m^4 + 1$ is divisible by $p^2$. `\n\n`First, let me recall that $\cdots$ |

draft mechanism to generate candidate tokens, which are then efficiently verified in a single pass by the larger target model. This process substantially reduces wall-clock time while remaining lossless. On the other hand, the repetitive nature of test-time scaling creates uniquely favorable conditions for speculative decoding to excel, particularly for retrieval-based solutions (Hu et al., 2025; Oliaro et al., 2024). These retrieval-based approaches are inspired by N-gram models, which predict the next token based on a specific sequence of prior tokens, such as pairs (bigrams) or triples (trigrams). As a model explores different reasoning paths, it frequently outputs highly repetitive token sequences. These can include recurring logical phrases (e.g., "Let's consider the case where..."), standard boilerplate code, or common transitional expressions (as shown in Table 1). This repetitive structure suggests that N-gram-based methods (Hu et al., 2025; Oliaro et al., 2024), which dynamically cache and reuse recently generated token sequences, are exceptionally well-suited to this scenario.

Despite promising acceleration, existing literature lacks comparative analysis to determine which speculative decoding architecture best exploits computational redundancies in test-time scaling. Current benchmarks evaluate these methods in generic tasks (Xia et al., 2024), overlooking the unique characteristics of advanced reasoning paradigms. Consequently, critical questions remain unanswered: *Can a simple, adaptive N-gram mechanism outperform a more complex, pre-trained draft model in a scenario ripe with repetition? How do the trade-offs between these methods—flexibility, training cost, and real-time adaptability—manifest when accelerating test-time scaling frameworks?* Without a dedicated benchmark to investigate these specific dynamics, the community cannot make informed decisions about how to efficiently scale up inference-time reasoning.

To address this critical gap, we introduce the first comprehensive benchmark designed to systematically compare different speculative decoding methods for accelerating test-time scaling. Specifically, our work provides a head-to-head comparison of 9 speculative decoding methods (detailed in Section 3.3), covering nearly all state-of-the-art (SOTA) techniques in this domain. We aim to elucidate the trade-offs among these approaches and, most importantly, test the hypothesis on the unique effectiveness of using N-gram patterns to accelerate test-time scaling. Our observations are summarized as follows. (i) The method that relies on training an auxiliary draft model shows promising acceleration potential, but its inference behavior is inherently coupled to the training process, limiting adaptability across diverse reasoning scenarios. (ii) Methods leveraging N-gram patterns excel at capturing redundancy in test-time scaling; in particular, those that store and reuse N-gram patterns during generation achieve progressive acceleration by reusing past computations, enabling efficiency gains that accumulate across generation steps. However, this advantage is sensitive to sampling temperature. (iii) A *brand new hybrid method* called SAM[EAGLE-3] is tested (detailed in Section 3.3). Hybrid speculative decoding uniquely combines methods that train an auxiliary draft model for semantic alignment with those leveraging N-gram patterns to capture repetition, unlocking distinct potential for reasoning under test-time scaling. However, current hybrid strategies remain heuristic, failing to effectively harness the semantic richness of draft models and the reusability of N-gram patterns. We hope these findings can provide practical guidance for the design of speculative decoding strategies in test-time scaling, balancing adaptability, efficiency, and robustness for complex reasoning tasks.

## 2 RELATED WORK

**Test-time scaling for enhanced reasoning.** The paradigm of test-time scaling enhances model performance by allocating more computational resources at inference time. This principle can be realized through two main avenues (Zhang et al., 2025): (1) specialized training that endows a model with an inherent capability to dynamically scale its computation, or (2) external, training-free frameworks that guide a fixed model. The former approach, inherent test-time scaling, is embodied by the development of "slow thinking" or System 2 reasoning models. Models such as OpenAI o1 series (OpenAI et al., 2024) are explicitly trained to internalize multi-step reasoning. Complementing this inherent capacity are external frameworks, which apply an outer loop of control to a base model, further amplifying its reasoning performance without modifying its parameters. This paper focuses on analyzing two foundational categories of such frameworks: (1) Sampling-based methods such as BoN sampling (Snell et al., 2025) and Self-Consistency (Wang et al., 2023). (2) Iterative refinement methods such as Self-Refine (Madaan et al., 2023) and multi-round thinking (Tian et al., 2025). More details about these frameworks can be found in Appendix A.1.1.

**Speculative decoding for inference acceleration.** Speculative decoding has emerged as a SOTA solution for reducing LLM inference latency losslessly. The landscape of speculative decoding can be classified into three families: (1) Model-based speculative sampling which uses a smaller, general-purpose language model as the drafter (Leviathan et al., 2023; Chen et al., 2023). (2) Training-based speculative decoding which trains a specialized draft model (Cai et al., 2024; Li et al., 2024a; 2025c; Sun et al., 2026). (3) N-gram-based method which builds a "draft" by retrieving token sequences from a dynamic cache of recent outputs (He et al., 2024; Zhao et al., 2024; Hu et al., 2025; Oliaro et al., 2024). While these methods have been benchmarked on generic tasks (Xia et al., 2024), their interplay with test-time scaling frameworks remains largely unexplored. The highly structured and often repetitive nature of reasoning chains generated by methods like BoN presents a unique environment where the trade-offs between these different speculative decoding families are unknown. Our work directly addresses this gap by systematically evaluating their performance in these latency-critical, redundancy-rich reasoning scenarios. Additional relevant work is discussed in Appendix A.1.2.

## 3 BENCHMARK CONSTRUCTION

This section outlines the three core components of our experimental setup: the test-time scaling frameworks to accelerate, the evaluation datasets, and the speculative decoding algorithms compared. The overall framework is illustrated in Figure 1.

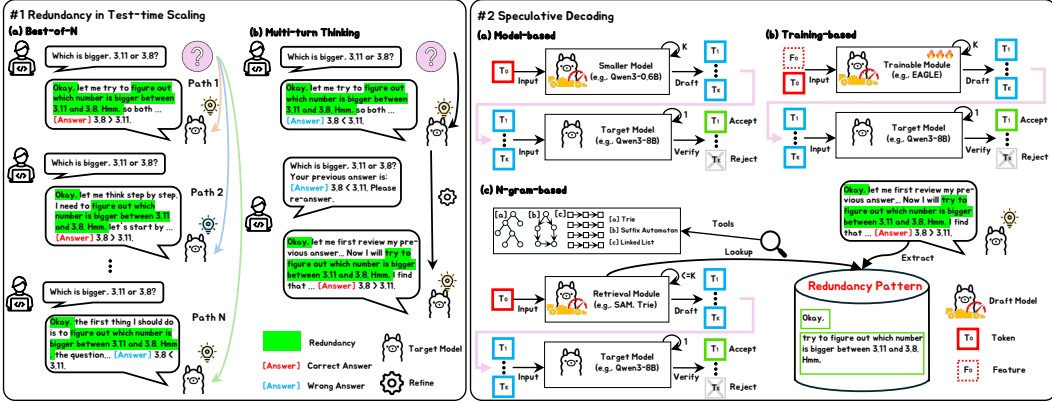

Figure 1: The framework of speculative decoding for efficient test-time scaling. (#1) An overview of test-time scaling frameworks and their inherent redundancies. (#2) Illustrations of three categories of methods. *Model-based method* uses a smaller, homologous model as draft model to generate speculative token sequences. *Training-based method* employs an additionally trained draft model using both input tokens and hidden features. *N-gram-based method* leverages tools such as suffix automata to retrieve repeated patterns for draft generation.

## 3.1 BENCHMARKED TEST-TIME SCALING FRAMEWORKS

This work selects two particularly representative frameworks in test-time scaling, namely BoN and multi-round thinking, as the basis of our benchmark.

### 3.1.1 BoN

The BoN (Snell et al., 2025) is a foundational technique designed to improve output quality by leveraging a generate-and-rank mechanism. Instead of relying on a single, potentially flawed output from the model, BoN operates in two distinct stages:

- **Generation:** Given a request, the LLM is prompted to generate $N$ independent candidate solutions or reasoning paths. This is typically achieved by sampling from the model's output distribution multiple times with a non-zero temperature to ensure diversity. The collection of these $N$ samples represents a broad exploration of the potential solution space.

- **Selection/Verification:** After generating the $N$ candidates, a verifier module is used to score and select the best solution. The verifier can range from a simple, heuristic-based checker to a more sophisticated, separately trained reward model. In its most canonical form, a process-based verifier might assess the correctness of each intermediate step in a solution, rather than just evaluating the final answer. The candidate with the highest score is then chosen as the final output.

### 3.1.2 MULTI-ROUND THINKING

Multi-round thinking (Tian et al., 2025) improves reasoning through iterative refinement. This method is inspired by the human cognitive process of reconsideration. It enhances a model's reasoning by prompting it to reflect on and improve its own previous answers over rounds, as follows:

- **Initial Response (Round 1):** The LLM generates an initial answer to the user's prompt. This represents the model's initial or intuitive response.

- **Iterative Refinement (Rounds 2 to *M*):** The process is repeated for a pre-defined total of $M$ rounds. In each subsequent round (from 2 to $M$), the model is prompted to refine its previous output. The new prompt is constructed by augmenting the original question with the answer from the preceding round. For example, the prompt for the 2nd round takes the following structure:

  ```
  {Original Question}
  The assistant's previous answer is: {Round 1 Answer}
  Please re-answer.
  ```

  This mechanism forces the model to reconsider its initial reasoning path, allowing it to self-correct errors and elaborate on its response with each iteration.

## 3.2 BENCHMARKED DATASETS

Having defined the computational frameworks that expand reasoning effort at inference time, we now turn to the datasets used for evaluation. The choice of datasets is critical to ensure that our benchmark measures reasoning capabilities across a diverse set of challenging problems. To this end, we have selected four widely-recognized benchmarks that have become staples in recent literature for evaluating the limits of LLM reasoning capabilities. Our evaluation suite is composed of 120 samples extracted from the following datasets:

- **AIME 2024 and AIME 2025:** The American Invitational Mathematics Examination (AIME) (AIME, 2025) features challenging competition-level mathematics problems that demand creative and rigorous problem-solving skills. We use the official problems from the 2024 and 2025 competitions, utilizing the full set of 30 problems from each.

- **MATH-500:** A subset of the MATH dataset (Hendrycks et al., 2021), a comprehensive benchmark of mathematics problems ranging from pre-algebra to calculus and beyond. It is the standard for evaluating mathematical reasoning in LLMs. Given the considerable size of the full 500-problem dataset, we selected the first 30 problems to ensure a focused and manageable evaluation.

- **GPQA:** The GPQA (Rein et al., 2023) consists of graduate-level, Google-proof Q&A problems across diverse domains, designed to test internalized knowledge and complex reasoning. The full dataset contains 448 expert-level questions, representing a challenging collection. To maintain consistency with other evaluation subsets, we selected the first 30 problems from this dataset.

This curated suite serves as a rigorous testbed for evaluating the acceleration effects of speculative decoding within the computationally intensive **BoN** and **multi-round thinking** frameworks, enabling us to quantify efficiency gains in high-stakes reasoning scenarios.

Table 2: Summary of speculative decoding methods, including their key ideas, categories (*model-*, *training-*, or *N-gram-based*), speculation structures (*Linear* or *Tree*), ability to reuse repetitive content during generation, and its compatibility with *Greedy* decoding and speculative *Sampling*.

| Methods | Key Idea | Drafting | | Verification | | Category |
|---------|----------|----------|-------|--------------|----------|----------|
| | | Speculation | Reuse | Greedy | Sampling | |
| SpS (Chen et al., 2023) | Speculative decoding with smaller homologous models and rejection sampling. | Linear | ✗ | ✓ | ✓ | Model-based |
| EAGLE-3 (Li et al., 2025c) | Introducing trainable lightweight auto-regressive heads on intermediate layers. | Tree | ✗ | ✓ | ✓ | Training-based |
| PLD (Saxena, 2023) | Generates drafts by reusing N-gram patterns extracted from the input prompt. | Linear | ✗ | ✓ | ✗ | N-gram-based |
| REST (He et al., 2024) | Drafting based on an external datastore for retrieving likely next tokens. | Tree | ✗ | ✓ | ✓ | N-gram-based |
| Lookahead (Fu et al., 2024) | Generating multiple N-gram candidates (extracted from generation) in parallel. | Tree | ✓ | ✓ | ✗ | N-gram-based |
| PIA (Zhao et al., 2024) | Speculative decoding via Trie-structured context caching. | Tree | ✓ | ✓ | ✓ | N-gram-based |
| SAM (Hu et al., 2025) | Generating drafts using a suffix automaton derived from the generation history. | Linear | ✓ | ✓ | ✓ | N-gram-based |
| Recycling (Luo et al., 2025) | Generating drafts by recycling top-K next tokens from history. | Tree | ✗ | ✓ | ✓ | N-gram-based |

## 3.3 BENCHMARKED SPECULATIVE DECODING ALGORITHMS

We benchmark a diverse set of methods, which can be broadly categorized into three groups: model-based speculative sampling, training-based speculative decoding, and N-gram-based methods. This collection of methods encompasses almost all existing speculative decoding techniques compatible with reasoning models, capturing the full spectrum of current SOTA solutions. We provide a summary of the above methods in Table 2. Detailed descriptions of each method are presented below.

- **Model-based Method**
  - *Speculative Sampling (SpS)* (Chen et al., 2023): As a pioneering speculative decoding method, SpS employs a smaller, faster draft model to generate a sequence of candidate tokens. The larger target model then validates these tokens in parallel using a rejection sampling mechanism.

- **Training-based Method**
  - *EAGLE-3* (Li et al., 2025c): This is an advanced speculative decoding approach that dispenses with a separate draft model. Instead, it attaches lightweight trained auto-regressive heads to the target model's intermediate layers to generate drafts.

- **N-gram-based Method**
  - *Prompt Lookup Decoding (PLD)* (Saxena, 2023): PLD identifies N-grams from the prompt that match the sequence generated so far and uses them as speculative candidates. PLD is particularly effective in tasks where the output is likely to contain substrings from the input.
  - *Retrieval-based Speculative Decoding (REST)* (He et al., 2024): REST utilizes a datastore of example sequences and a retrieval model to find continuations relevant to the current context. These continuations serve as the speculative draft, which is then verified by the target model.
  - *Lookahead* (Fu et al., 2024): A parallel decoding algorithm generating and validating multiple N-grams simultaneously. Unlike single-sequence speculative decoding, Lookahead maintains multiple continuations and uses a tree structure to verify candidates in a single forward pass.

- *PIA* (Zhao et al., 2024): This method utilizes a trie structure to manage and retrieve contextual information from the prompt and decoding context efficiently. It supports tree-based speculative decoding, where multiple candidate token sequences are generated and verified in parallel.
- *SAM* (Hu et al., 2025): SAM employs a suffix automaton to maintain the contextual history of the generated sequence. This data structure allows for the effective identification of previously generated token sequences (suffixes) that best match the current context.
- *Recycling* (Luo et al., 2025): Recycling enhances speculative decoding by caching the top-K most probable subsequent tokens for each token verified by the target model. This cache, constructed from the highest logits observed during past generation steps, serves as a high-quality source of draft candidates for future occurrences of the same token.

- **Hybrid Method**
  - *SAM with EAGLE-3 (SAM[EAGLE-3])* (Hu et al., 2025): This approach is a hybrid strategy that dynamically switches between the SAM and EAGLE-3 frameworks. When SAM identifies a matched suffix that is too short to be beneficial for speculation, the system defaults to EAGLE-3.

# 4 SPECULATIVE DECODING FOR REASONING UNDER TEST-TIME SCALING

## 4.1 EXPERIMENT SETUP

We investigate the effect of speculative decoding methods in reasoning models DeepSeek-R1-Distill-Llama-8B (DSL-8B) (DeepSeek-AI et al., 2025) and Qwen3-8B (QW3-8B) (Yang et al., 2025), employing the widely-used test-time scaling paradigms of multi-round thinking (with 2 rounds) and BoN (with 4 trajectories). We evaluate methods on the reasoning datasets AIME24/25, MATH500, and GPQA (detailed in Section 3.2). The evaluation pipeline from prior work (Xia et al., 2024) is adopted to ensure fair comparison. For the reasoning models, we use the float16 data type and apply greedy/stochastic decoding with a batch size of 1, following the original settings of the evaluated methods. The evaluation metrics include: *Mean Accepted Tokens (MAT)*, which measures the average number of tokens accepted per speculative decoding step, and the *Walltime Speedup Ratio (Speed)*, which quantifies the inference efficiency gain relative to vanilla autoregressive decoding. The evaluated methods including model-based speculative sampling (e.g., SpS), training-based speculative decoding (e.g., EAGLE-3), and N-gram-based methods (e.g., PLD, REST, Lookahead, PIA, SAM, and Recycling). More implementation details are provided in the Appendix A.2.

## 4.2 DISCUSSION ON RESULTS

The performance of different methods under multi-round thinking is shown in Table 3. As DSL-8B does not have a smaller model variant, results for the combination of SpS and DSL-8B are not provided. For QW3-8B, we evaluate SpS using Qwen3-0.6B (QW3-0.6B) (Yang et al., 2025) as the draft model. Additionally, under the speculative sampling, methods such as Lookahead do not support non-greedy decoding, hence their non-greedy variants are not included in Table 3. Table 4 illustrates the performance of speculative decoding methods under the BoN. Since BoN requires sampling to generate diverse reasoning traces, we do not evaluate greedy decoding in this context.

### 4.2.1 TRAINING-BASED METHOD

• **The adaptability of the training-based method is constrained by its training process.** For the training-based method, we evaluate the SOTA EAGLE-3, which supports reasoning model acceleration. Other training-based methods, such as EAGLE-1/2 (Li et al., 2024b;a) and Medusa (Cai et al., 2024), are not included in our comparison because they do not provide corresponding draft models for reasoning models. For DSL-8B, we used a draft model trained by the EAGLE-3 team, while for QW3-8B, we use a draft model provided by the Tengyunw team. As shown in Table 3 and Table 4, EAGLE-3 achieves considerable speedup across different scenarios. However, we also observe some performance deviations from expectations. Specifically, the average speedup ratio $(T = 0)$ in Table 3 decreases by 34% when running EAGLE-3 on DSL-8B compared to QW3-8B

---

https://huggingface.co/yuhuili/EAGLE3-DeepSeek-R1-Distill-LLaMA-8B
https://huggingface.co/Tengyunw/qwen3_8b_eagle3

Table 3: Performance comparison of speculative decoding methods for reasoning models under multi-round thinking framework with different temperature $T$ ( **Best** , Second Best ).

| Model | Bench
Method | AIME24 | | AIME25 | | MATH500 | | GPQA | | Overall | |
|---|---|---|---|---|---|---|---|---|---|---|---|
| | | MAT | Speed | MAT | Speed | MAT | Speed | MAT | Speed | MAT | Speed |
| DSL-8B
($T=0$) | AR | 1.00 | 1.00× | 1.00 | 1.00× | 1.00 | 1.00× | 1.00 | 1.00× | 1.00 | 1.00× |
| | EAGLE-3 | 2.26 | 1.56× | 2.24 | 1.53× | 2.63 | 2.86× | 2.46 | 1.61× | 2.35 | 1.93× |
| | SAM[EAGLE-3] | **3.91** | **3.09×** | **4.50** | **3.85×** | **4.94** | **4.11×** | **7.00** | **4.70×** | **4.72** | **3.97×** |
| | SAM | 2.64 | 2.41× | 3.05 | 2.96× | 2.60 | 2.14× | 3.57 | 3.20× | 2.93 | 2.66× |
| | Recycling | 2.98 | 2.08× | 2.98 | 2.05× | 2.97 | 2.18× | 3.02 | 2.08× | 2.99 | 2.10× |
| | PLD | 2.14 | 1.76× | 2.38 | 1.89× | 2.25 | 1.71× | 2.65 | 2.02× | 2.33 | 1.84× |
| | REST | 1.33 | 1.00× | 1.35 | 0.95× | 1.37 | 1.08× | 1.31 | 1.00× | 1.34 | 1.01× |
| | Lookahead | 2.21 | 1.59× | 2.26 | 1.53× | 2.28 | 1.55× | 2.44 | 1.62× | 2.28 | 1.57× |
| | PIA | 1.84 | 1.51× | 1.96 | 1.63× | 1.91 | 1.48× | 2.15 | 1.63× | 1.95 | 1.56× |
| DSL-8B
($T=0.6$) | AR | 1.00 | 1.00× | 1.00 | 1.00× | 1.00 | 1.00× | 1.00 | 1.00× | 1.00 | 1.00× |
| | EAGLE-3 | 2.21 | 1.31× | 2.17 | 1.50× | 3.21 | 3.02× | 2.45 | 1.61× | 2.33 | 1.91× |
| | SAM[EAGLE-3] | 2.54 | 1.82× | 2.48 | 1.89× | 3.28 | 3.13× | 2.86 | 2.15× | 2.66 | **2.29×** |
| | SAM | 1.84 | 1.65× | 1.84 | 1.65× | 1.93 | 1.71× | 1.93 | 1.73× | 1.87 | 1.69× |
| | Recycling | **2.84** | **1.91×** | **2.83** | **1.93×** | 2.88 | 2.03× | 2.81 | 1.98× | **2.84** | 1.96× |
| | REST | 1.35 | 1.00× | 1.37 | 0.97× | 1.38 | 1.03× | 1.35 | 1.00× | 1.36 | 1.00× |
| | PIA | 1.58 | 1.26× | 1.58 | 1.27× | 1.63 | 1.39× | 1.68 | 1.36× | 1.61 | 1.33× |
| QW3-8B
($T=0$) | AR | 1.00 | 1.00× | 1.00 | 1.00× | 1.00 | 1.00× | 1.00 | 1.00× | 1.00 | 1.00× |
| | EAGLE-3 | 4.31 | 2.80× | 4.37 | 2.87× | 4.50 | 3.04× | 4.39 | 2.91× | 4.38 | 2.91× |
| | SAM[EAGLE-3] | 4.40 | **3.08×** | 4.47 | **3.18×** | 4.82 | **3.47×** | **5.90** | **4.17×** | 4.76 | **3.49×** |
| | SAM | 2.15 | 1.97× | 2.27 | 2.07× | 2.18 | 1.95× | 3.19 | 3.11× | 2.37 | 2.28× |
| | Recycling | 3.02 | 2.08× | 2.98 | 2.13× | 3.00 | 2.20× | 3.07 | 2.17× | 3.01 | 2.15× |
| | PLD | 1.95 | 1.61× | 1.97 | 1.66× | 1.91 | 1.63× | 2.49 | 2.04× | 2.05 | 1.74× |
| | SpS | **7.40** | 0.93× | **7.50** | 0.96× | **6.69** | 0.77× | **6.33** | 0.83× | **7.07** | 0.87× |
| | REST | 1.39 | 1.12× | 1.39 | 1.14× | 1.42 | 1.16× | 1.33 | 1.09× | 1.38 | 1.13× |
| | Lookahead | 2.14 | 1.54× | 2.12 | 1.54× | 2.10 | 1.58× | 2.28 | 1.69× | 2.15 | 1.59× |
| | PIA | 1.83 | 1.48× | 1.96 | 1.68× | 1.89 | 1.63× | 2.08 | 1.72× | 1.93 | 1.63× |
| QW3-8B
($T=0.6$) | AR | 1.00 | 1.00× | 1.00 | 1.00× | 1.00 | 1.00× | 1.00 | 1.00× | 1.00 | 1.00× |
| | EAGLE-3 | 4.11 | 2.61× | 4.21 | 2.71× | 4.32 | 2.87× | 4.03 | 2.68× | 4.16 | **2.73×** |
| | SAM[EAGLE-3] | 3.92 | **2.68×** | 3.93 | **2.72×** | 4.19 | **2.93×** | 3.98 | **2.82×** | 3.97 | **2.79×** |
| | SAM | 1.91 | 1.73× | 1.95 | 1.78× | 1.93 | 1.74× | 2.09 | 1.86× | 1.96 | 1.78× |
| | Recycling | 2.88 | 2.00× | 2.92 | 2.04× | 2.90 | 2.11× | 2.92 | 2.08× | 2.90 | 2.06× |
| | SpS | **6.22** | 0.91× | **6.54** | 0.95× | **6.17** | 0.80× | **6.32** | 0.86× | **6.34** | 0.88× |
| | REST | 1.41 | 1.08× | 1.41 | 1.10× | 1.42 | 1.12× | 1.36 | 1.03× | 1.40 | 1.08× |
| | PIA | 1.69 | 1.40× | 1.70 | 1.45× | 1.71 | 1.45× | 1.75 | 1.48× | 1.71 | 1.44× |

($2.91\times \rightarrow 1.93\times$). Moreover, the overall MAT ($T=0$) of EAGLE-3 on DSL-8B in Table 3 is approximately half of that on QW3-8B ($4.38 \rightarrow 2.35$). Similar trends are observed in the temperature sampling setting and Table 4. These differences exceed the typical performance variation that can be attributed to intrinsic model factors such as differences in vocabulary or architectural design. We hypothesize two potential reasons for this phenomenon: (i) As noted by the Tengyunw team, they used 600k samples to train the draft model for QW3-8B, which is larger than the dataset used by the EAGLE-3 team for training the DSL-8B draft model. A *larger training set can improve the alignment* between the draft and target models' output distributions, leading to better performance. (ii) The original EAGLE-3 evaluation focused on short-generation scenarios. We conjecture that the training process for the draft model may *not adequately model long-generation settings* (for example, larger position indices may receive insufficient training updates), leading to performance degradation in long-generation scenarios of the reasoning model.

• **Training-based method is insensitive to temperature.** As the temperature $T$ increases, the target model's output distribution becomes more uniform, making it more challenging for the draft model to accurately predict the subsequent tokens compared to greedy decoding. Despite this increased difficulty, the EAGLE-3 demonstrates considerable robustness to temperature variations. For instance, under the multi-round thinking paradigm, when the $T$ is raised from 0 to 0.6, the average speedup of EAGLE-3 on QW3-8B decreases by only 6%. This relative stability suggests that the training-based method has the potential to generalize across different levels of decoding stochasticity.

Table 4: Performance comparison of speculative decoding methods for reasoning models under BoN framework with different temperature $T$ ( **Best** , Second Best ).

| Model | Bench | AIME24 | | AIME25 | | MATH500 | | GPQA | | Overall | |
|---|---|---|---|---|---|---|---|---|---|---|---|
| | Method | MAT | Speed | MAT | Speed | MAT | Speed | MAT | Speed | MAT | Speed |
| DSL-8B $(T = 0.6)$ | AR | 1.00 | 1.00× | 1.00 | 1.00× | 1.00 | 1.00× | 1.00 | 1.00× | 1.00 | 1.00× |
| | EAGLE-3 | 2.19 | 1.17× | 2.18 | 1.43× | 3.09 | 2.71× | 2.37 | 1.52× | 2.31 | 1.74× |
| | SAM[EAGLE-3] | 2.44 | 1.53× | 2.41 | 1.72× | 3.31 | 3.05× | 2.72 | 1.96× | 2.56 | 2.10× |
| | SAM | 1.89 | 1.62× | 1.86 | 1.62× | 1.97 | 1.77× | 2.00 | 1.75× | 1.91 | 1.70× |
| | Recycling | 2.81 | 1.91× | 2.80 | 1.86× | 2.88 | 2.04× | 2.80 | 1.93× | 2.82 | 1.94× |
| | REST | 1.36 | 0.99× | 1.38 | 0.99× | 1.40 | 1.07× | 1.33 | 0.98× | 1.37 | 1.01× |
| | PIA | 1.53 | 1.18× | 1.55 | 1.21× | 1.62 | 1.35× | 1.68 | 1.37× | 1.57 | 1.20× |
| QW3-8B $(T = 0.6)$ | AR | 1.00 | 1.00× | 1.00 | 1.00× | 1.00 | 1.00× | 1.00 | 1.00× | 1.00 | 1.00× |
| | EAGLE-3 | 4.09 | 2.63× | 4.17 | 2.64× | 4.30 | 2.85× | 4.00 | 2.64× | 4.12 | 2.70× |
| | SAM[EAGLE-3] | 3.81 | 2.66× | 3.91 | 2.72× | 4.18 | 2.99× | 3.87 | 2.68× | 3.91 | 2.77× |
| | SAM | 2.05 | 1.88× | 2.04 | 1.88× | 2.17 | 2.01× | 2.19 | 2.00× | 2.10 | 1.95× |
| | Recycling | 2.90 | 2.03× | 2.88 | 1.99× | 2.90 | 2.06× | 2.91 | 2.04× | 2.89 | 2.03× |
| | SpS | 6.05 | 0.93× | 6.22 | 0.95× | 6.01 | 0.80× | 5.87 | 0.86× | 6.05 | 0.88× |
| | REST | 1.41 | 1.12× | 1.40 | 1.10× | 1.42 | 1.11× | 1.36 | 1.07× | 1.40 | 1.10× |
| | PIA | 1.67 | 1.38× | 1.70 | 1.41× | 1.72 | 1.44× | 1.76 | 1.45× | 1.70 | 1.42× |

> **Takeaway 1:** While the training-based methods demonstrate promising acceleration potential, their performance is inherently tied to the training process, raising practical concerns about their adaptability to diverse reasoning scenarios.

### 4.2.2 N-GRAM-BASED METHOD — TOKEN N-GRAM

• **Among token N-gram methods, which identify repeated patterns at the token level, SAM most efficiently captures redundant patterns.** For token N-gram methods, we assess SAM, PLD, REST, Lookahead, and PIA. Overall, SAM achieves the best performance among these token N-gram methods, even rivaling the training-based method EAGLE-3. For instance, under the greedy decoding setting with multi-round thinking, SAM achieves a 38% higher overall speedup ratio than EAGLE-3 on DSL-8B (2.66×→1.93×). This advantage primarily stems from the high efficiency of SAM in suffix matching, as well as the repetitive patterns inherent in reasoning traces. Other methods, such as PLD, PIA, and Lookahead, also demonstrate considerable potential, achieving around 1.5× speedup across various test-time scaling paradigms. However, due to the higher time complexity of their N-gram matching mechanisms, they still fall short of SAM in acceleration. REST performs relatively poorly. We hypothesize this is because the external chat datastore used by REST has limited overlap with reasoning datasets, resulting in fewer effective token reuse opportunities.

• **Sampling temperature significantly affects token N-gram-based methods.** We note that token N-gram methods exhibit notable sensitivity to sampling temperature. Taking SAM as an example, under the multi-round thinking paradigm, when the temperature increases from 0 to 0.6, its overall speedup on QW3-8B drops by approximately 22% (from 2.28× to 1.78×), and on MAT it decreases from 2.37 to 1.96, representing a reduction of about 17%. This highlights a key trade-off: while these methods are particularly effective at leveraging the repetitive patterns in reasoning model outputs under test-time scaling, their acceleration benefits tend to decrease as output diversity increases.

> **Takeaway 2:** Token N-gram methods excel at capturing redundancy in reasoning traces under test-time scaling, yet this advantage comes with sensitivity to sampling temperature variations.

### 4.2.3 N-GRAM-BASED METHOD — PROBABILISTIC N-GRAM

• **Probabilistic N-gram method Recycling, which captures the co-occurrence of top-K next tokens through logits, achieves the highest MAT among N-gram-based methods.** As shown in Table 3, Recycling achieves performance comparable to the strongest retrieval-based method SAM. Under greedy decoding, Recycling slightly lags behind SAM in speedup ratio, yet attains a higher MAT. We attribute this to Recycling's higher computational complexity in draft generation. Specifically, Recycling employs a tree-structured speculative verification, resulting in significantly

higher computational latency compared to SAM's linear verification. Moreover, Recycling speculates 81 tokens, which is nearly twice the number of tokens speculated by SAM (40 tokens), which further contributes to its overall computational overhead.

• **Probabilistic N-gram method exhibits low sensitivity to temperature sampling.** As the temperature increases from 0 to 0.6 in the multi-round thinking paradigm, Recycling's overall speedup ratio on QW3-8B decreases only from 2.15 to 2.06—a decline of less than 5%. In contrast, SAM's speedup drops by 22% under the same temperature shift. We hypothesize that this robustness stems from Recycling's preservation of approximate next-token prediction logits, specifically the top-k vocabulary distribution, which enables substantial overlap between its tree-based speculative tokens and the target model's sampled tokens even as temperature varies.

> **Takeaway 3:** The probabilistic N-gram method offers superior stability and higher draft token acceptance rates among N-gram-based methods, yet its computational bottleneck prevents acceptance gains from translating into proportional speedup.

### 4.2.4 HYBRID APPROACH: INTEGRATING TRAINING-BASED AND N-GRAM METHOD

• **Hybrid speculative decoding can leverage the strengths of different methods.** For the hybrid approach, we conduct evaluations on SAM[EAGLE-3], which integrates EAGLE-3 and SAM (detailed hybrid strategies are provided in Section 3.3). As shown in Table 3, SAM[EAGLE-3] achieves the highest overall speedup ratio across all scenarios. Under the greedy decoding setting, the SAM[EAGLE-3] demonstrates significant improvements over individual methods. Within the multi-turn thinking paradigm, SAM[EAGLE-3] delivers speedup gains of 20% and 53% compared to EAGLE-3 and SAM on QW3-8B, respectively. The reason behind this improvement is straightforward: SAM[EAGLE-3] preserves EAGLE-3's powerful token speculation capabilities while addressing its limitation in capturing repetitive patterns that emerge during dynamic generation.

• **Hybrid speculative decoding retains the limitations of its components.** While inheriting the strengths of SAM, SAM[EAGLE-3] also retains its key drawback, namely, a high sensitivity to sampling temperature. When the sampling temperature increases from 0 to 0.6, the speedup ratio of SAM[EAGLE-3] under the multi-round thinking paradigm decreases by 42% on DSL-8B ($3.97\times \rightarrow 2.29\times$) and by 20% on QW3-8B ($3.49\times \rightarrow 2.79\times$), respectively. Furthermore, as observed in Table 3 and Table 4, the performance gain of SAM[EAGLE-3] over EAGLE-3 is less significant. This indicates that under temperature sampling with $T = 0.6$, the accuracy of using SAM for speculative decoding is not satisfactory, leading to the overall performance of SAM[EAGLE-3] being largely dependent on that of EAGLE-3. Despite certain limitations, the consistent SOTA performance of SAM[EAGLE-3] across various evaluation scenarios underscores its practical effectiveness. By integrating the strengths of different methods, the hybrid method achieves well-balanced performance across various settings, highlighting its potential for reasoning with test-time scaling.

> **Takeaway 4:** Hybrid speculative decoding uniquely combines the semantic alignment strengths of training-based methods with the repetitive pattern capture capabilities of N-gram methods, unlocking distinct potential for reasoning with test-time scaling.

### 4.2.5 MODEL-BASED METHOD

• **Model-based method SpS generates high-quality drafts, but faces computational bottlenecks.** As shown in Table 3 and Table 4, SpS achieves the highest MAT. For example, in the multi-round thinking scenario with QW3-8B ($T = 0$), SpS's overall MAT is 1.49 times that of the second-best method. This superiority primarily stems from SpS employing a smaller, homogenous model as the draft model, which ensures a high degree of alignment between the output distributions of the draft and target models. However, this high MAT does not translate into proportional speedup. The primary bottleneck lies in the computational cost of generating drafts. Despite being smaller, such as a 0.6B model, the draft model is still too heavy compared to the target models like 8B, particularly under tight latency constraints. As a result, the time saved by reducing target model evaluations is largely offset by the overhead of running the draft model, which limits the overall acceleration.

**Takeaway 5:** Model-based method SpS achieves high MAT by leveraging a well-aligned draft model, but its acceleration is limited due to the relatively small size difference between the draft model (e.g., 0.6B) and the target model (e.g., 8B).

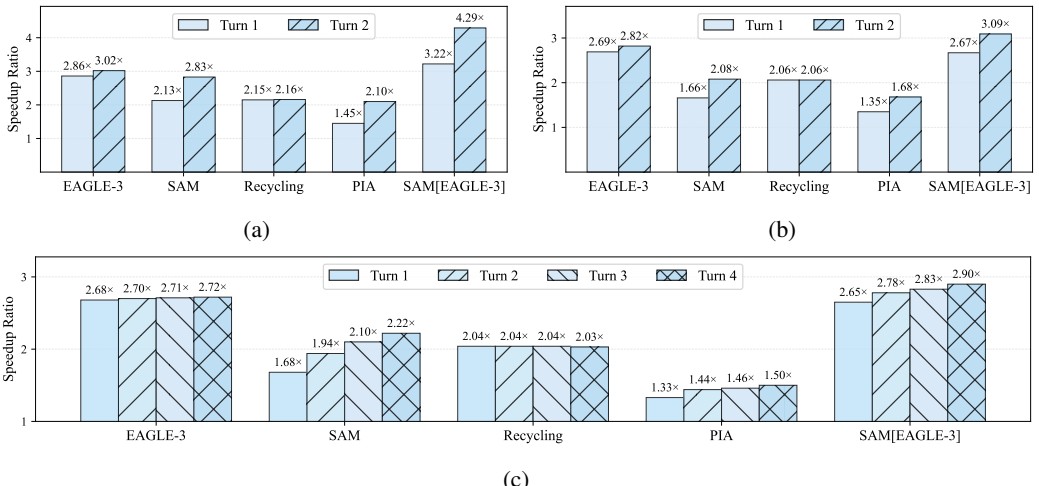

Figure 2: Speedup ratio across different turns under different settings across various temperature $T$. (a) Multi-round thinking ($T = 0$). (b) Multi-round thinking ($T = 0.6$). (c) BoN ($T = 0.6$).

### 4.2.6 ANALYSIS OF SPEED SCALING OVER TURNS

• **Token N-gram methods achieve progressive acceleration**. Figure 2 compares the acceleration ratios of different methods across reasoning turns. A key observation is that Token N-gram methods exhibit a steadily increasing acceleration ratio as the number of turns grows. As illustrated in Figure 2a, SAM and PIA show acceleration ratio increases of 33% ($2.13\times \to 2.83\times$) and 45% ($1.45\times \to 2.10\times$) in the second turn over the first, respectively. This trend demonstrates their ability to effectively identify and exploit repetitive patterns that naturally emerge during test-time scaling. Specifically, by retrieving and reusing relevant intermediate results or reasoning traces from prior turns, these methods reduce the number of required decoding steps in subsequent turns, leading to progressively greater efficiency gains over time. In contrast, other methods exhibit nearly constant acceleration ratios across turns, indicating that each turn is processed independently without leveraging reusable patterns from prior decoding steps. Further analysis can be found in Appendix A.3.

**Takeaway 6:** Token N-gram methods PIA and SAM achieve progressive acceleration by reusing past computations, enabling efficiency that scales across turns.

## 5 CONCLUSION

In this paper, we introduce the first benchmark for speculative decoding in accelerating LLM test-time scaling. Specifically, we evaluate a diverse range of methods, including model-based, training-based, and N-gram-based approaches, under prominent frameworks such as BoN sampling and multi-round thinking. Through extensive experiments, we find that simple N-gram-based methods show strong potential in accelerating test-time scaling by effectively capturing repetitive reasoning patterns. This highlights the value of integrating N-gram-based methods with other categories of approaches, thereby balancing acceleration for both repetitive and diverse reasoning in test-time scaling. We hope this benchmark serves as a foundation for advancing speculative decoding techniques tailored to test-time scaling, fostering more efficient and practical LLM reasoning.

## ACKNOWLEDGEMENTS

This work was supported by the Early Career Scheme (No. CityU 21219323) and the General Research Fund (No. CityU 11220324) of the University Grants Committee (UGC), the NSFC Young Scientists Fund (No. 9240127), and the Donation for Research Projects (No. 9220187 and No. 9229164).

## ETHICS STATEMENT

This benchmark's dataset collection adheres to strict ethical principles and responsible research practices. All data sources are derived exclusively from publicly available materials for academic research purposes only, with no commercial applications intended. We maintain full compliance with copyright laws and respect intellectual property rights of original content creators. We have implemented comprehensive screening procedures to ensure datasets are free from harmful content, including discriminatory material, violent imagery, and personally identifiable information. This research initiative aims to advance scientific knowledge and contribute positively to societal welfare while protecting individual privacy and promoting social responsibility in AI research.

## REPRODUCIBILITY STATEMENT

To ensure reproducibility of our work, we provide comprehensive implementation details for benchmark construction in Section 3, Section 4.1, and Appendix A.2. All open-source LLMs used in this study are accessed through Hugging Face (`https://huggingface.co/`), and we strictly adhere to their respective terms and conditions for use. To facilitate the reproduction of our experiments, the code and data are provided in `https://github.com/sunshy-1/SpecTTS-Bench`.

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

# A    APPENDIX

## A.1    ADDITIONAL RELATED WORK

### A.1.1    TRAINING-FREE FRAMEWORKS FOR TEST-TIME SCALING

Models can acquire reasoning abilities through specialized training that teaches them to generate long chains of thought internally (Chen et al., 2025; Sun et al., 2023; Yu et al., 2025a; Yue et al., 2025). Complementing these trained capabilities are external test-time scaling frameworks. These techniques implement an outer control loop around a base model to enhance its reasoning performance without altering the model's parameters (Yu et al., 2025b; Yuan et al., 2025; Setlur et al., 2025). The advantage of these external methods lies in their universal applicability: they can augment any model's performance, including those that already incorporate built-in reasoning capabilities. This work examines two primary categories of external training-free scaling approaches:

- **Sampling-Based Methods:** The most prominent approach in this category is BoN sampling (Snell et al., 2025), where a model generates $N$ candidate outputs, and a verifier or the model itself selects the most plausible one. A more sophisticated variant, Self-Consistency (Wang et al., 2023), generates multiple reasoning paths and selects the final answer by a majority vote, demonstrating significant improvements in arithmetic and commonsense reasoning. These methods operate on the principle that a model is more likely to generate a correct reasoning path than an incorrect one, and sampling multiple paths increases the probability of finding a valid solution.

- **Iterative Refinement:** This class of methods involves a multi-step process where the model progressively improves its own output. For instance, Self-Refine (Madaan et al., 2023) enables a model to generate an initial response, critique it, and then use that feedback to produce a refined answer in a subsequent pass. A more streamlined approach is multi-round thinking (Tian et al., 2025), which simplifies the feedback loop. Instead of generating an explicit critique, it provides only the final answer from the previous turn as context, prompting the model to re-evaluate its conclusion. This method is designed to break "cognitive inertia" by discarding the prior reasoning chain, forcing the model to approach the problem anew.

Other search algorithms, such as Tree-of-Thoughts (ToT) (Yao et al., 2023), Graph-of-Thoughts (Besta et al., 2024), and those based on Monte-Carlo Tree Search (MCTS) (Lin et al., 2025a), can be understood as sophisticated hybrids of these foundational principles. For instance, ToT explores multiple candidate reasoning steps (a form of sampling) and evaluates them to construct a solution tree, while MCTS iteratively builds and refines a search policy. By combining multi-candidate generation with progressive construction and evaluation, they represent a powerful but more complex class of techniques built upon the core ideas of sampling and refinement.

### A.1.2    EFFICIENT REASONING

While proven effective for unlocking the advanced reasoning capabilities of LLMs, test-time scaling paradigms are inherently resource-intensive. This significant increase in computational overhead and inference latency presents a substantial challenge to their practical deployment in real-world, time-sensitive applications. Efficient reasoning offers a promising path to alleviate this burden by generating smarter, more concise reasoning with less computation (Wu et al., 2025; Sui et al., 2025; Xing et al., 2025). Research in this area can be organized into three main categories. (i) Model-based efficient reasoning (Wang et al., 2025a; Lou et al., 2025), which aims to optimize full-length reasoning models into more concise variants or directly train compact models specialized for fast and effective reasoning. (ii) Reasoning output-based efficient reasoning (Hao et al., 2024; Pan et al., 2025; Lin et al., 2025b), which dynamically reduces the computational work during inference by adaptively shortening the reasoning chain and eliminating unnecessary steps. (iii) Input prompts-based efficient reasoning (Ma et al., 2025; Liang et al., 2025), which enhances efficiency by tailoring the reasoning process according to the properties of the input, such as its specified length constraints.

However, the speculative decoding methods for accelerating test-time scaling examined in this paper are fundamentally orthogonal to the aforementioned efficient reasoning methodologies (Liu et al., 2024a), representing a distinctly different paradigm for performance optimization. Recent studies have also explored speculative approaches within the context of test-time scaling, including but not limited to speculative acceleration analyses for test-time scaling (Wang et al., 2025b), logit-based N-gram speculation (Song et al., 2025), and multi-sample inference speculation (Li et al., 2025a).

However, a unified and standardized benchmark for evaluating speculative decoding within the test-time scaling setting is still lacking, which precisely constitutes the focus of our work. Fundamentally, speculative decoding constitutes a lossless acceleration technique that preserves the original model's output distribution exactly, ensuring that both the quality and inherent characteristics of generated reasoning remain completely unchanged throughout the acceleration process. This preservation of distributional fidelity stands in stark contrast to efficient reasoning approaches, which typically achieve speedup by introducing modifications to the reasoning process itself, whether through aggressive model quantization that reduces numerical precision (Frantar et al., 2022; Lin et al., 2024; Hao et al., 2025; Liu et al., 2024b), output compression techniques that sacrifice detail for brevity (Cheng & Durme, 2024; Hao et al., 2024), structured prompt engineering that constrains the reasoning pathway (Han et al., 2024; Lee et al., 2025; Aytes et al., 2025), or architectural modifications that streamline computational flow (Li et al., 2025b; Srivastava et al., 2025; Sun et al., 2025; Chen et al., 2024). More critically, speculative decoding operates exclusively at the inference execution level without fundamentally altering the underlying reasoning methodology, maintaining complete fidelity to the original model's cognitive processes, decision-making patterns, and output generation mechanisms. This approach ensures that no aspect of the original model's performance characteristics, reasoning depth, or solution quality is compromised or sacrificed in pursuit of computational efficiency, making it a particularly attractive acceleration strategy for scenarios where maintaining exact model behavior is paramount.

## A.2 Implementation Details

All experiments are conducted on a server equipped with a 16-core CPU, utilizing PyTorch 2.1.1 with CUDA 12.1. The overall evaluation pipeline follows the previous work (Xia et al., 2024). For greedy decoding with temperature $T = 0$, we employ exact match verification between the draft model and target model outputs to ensure generation consistency. For sampling scenarios where temperature $T \neq 0$, we utilize speculative sampling (Chen et al., 2023) for verification, which maintains mathematically proven identical output distributions to the target model. Given that reasoning models may generate extremely long outputs during their internal deliberation process, we implement specific constraints to ensure practical evaluation while preserving the models' reasoning capabilities. We set the maximum output token limit to 15,000 tokens and establish a thinking budget mechanism that allows up to 80% of the maximum token allocation for internal reasoning processes. For the DeepSeek-R1-Distill-Llama-8B model, we implement a forced thinking termination strategy where designated termination tokens `\n\n</think>\n\n` are inserted starting from the 12,000th generated token to compel the model to conclude its reasoning phase and provide the final response. For the Qwen3 series models, we employ a consistent termination strategy by inserting the official stop-thinking instruction starting from the 12,000th generated token, following the protocol specified in their technical report (Yang et al., 2025). Specifically, we employ the designated stop-thinking instruction, i.e., `Considering the limited time by the user, I have to give the solution based on the thinking directly now.\n</think>.\n\n`, to gracefully terminate the reasoning phase. Additionally, the detailed information of the open-source trained models used in the experiments is shown in Table 5.

| Model | Type | URL |
|---|---|---|
| DeepSeek-R1-Distill-Llama-8B | Target Model | https://huggingface.co/deepseek-ai/DeepSeek-R1-Distill-Llama-8B |
| Qwen3-4B | Target Model | https://huggingface.co/Qwen/Qwen3-4B |
| Qwen3-8B | Target Model | https://huggingface.co/Qwen/Qwen3-8B |
| Qwen3-14B | Target Model | https://huggingface.co/Qwen/Qwen3-14B |
| Qwen3-0.6B | Draft Model | https://huggingface.co/Qwen/Qwen3-0.6B |
| EAGLE3-DeepSeek-R1-Distill-LLaMA-8B | Draft Model | https://huggingface.co/yuhuili/EAGLE3-DeepSeek-R1-Distill-LLaMA-8B |
| EAGLE3-Qwen3-4B | Draft Model | https://huggingface.co/AngelSlim/Qwen3-4B_eagle3 |
| EAGLE3-Qwen3-8B | Draft Model | https://huggingface.co/Tengyunw/qwen3_8b_eagle3 |
| EAGLE3-Qwen3-14B | Draft Model | https://huggingface.co/AngelSlim/Qwen3-14B_eagle3 |

Table 5: Information about the open source trained models.

## A.3 Further Analysis

In this section, we further analyze speculative decoding in terms of computational overhead of key modules, the potential of hybrid strategies, performance on models of different scales, acceleration across reasoning turns, and visualization of inference acceleration process, offering insights into its practical efficiency and limitations.

• **Potential analysis of hybrid speculative decoding**. We present the average accept length across different suffix matching lengths and temperature settings in Figure 3. The results indicate that SAM demonstrates strong potential when the matching length exceeds 10. For instance, under greedy decoding ($T = 0$), SAM achieves an average accept length greater than 9 at a suffix matching length of 11, which is nearly twice the average accept length of EAGLE-3. However, we also observe notable fluctuations in SAM's average accept length as the matching suffix length increases. This sensitivity suggests that the performance of speculative decoding can vary significantly depending on the hybrid strategy (based on the matched suffix length). It is promising to design more refined hybrid strategies to unlock SAM's potential for long-sequence matching in test-time scaling, which will further improve the performance of the hybrid speculative decoding.

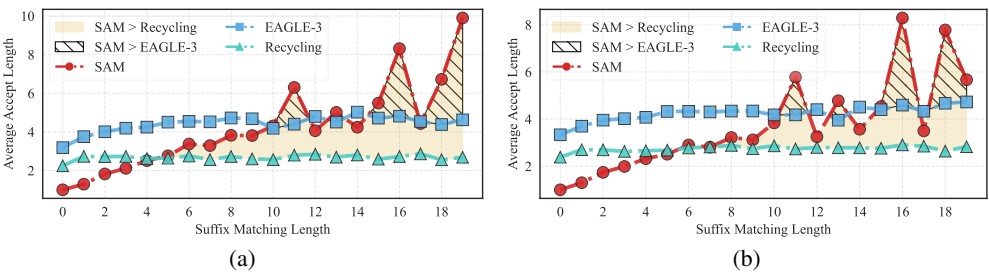

Figure 3: Average accept length under different suffix matching lengths across various temperature $T$. (a) $T = 0$. (b) $T = 0.6$.

> **Takeaway 7:** SAM's capability to capture redundant patterns in the test-time scaling paradigm remains underexploited in hybrid speculative decoding, highlighting the need for more refined and dynamic hybrid strategies to fully unlock its potential.

• **Time consumption proportions across modules in speculative decoding.** Figure 4 illustrates the proportion of time consumed by each stage within PIA, EAGLE-3, Recycling, and SAM. During draft generation, the model produces a sequence of tokens based on the current state. The decoding stage processes this draft sequence to compute representations and logits for each position. In the verification phase, the model determines which tokens in the draft are correct by comparing predicted probabilities given by the target model. Finally, the updating stage accepts the verified correct tokens and updates the model state accordingly, preparing for the next iteration. As shown in the Figure 4, PIA achieves the lowest proportion of draft generation time, which benefits from its pruning strategy for the N-gram draft tree. In contrast, EAGLE-3 exhibits the highest draft generation time proportion, reaching up to 35.7%, as it requires invoking an additional draft model that is more time-consuming than N-gram methods. Recycling demonstrates the highest update time proportion among all methods, attributed to its need to dynamically maintain a top-k next token list throughout the generation process. For SAM, the majority of time consumption is concentrated in the target model's decoding phase,

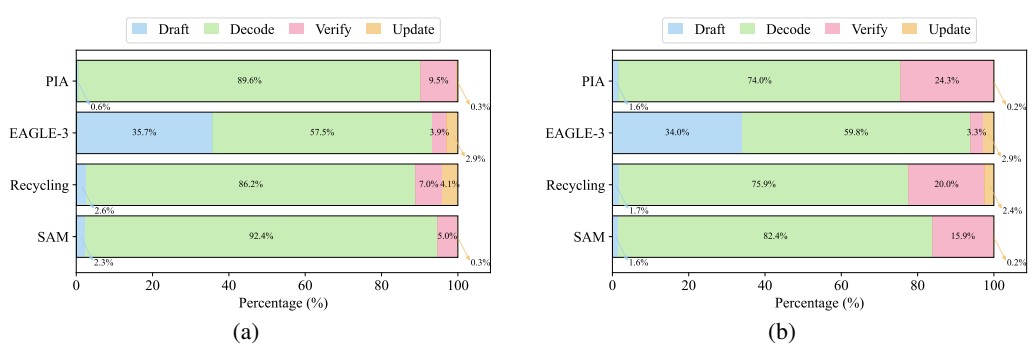

Figure 4: Analysis of time consumption proportions across modules in speculative decoding with different model sizes. (a) QW3-8B. (b) QW3-14B.

Table 6: Performance comparison of speculative decoding methods on reasoning models with different scales across various temperature $T$ ( **Best** , Second Best ).

| Model | Bench | AIME24 | | AIME25 | | MATH500 | | GPQA | | Overall | |
|---|---|---|---|---|---|---|---|---|---|---|---|
| | Method | MAT | Speed | MAT | Speed | MAT | Speed | MAT | Speed | MAT | Speed |
| *Under the Multi-turn Thinking Paradigm* | | | | | | | | | | | |
| QW3-4B $(T=0.6)$ | AR | 1.00 | 1.00× | 1.00 | 1.00× | 1.00 | 1.00× | 1.00 | 1.00× | 1.00 | 1.00× |
| | EAGLE-3 | **3.45** | 2.28× | **3.48** | 2.22× | **3.54** | 2.38× | 3.31 | 2.20× | **3.44** | 2.27× |
| | SAM[EAGLE-3] | 3.40 | **2.37×** | 3.40 | 2.28× | 3.50 | **2.55×** | **3.47** | **2.39×** | 3.43 | **2.40×** |
| | SAM | 1.99 | 1.92× | 1.94 | 1.81× | 1.95 | 1.82× | 2.19 | 1.99× | 2.01 | 1.89× |
| | Recycling | 2.92 | 2.20× | 2.91 | 2.11× | 2.92 | 2.38× | 2.94 | 2.25× | 2.92 | 2.24× |
| | REST | 1.41 | 1.12× | 1.42 | 1.10× | 1.44 | 1.15× | 1.35 | 1.04× | 1.40 | 1.10× |
| | PIA | 1.71 | 1.45× | 1.70 | 1.39× | 1.69 | 1.42× | 1.78 | 1.46× | 1.72 | 1.43× |
| QW3-14B $(T=0.6)$ | AR | 1.00 | 1.00× | 1.00 | 1.00× | 1.00 | 1.00× | 1.00 | 1.00× | 1.00 | 1.00× |
| | EAGLE-3 | 3.28 | **2.26×** | 3.38 | **2.31×** | 3.36 | 2.27× | 3.05 | 2.10× | 3.28 | 2.23× |
| | SAM[EAGLE-3] | 3.18 | 2.24× | 3.26 | 2.29× | 3.31 | **2.31×** | 3.12 | 2.15× | 3.21 | **2.25×** |
| | SAM | 1.84 | 1.70× | 1.86 | 1.72× | 1.85 | 1.66× | 1.90 | 1.73× | 1.86 | 1.70× |
| | Recycling | 2.86 | 2.10× | 2.86 | 2.08× | 2.89 | 2.17× | 2.85 | 2.13× | 2.86 | 2.12× |
| | REST | 1.41 | 1.17× | 1.40 | 1.17× | 1.42 | 1.17× | 1.33 | 1.11× | 1.39 | 1.15× |
| | SpS | **5.78** | 1.22× | **5.96** | 1.25× | **5.73** | 1.11× | **4.59** | 1.02× | **5.59** | 1.14× |
| | PIA | 1.62 | 1.43× | 1.69 | 1.44× | 1.66 | 1.45× | 1.69 | 1.46× | 1.66 | 1.45× |
| *Under the BoN Paradigm* | | | | | | | | | | | |
| QW3-4B $(T=0.6)$ | AR | 1.00 | 1.00× | 1.00 | 1.00× | 1.00 | 1.00× | 1.00 | 1.00× | 1.00 | 1.00× |
| | EAGLE-3 | **3.45** | 2.24× | **3.48** | 2.23× | 3.52 | 2.31× | 3.33 | 2.11× | **3.44** | 2.22× |
| | SAM[EAGLE-3] | 3.37 | **2.37×** | 3.37 | 2.33× | **3.58** | **2.61×** | **3.38** | **2.36×** | 3.40 | **2.42×** |
| | SAM | 2.12 | 1.94× | 2.13 | 1.92× | 2.22 | 2.07× | 2.32 | 2.07× | 2.18 | 2.00× |
| | Recycling | 2.92 | 2.14× | 2.91 | 2.11× | 2.92 | 2.31× | 2.96 | 2.20× | 2.93 | 2.19× |
| | REST | 1.42 | 1.11× | 1.41 | 1.11× | 1.44 | 1.16× | 1.38 | 1.03× | 1.41 | 1.10× |
| | PIA | 1.70 | 1.40× | 1.72 | 1.41× | 1.75 | 1.50× | 1.83 | 1.51× | 1.74 | 1.46× |
| QW3-14B $(T=0.6)$ | AR | 1.00 | 1.00× | 1.00 | 1.00× | 1.00 | 1.00× | 1.00 | 1.00× | 1.00 | 1.00× |
| | EAGLE-3 | 3.28 | 2.27× | 3.31 | 2.27× | 3.33 | 2.27× | 3.04 | 2.07× | 3.24 | 2.22× |
| | SAM[EAGLE-3] | 3.21 | **2.33×** | 3.22 | **2.34×** | 3.35 | **2.47×** | 3.10 | 2.20× | 3.21 | **2.33×** |
| | SAM | 1.98 | 1.84× | 1.99 | 1.87× | 2.09 | 1.93× | 2.05 | 1.85× | 2.01 | 1.88× |
| | Recycling | 2.86 | 2.12× | 2.86 | 2.10× | 2.89 | 2.20× | 2.87 | 2.15× | 2.87 | 2.15× |
| | SpS | **5.76** | 1.22× | **5.76** | 1.24× | **5.62** | 1.09× | **4.87** | 1.04× | **5.53** | 1.14× |
| | REST | 1.40 | 1.16× | 1.40 | 1.16× | 1.42 | 1.18× | 1.34 | 1.11× | 1.39 | 1.15× |
| | PIA | 1.64 | 1.42× | 1.69 | 1.44× | 1.70 | 1.46× | 1.73 | 1.45× | 1.68 | 1.45× |

which benefits from its advanced suffix automaton data structure. This approach finds the longest suffix that matches the sequence in a suffix automaton at each step of the generation with an average $\mathcal{O}(1)$ time complexity, while the linear drafting strategy further reduces computational overhead, significantly minimizing the time spent in other components.

> **Takeaway 8:** N-gram-based methods, such as SAM, incur lower draft generation time overhead, enabling a greater share of computational resources to be allocated to the decoding phase.

• **Analysis of speculative decoding on models with different scales.** In addition to evaluating Qwen3-8B models, we further investigate the effectiveness of speculative decoding methods across different model scales (as shown in Table 6), including the smaller Qwen3-4B and larger models such as Qwen3-14B. Among them, SAM[EAGLE-3] achieves the best performance across all model sizes, demonstrating the potential of hybrid speculative decoding. We also observe that EAGLE shows slight performance degradation when accelerating 4B/14B models compared to the 8B scenario. This stems from the training process, as the EAGLE draft models for 4B/14B come from a different team AngelSlim, highlighting how training-based speculative decoding methods are highly dependent on the draft model's training quality. Benefiting from the increased model size, we find that the SpS for the 14B model shows significant improvement compared to the 8B model. This is primarily because the 14B target model is proportionally much larger relative to the 0.6B draft model than the 8B model is, enabling more effective speculation.

• **Analysis of overall performance**. Figure 5 presents a comprehensive radar chart comparing various speculative decoding methods on the Qwen3-8B model under multi-turn thinking with greedy decoding, displaying both overall MAT and Speedup Ratio across multiple datasets. The most striking observation is that SpS demonstrates exceptional performance with its MAT scores essentially encompassing all other methods, which can be attributed to the high alignment between SpS's draft

https://huggingface.co/AngelSlim

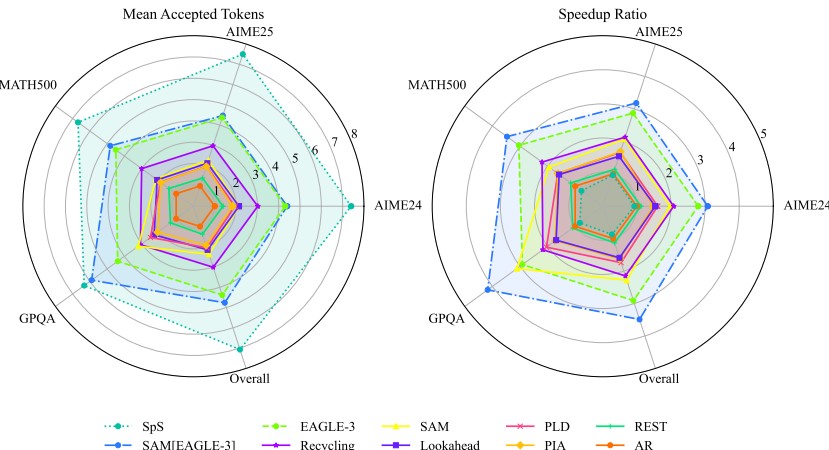

Figure 5: The performance of various speculative decoding methods on Qwen3-8B under the multi-turn thinking framework with temperature $T = 0$.

model and target model. This comprehensive coverage underscores that there remains substantial untapped potential in model alignment for training-based speculative decoding methods. Meanwhile, our novel hybrid combination SAM[EAGLE-3] exhibits considerable promise, achieving the second-highest MAT performance while securing the highest speedup ratio among all evaluated methods. This dual excellence validates our hypothesis regarding the necessity of combining repetition pattern capture through the N-gram-based methods (SAM component) and semantic alignment through the training-based approaches (EAGLE-3 component) in test-time scaling acceleration. The success of SAM[EAGLE-3] demonstrates that strategically combining different speculative decoding paradigms can yield superior performance compared to individual approaches, leveraging both the efficiency of pattern-based prediction for repetitive sequences and the semantic understanding of alignment-based methods for complex reasoning. Moreover, our other findings are summrized as follows:

- While speculative decoding relies on training an auxiliary draft model demonstrates promising acceleration potential, its performance is **inherently tied to the training process**, raising practical concerns about its adaptability to diverse reasoning scenarios.

- Methods leveraging N-gram patterns excel at capturing redundancy within test-time scaling, yet this advantage is accompanied by **the sensitivity to sampling temperature**.

- Hybrid speculative decoding uniquely **combines** methods that train an auxiliary draft model for **semantic alignment** with those leveraging **N-gram patterns to capture repetition**, unlocking distinct potential for reasoning under test-time scaling.

- Methods that store N-gram patterns during generation can achieve progressive acceleration by reusing past computations, **enabling efficiency that scales across turns**.

- In hybrid speculative decoding, the acceleration from reusing repetitive token sequences is **limited by suboptimal integration strategies**, calling for more dynamic approaches to realize its potential under test-time scaling.

- Methods leveraging N-gram patterns incur **lower draft generation time overhead**, enabling a greater share of computational resources to be allocated to the decoding phase.

• **Analysis of generalization to other domains**. To assess the generalizability of our findings, the evaluation is extended to additional domains, including code generation LiveCodeBench (Jain et al., 2025) and summarization CNN/Daily Mail (Nallapati et al., 2016). The results are given in Table 7. It can be observed that on the code and summarization datasets, SAM[EAGLE-3] consistently achieves the highest speedup across nearly all scenarios. Moreover, the N-gram-based methods can even outperform the training-based method in some cases. For example, both SAM and Recycling achieve higher acceleration ratios than EAGLE-3 when applied to DSL-8B on the LiveCodeBench. These observations are consistent with the findings reported before. This indicates that redundancy is not

Table 7: Performance comparison of speculative decoding methods for reasoning models under the multi-round thinking framework at different temperatures $T$ on other domains ( Best , Second Best ).

(a) Experiments on LiveCodeBench.

| Model | DSL-8B ($T = 0$) | | DSL-8B ($T = 0.6$) | | QW3-8B ($T = 0$) | | QW3-8B ($T = 0.6$) | |
|---|---|---|---|---|---|---|---|---|
| Method | MAT | Speed | MAT | Speed | MAT | Speed | MAT | Speed |
| EAGLE-3 | 2.12 | 1.64× | 2.00 | 1.33× | 4.37 | 2.85× | 4.02 | 2.58× |
| SAM[EAGLE-3] | 4.17 | 3.34× | 2.30 | 1.80× | 4.46 | 3.27× | 3.70 | 2.54× |
| SAM | 2.90 | 2.79× | 1.72 | 1.44× | 2.19 | 1.92× | 1.77 | 1.58× |
| Recycling | 2.91 | 2.02× | 2.69 | 1.71× | 2.90 | 1.98× | 2.83 | 1.92× |
| PLD | 2.24 | 1.81× | – | – | 1.94 | 1.57× | – | – |
| SpS | – | – | – | – | 5.93 | 0.83× | 5.16 | 0.85× |
| REST | 1.33 | 1.11× | 1.37 | 1.02× | 1.35 | 1.14× | 1.37 | 1.09× |
| Lookahead | 2.15 | 1.57× | – | – | 2.08 | 1.53× | – | – |
| PIA | 2.40 | 1.71× | 1.63 | 1.04× | 2.04 | 1.46× | 1.72 | 1.32× |
| AR | 1.00 | 1.00× | 1.00 | 1.00× | 1.00 | 1.00× | 1.00 | 1.00× |

(b) Experiments on CNN/Daily Mail.

| Model | DSL-8B ($T = 0$) | | DSL-8B ($T = 0.6$) | | QW3-8B ($T = 0$) | | QW3-8B ($T = 0.6$) | |
|---|---|---|---|---|---|---|---|---|
| Method | MAT | Speed | MAT | Speed | MAT | Speed | MAT | Speed |
| EAGLE-3 | 4.76 | 2.83× | 4.60 | 2.65× | 4.11 | 2.63× | 4.01 | 2.53× |
| SAM[EAGLE-3] | 4.92 | 3.05× | 4.70 | 2.79× | 4.31 | 2.77× | 4.17 | 2.61× |
| SAM | 1.53 | 1.45× | 1.39 | 1.33× | 1.55 | 1.45× | 1.52 | 1.42× |
| Recycling | 2.57 | 1.81× | 2.50 | 1.71× | 2.73 | 1.95× | 2.70 | 1.86× |
| PLD | 1.47 | 1.32× | – | – | 1.46 | 1.30× | – | – |
| SpS | – | – | – | – | 3.57 | 0.51× | 3.44 | 0.54× |
| REST | 1.28 | 1.04× | 1.29 | 1.00× | 1.28 | 1.05× | 1.28 | 1.01× |
| Lookahead | 1.59 | 1.29× | – | – | 1.61 | 1.30× | – | – |
| PIA | 1.61 | 1.42× | 1.48 | 1.30× | 1.58 | 1.39× | 1.55 | 1.36× |
| AR | 1.00 | 1.00× | 1.00 | 1.00× | 1.00 | 1.00× | 1.00 | 1.00× |

Table 8: Error analysis of Qwen3-8B on the multi-round thinking ( Best , Second Best ).

(a) Experiments with Qwen3-8B ($T = 0$).

| Dataset | AIME24 | | AIME25 | | MATH500 | | GPQA | | Overall | |
|---|---|---|---|---|---|---|---|---|---|---|
| Method | MAT | Speed | MAT | Speed | MAT | Speed | MAT | Speed | MAT | Speed |
| EAGLE-3 | 4.31±0.00 | 2.77±0.03× | 4.39±0.02 | 2.87±0.01× | 4.50±0.00 | 3.01±0.03× | 4.39±0.00 | 2.89±0.02× | 4.38±0.00 | 2.88±0.02× |
| SAM[EAGLE-3] | 4.32±0.07 | 3.00±0.07× | 4.46±0.01 | 3.15±0.03× | 4.91±0.08 | 3.43±0.04× | 5.80±0.09 | 4.10±0.06× | 4.72±0.03 | 3.43±0.05× |
| Recycling | 2.99±0.03 | 2.03±0.04× | 2.96±0.02 | 2.07±0.06× | 2.99±0.01 | 2.15±0.05× | 3.06±0.01 | 2.14±0.04× | 3.00±0.01 | 2.10±0.05× |
| AR | 1.00±0.00 | 1.00±0.00× | 1.00±0.00 | 1.00±0.00× | 1.00±0.00 | 1.00±0.00× | 1.00±0.00 | 1.00±0.00× | 1.00±0.00 | 1.00±0.00× |

(b) Experiments with Qwen3-8B ($T = 0.6$).

| Dataset | AIME24 | | AIME25 | | MATH500 | | GPQA | | Overall | |
|---|---|---|---|---|---|---|---|---|---|---|
| Method | MAT | Speed | MAT | Speed | MAT | Speed | MAT | Speed | MAT | Speed |
| EAGLE-3 | 4.12±0.01 | 2.60±0.01× | 4.21±0.02 | 2.71±0.01× | 4.34±0.02 | 2.85±0.02× | 4.02±0.02 | 2.60±0.07× | 4.16±0.01 | 2.70±0.03× |
| SAM[EAGLE-3] | 3.89±0.04 | 2.61±0.06× | 3.94±0.01 | 2.70±0.03× | 4.18±0.01 | 2.90±0.03× | 3.99±0.03 | 2.72±0.09× | 3.97±0.02 | 2.74±0.05× |
| Recycling | 2.89±0.01 | 1.95±0.05× | 2.90±0.02 | 1.97±0.06× | 2.90±0.01 | 2.05±0.06× | 2.92±0.01 | 2.00±0.07× | 2.90±0.00 | 2.00±0.06× |
| AR | 1.00±0.00 | 1.00±0.00× | 1.00±0.00 | 1.00±0.00× | 1.00±0.00 | 1.00±0.00× | 1.00±0.00 | 1.00±0.00× | 1.00±0.00 | 1.00±0.00× |

specific to mathematical reasoning but a universal phenomenon in generation tasks. The consistent superiority of hybrid methods across domains further validates our benchmark's core insights.

• **Analysis of error bars**. We conduct error analysis on a representative subset of methods (EAGLE-3, SAM[EAGLE-3], and Recycling) by running them three times. The results in Table 8 demonstrate that the impact of randomness is minimal, with standard deviations consistently below 0.1 across tested configurations (Qwen3-8B, multi-round thinking).

• **Analysis of varying batch sizes**. As our work primarily serves as a benchmark, to ensure objectivity and fairness, we intentionally avoid altering the logic of the evaluated methods' original implementations. However, many existing codes are not well-suited for batched inference. For example, the official implementation of EAGLE-3 does not support batched inference; the SAM method's suffix automaton does not support batched writing or lookup; and Recycling does not support batched adjacency matrix construction. Given these architectural constraints, implementing varying batch sizes would require significant modifications to the original codebases. We maintain that our single-batch results serve as a reliable proxy for performance in moderate-batch regimes due to the memory-bound nature of LLM decoding. Speculative decoding improves hardware utilization by

Table 9: Performance comparison with varying batch sizes on DSL-8B ( Best , Second Best ).

(a) Batch Size = 4

| Dataset | AIME24 | | AIME25 | | MATH500 | | GPQA | |
|---|---|---|---|---|---|---|---|---|
| Method | MAT | Speed | MAT | Speed | MAT | Speed | MAT | Speed |
| EAGLE-3 | **3.30** | **1.52×** | **3.34** | 1.56× | 3.10 | 1.46× | **3.08** | 1.41× |
| SAM-Single | 1.81 | 1.33× | 1.87 | 1.44× | 2.91 | 2.72× | 2.01 | 1.58× |
| SAM-Cross | 1.86 | 1.34× | 2.11 | **1.62×** | 3.19 | **2.89×** | 2.21 | **1.73×** |

(b) Batch Size = 8

| Dataset | AIME24 | | AIME25 | | MATH500 | | GPQA | |
|---|---|---|---|---|---|---|---|---|
| Method | MAT | Speed | MAT | Speed | MAT | Speed | MAT | Speed |
| EAGLE-3 | **3.34** | **1.17×** | **3.37** | 1.18× | **3.13** | 1.11× | **3.06** | 1.03× |
| SAM-Single | 1.75 | 1.06× | 1.81 | 1.18× | 2.52 | 2.24× | 1.92 | **1.36×** |
| SAM-Cross | 1.82 | 1.12× | 1.92 | **1.20×** | 3.12 | **2.61×** | 1.99 | 1.34× |

verifying multiple tokens in a single forward pass, thereby reducing the frequency of memory accesses and overall latency. From a roofline model perspective (Yan et al., 2025; Yuan et al., 2024), while extremely large batch sizes may eventually shift the bottleneck to computation where verification overhead could negate gains, increasing the batch size within the memory-bound region before reaching the compute-bound consistently yields inference acceleration. Crucially, for N-gram-based methods, the cost of draft generation is negligible, making the speedup highly positively correlated with the MAT. Consequently, the high MAT observed in our single-batch experiments serves as a strong indicator of potential speedups in batched scenarios. We focused on single-batch inference to isolate these intrinsic algorithmic capabilities from the uneven system-level engineering (Meta, 2025; Sadhukhan et al., 2025) required to optimize batched execution across different baselines. Despite these challenges, we implement preliminary batched versions of EAGLE-3 and SAM for exploration under the BoN scenario. For SAM, to enable batched inference, we modify the suffix automaton's query mechanism: instead of maintaining a single shared automaton, we construct one for each reasoning path, and draft tokens across branches are cross-looked-up for matches. While this design makes batching possible, it inevitably increases the query complexity beyond $\mathcal{O}(1)$, representing a necessary compromise under the current framework. We test BoN with N = 4 and N = 8 on a subset of the four reasoning tasks using DSL-8B, and the results are shown in Table 9 (where SAM-Single is the original implementation using a single automaton per reasoning path, and SAM-Cross is the modified version that supports cross–branch token look-ups). The results suggest that increasing the batch size indeed tends to trigger compute-bound behavior. For instance, EAGLE-3 achieves an average speedup of around $1.5\times$ at batch size 4, which is notably lower than its MAT and, in some cases, even slightly behind the N-gram-based method SAM. This observation is consistent with our earlier findings, indicating that redundancy plays a crucial role in test-time scaling acceleration. In practice, when the batch size increases, the speculative size typically needs to be reduced to avoid becoming compute-bound; the observed slowdown is therefore likely due to the current testing configuration, which does not carefully calibrate this trade-off. In addition, the parallelization of BoN inherently reduces overlapping redundancy across different inference paths, which further explains the diminishing speedup. This reinforces our conclusion that redundancy should be preserved and fully utilized in speculative decoding. Our benchmark therefore substantiates a key practical insight that BoN should be applied along the sequential dimension, while the batch dimension should be reserved for handling different queries to maximize throughput (as leaving the available redundancy unexploited would be an unacceptable waste).

● **Analysis of acceleration across turns**. Tables 10–13 present detailed acceleration ratios and MAT values across different turns for various model sizes under different test-time scaling paradigms, serving as a complement to Section 4.2.6. The data reveals a particularly encouraging pattern for methods capable of caching historical generations, such as SAM and PIA, which demonstrate increasing acceleration ratios as the number of turns progresses. This scaling up and speeding up trend represents a crucial advantage for test-time scaling acceleration, as it suggests that the computational benefits compound over longer reasoning sequences. The ability to leverage previously generated content not only improves efficiency in individual turns but creates a cumulative effect where each subsequent turn becomes progressively faster, making these caching-enabled approaches especially valuable for complex multi-turn reasoning tasks that require extended computational processes.

Table 10: Performance comparison of speculative decoding methods across turns under the multi-round thinking framework with different temperature $T$ on DSL-8B and QW3-8B.

| Model | Bench / Method | Turn | AIME24 | | AIME25 | | MATH500 | | GPQA | | Overall | |
|---|---|---|---|---|---|---|---|---|---|---|---|---|
| | | | MAT | Speed | MAT | Speed | MAT | Speed | MAT | Speed | MAT | Speed |
| DSL-8B ($T=0$) | AR | Turn 1 | 1.00 | 1.00× | 1.00 | 1.00× | 1.00 | 1.00× | 1.00 | 1.00× | 1.00 | 1.00× |
| | | Turn 2 | 1.00 | 1.00× | 1.00 | 1.00× | 1.00 | 1.00× | 1.00 | 1.00× | 1.00 | 1.00× |
| | EAGLE-3 | Turn 1 | 2.39 | 1.70× | 2.30 | 1.51× | 2.60 | 2.83× | 2.41 | 1.60× | 2.40 | 1.94× |
| | | Turn 2 | 2.12 | 1.80× | 2.18 | 1.79× | 2.68 | 3.27× | 2.59 | 2.55× | 2.27 | 2.41× |
| | SAM[EAGLE-3] | Turn 1 | 3.47 | 2.71× | 4.34 | 3.50× | 4.28 | 3.71× | 6.91 | 4.80× | 4.44 | 3.71× |
| | | Turn 2 | 4.61 | 4.18× | 4.71 | 5.53× | 6.39 | 5.22× | 7.17 | 4.25× | 5.17 | 4.79× |
| | SAM | Turn 1 | 2.38 | 2.24× | 2.64 | 2.51× | 2.25 | 1.86× | 3.61 | 3.31× | 2.67 | 2.47× |
| | | Turn 2 | 3.04 | 3.21× | 3.81 | 3.89× | 3.26 | 3.60× | 3.48 | 2.75× | 3.38 | 3.35× |
| | Recycling | Turn 1 | 2.95 | 2.11× | 2.96 | 2.01× | 2.96 | 2.15× | 3.04 | 2.09× | 2.97 | 2.09× |
| | | Turn 2 | 3.03 | 2.07× | 3.02 | 2.15× | 2.99 | 2.24× | 2.99 | 2.11× | 3.01 | 2.15× |
| | PLD | Turn 1 | 2.10 | 1.76× | 2.23 | 1.75× | 2.18 | 1.57× | 2.60 | 1.97× | 2.25 | 1.76× |
| | | Turn 2 | 2.19 | 1.80× | 2.58 | 2.06× | 2.38 | 2.11× | 2.75 | 1.92× | 2.44 | 1.98× |
| | REST | Turn 1 | 1.32 | 1.01× | 1.35 | 0.96× | 1.38 | 1.08× | 1.31 | 1.01× | 1.34 | 1.02× |
| | | Turn 2 | 1.34 | 1.02× | 1.34 | 0.96× | 1.35 | 1.09× | 1.31 | 1.02× | 1.34 | 1.02× |
| | Lookahead | Turn 1 | 2.19 | 1.60× | 2.24 | 1.52× | 2.26 | 1.53× | 2.47 | 1.64× | 2.28 | 1.57× |
| | | Turn 2 | 2.24 | 1.58× | 2.29 | 1.51× | 2.30 | 1.59× | 2.38 | 1.55× | 2.29 | 1.56× |
| | PIA | Turn 1 | 1.62 | 1.35× | 1.70 | 1.41× | 1.68 | 1.28× | 2.01 | 1.51× | 1.74 | 1.38× |
| | | Turn 2 | 2.16 | 1.84× | 2.38 | 2.15× | 2.47 | 2.33× | 2.44 | 1.81× | 2.32 | 2.04× |
| DSL-8B ($T=0.6$) | AR | Turn 1 | 1.00 | 1.00× | 1.00 | 1.00× | 1.00 | 1.00× | 1.00 | 1.00× | 1.00 | 1.00× |
| | | Turn 2 | 1.00 | 1.00× | 1.00 | 1.00× | 1.00 | 1.00× | 1.00 | 1.00× | 1.00 | 1.00× |
| | EAGLE-3 | Turn 1 | 2.26 | 1.40× | 2.22 | 1.45× | 3.42 | 3.20× | 2.39 | 1.63× | 2.36 | 1.98× |
| | | Turn 2 | 2.16 | 1.47× | 2.12 | 1.70× | 3.03 | 3.15× | 2.56 | 1.97× | 2.30 | 2.12× |
| | SAM[EAGLE-3] | Turn 1 | 2.51 | 1.76× | 2.42 | 1.91× | 3.35 | 3.03× | 2.72 | 2.08× | 2.60 | 2.23× |
| | | Turn 2 | 2.58 | 2.08× | 2.55 | 2.16× | 3.21 | 4.06× | 3.12 | 2.52× | 2.72 | 2.76× |
| | SAM | Turn 1 | 1.73 | 1.52× | 1.73 | 1.53× | 1.74 | 1.50× | 1.85 | 1.68× | 1.76 | 1.56× |
| | | Turn 2 | 2.02 | 1.99× | 1.98 | 1.89× | 2.13 | 2.38× | 2.08 | 1.81× | 2.03 | 2.02× |
| | Recycling | Turn 1 | 2.82 | 1.88× | 2.80 | 1.91× | 2.84 | 2.02× | 2.78 | 1.97× | 2.81 | 1.95× |
| | | Turn 2 | 2.86 | 2.00× | 2.87 | 1.96× | 2.91 | 2.09× | 2.86 | 1.98× | 2.87 | 2.01× |
| | REST | Turn 1 | 1.35 | 0.99× | 1.37 | 0.99× | 1.40 | 1.03× | 1.36 | 0.99× | 1.37 | 1.00× |
| | | Turn 2 | 1.35 | 0.99× | 1.37 | 0.99× | 1.40 | 1.03× | 1.36 | 0.99× | 1.37 | 1.00× |
| | PIA | Turn 1 | 1.47 | 1.18× | 1.47 | 1.20× | 1.48 | 1.23× | 1.57 | 1.29× | 1.50 | 1.23× |
| | | Turn 2 | 1.73 | 1.41× | 1.71 | 1.44× | 1.82 | 1.79× | 1.90 | 1.52× | 1.76 | 1.55× |
| QW3-8B ($T=0$) | AR | Turn 1 | 1.00 | 1.00× | 1.00 | 1.00× | 1.00 | 1.00× | 1.00 | 1.00× | 1.00 | 1.00× |
| | | Turn 2 | 1.00 | 1.00× | 1.00 | 1.00× | 1.00 | 1.00× | 1.00 | 1.00× | 1.00 | 1.00× |
| | EAGLE-3 | Turn 1 | 4.23 | 2.73× | 4.31 | 2.84× | 4.44 | 2.99× | 4.27 | 2.84× | 4.30 | 2.86× |
| | | Turn 2 | 4.45 | 3.01× | 4.47 | 3.00× | 4.62 | 3.10× | 4.63 | 2.95× | 4.52 | 3.02× |
| | SAM[EAGLE-3] | Turn 1 | 4.05 | 2.79× | 4.13 | 2.96× | 4.60 | 3.23× | 5.15 | 3.84× | 4.38 | 3.22× |
| | | Turn 2 | 4.98 | 3.82× | 4.98 | 3.77× | 5.36 | 4.25× | 7.82 | 5.19× | 5.44 | 4.29× |
| | SAM | Turn 1 | 1.94 | 1.77× | 2.03 | 2.08× | 1.90 | 1.70× | 2.91 | 3.01× | 2.13 | 2.13× |
| | | Turn 2 | 2.58 | 2.54× | 2.71 | 2.70× | 2.92 | 2.75× | 3.88 | 3.27× | 2.87 | 2.83× |
| | Recycling | Turn 1 | 2.97 | 2.10× | 2.94 | 2.14× | 3.00 | 2.18× | 3.06 | 2.15× | 2.99 | 2.15× |
| | | Turn 2 | 3.08 | 2.11× | 3.03 | 2.14× | 3.00 | 2.20× | 3.09 | 2.17× | 3.05 | 2.16× |
| | PLD | Turn 1 | 1.84 | 1.52× | 1.84 | 1.54× | 1.84 | 1.54× | 2.41 | 2.00× | 1.95 | 1.65× |
| | | Turn 2 | 2.14 | 1.82× | 2.18 | 1.93× | 2.07 | 1.83× | 2.65 | 2.07× | 2.24 | 1.91× |
| | SpS | Turn 1 | 6.87 | 0.88× | 7.09 | 0.94× | 6.52 | 0.77× | 6.10 | 0.82× | 6.68 | 0.85× |
| | | Turn 2 | 8.40 | 1.00× | 8.14 | 1.00× | 7.01 | 0.78× | 6.90 | 0.79× | 7.80 | 0.89× |
| | REST | Turn 1 | 1.40 | 1.14× | 1.39 | 1.14× | 1.42 | 1.17× | 1.34 | 1.09× | 1.38 | 1.14× |
| | | Turn 2 | 1.38 | 1.10× | 1.39 | 1.14× | 1.41 | 1.14× | 1.32 | 1.07× | 1.38 | 1.11× |
| | Lookahead | Turn 1 | 2.13 | 1.52× | 2.11 | 1.54× | 2.13 | 1.58× | 2.29 | 1.70× | 2.16 | 1.59× |
| | | Turn 2 | 2.16 | 1.57× | 2.12 | 1.52× | 2.05 | 1.53× | 2.25 | 1.60× | 2.15 | 1.55× |
| | PIA | Turn 1 | 1.64 | 1.35× | 1.71 | 1.49× | 1.66 | 1.41× | 1.87 | 1.58× | 1.72 | 1.45× |
| | | Turn 2 | 2.18 | 1.84× | 2.43 | 2.23× | 2.40 | 2.15× | 2.72 | 2.17× | 2.38 | 2.10× |
| QW3-8B ($T=0.6$) | AR | Turn 1 | 1.00 | 1.00× | 1.00 | 1.00× | 1.00 | 1.00× | 1.00 | 1.00× | 1.00 | 1.00× |
| | | Turn 2 | 1.00 | 1.00× | 1.00 | 1.00× | 1.00 | 1.00× | 1.00 | 1.00× | 1.00 | 1.00× |
| | EAGLE-3 | Turn 1 | 4.08 | 2.60× | 4.13 | 2.63× | 4.30 | 2.85× | 3.99 | 2.63× | 4.11 | 2.69× |
| | | Turn 2 | 4.16 | 2.66× | 4.35 | 2.88× | 4.36 | 2.94× | 4.14 | 2.78× | 4.25 | 2.82× |
| | SAM[EAGLE-3] | Turn 1 | 3.84 | 2.59× | 3.86 | 2.61× | 4.03 | 2.75× | 3.90 | 2.71× | 3.89 | 2.67× |
| | | Turn 2 | 4.06 | 2.95× | 4.04 | 2.98× | 4.54 | 3.33× | 4.14 | 3.05× | 4.12 | 3.09× |
| | SAM | Turn 1 | 1.82 | 1.63× | 1.83 | 1.68× | 1.79 | 1.58× | 1.96 | 1.77× | 1.85 | 1.66× |
| | | Turn 2 | 2.08 | 2.01× | 2.14 | 2.01× | 2.28 | 2.20× | 2.38 | 2.09× | 2.18 | 2.08× |
| | Recycling | Turn 1 | 2.87 | 2.02× | 2.91 | 2.04× | 2.89 | 2.11× | 2.90 | 2.06× | 2.89 | 2.06× |
| | | Turn 2 | 2.91 | 1.96× | 2.92 | 2.04× | 2.90 | 2.11× | 2.96 | 2.12× | 2.92 | 2.06× |
| | SpS | Turn 1 | 6.06 | 0.90× | 6.41 | 0.94× | 5.97 | 0.79× | 5.74 | 0.84× | 6.07 | 0.86× |
| | | Turn 2 | 6.47 | 0.93× | 6.73 | 0.98× | 6.53 | 0.83× | 7.65 | 0.84× | 6.79 | 0.89× |
| | REST | Turn 1 | 1.41 | 1.09× | 1.40 | 1.10× | 1.42 | 1.12× | 1.37 | 1.04× | 1.40 | 1.09× |
| | | Turn 2 | 1.40 | 1.06× | 1.41 | 1.09× | 1.41 | 1.12× | 1.35 | 1.00× | 1.39 | 1.07× |
| | PIA | Turn 1 | 1.56 | 1.32× | 1.58 | 1.36× | 1.58 | 1.32× | 1.63 | 1.40× | 1.59 | 1.35× |
| | | Turn 2 | 1.91 | 1.59× | 1.89 | 1.62× | 2.02 | 1.79× | 2.02 | 1.70× | 1.94 | 1.68× |

Table 11: Performance comparison of speculative decoding methods across turns under BoN framework with different temperature $T$ on DSL-8B and QW3-8B.

| Model | Method | Turn | AIME24 | | AIME25 | | MATH500 | | GPQA | | Overall | |
|---|---|---|---|---|---|---|---|---|---|---|---|---|
| | | | MAT | Speed | MAT | Speed | MAT | Speed | MAT | Speed | MAT | Speed |
| DSL-8B $(T=0.6)$ | AR | Turn 1 | 1.00 | 1.00× | 1.00 | 1.00× | 1.00 | 1.00× | 1.00 | 1.00× | 1.00 | 1.00× |
| | | Turn 2 | 1.00 | 1.00× | 1.00 | 1.00× | 1.00 | 1.00× | 1.00 | 1.00× | 1.00 | 1.00× |
| | | Turn 3 | 1.00 | 1.00× | 1.00 | 1.00× | 1.00 | 1.00× | 1.00 | 1.00× | 1.00 | 1.00× |
| | | Turn 4 | 1.00 | 1.00× | 1.00 | 1.00× | 1.00 | 1.00× | 1.00 | 1.00× | 1.00 | 1.00× |
| | EAGLE-3 | Turn 1 | 2.23 | 1.49× | 2.22 | 1.52× | 3.49 | 3.19× | 2.39 | 1.88× | 2.36 | 2.07× |
| | | Turn 2 | 2.22 | 1.36× | 2.18 | 1.57× | 3.31 | 3.14× | 2.42 | 1.68× | 2.34 | 1.97× |
| | | Turn 3 | 2.21 | 1.24× | 2.18 | 1.58× | 2.85 | 2.89× | 2.51 | 1.78× | 2.33 | 1.92× |
| | | Turn 4 | 2.12 | 1.11× | 2.12 | 1.58× | 2.88 | 2.95× | 2.20 | 1.60× | 2.21 | 1.86× |
| | SAM[EAGLE-3] | Turn 1 | 2.41 | 1.68× | 2.43 | 1.83× | 3.63 | 3.26× | 2.72 | 2.15× | 2.57 | 2.28× |
| | | Turn 2 | 2.45 | 1.72× | 2.43 | 1.83× | 3.15 | 3.15× | 2.74 | 2.19× | 2.58 | 2.26× |
| | | Turn 3 | 2.41 | 1.75× | 2.41 | 1.75× | 3.29 | 3.37× | 2.70 | 2.07× | 2.54 | 2.30× |
| | | Turn 4 | 2.49 | 1.79× | 2.49 | 1.79× | 3.24 | 3.33× | 2.73 | 2.11× | 2.57 | 2.34× |
| | SAM | Turn 1 | 1.79 | 1.54× | 1.75 | 1.52× | 1.77 | 1.47× | 1.87 | 1.65× | 1.79 | 1.54× |
| | | Turn 2 | 1.91 | 1.64× | 1.85 | 1.61× | 1.96 | 1.80× | 2.01 | 1.74× | 1.91 | 1.70× |
| | | Turn 3 | 1.92 | 1.69× | 1.90 | 1.69× | 2.10 | 2.02× | 2.05 | 1.78× | 1.96 | 1.81× |
| | | Turn 4 | 1.97 | 1.73× | 1.95 | 1.70× | 2.12 | 2.14× | 2.10 | 1.79× | 2.01 | 1.85× |
| | Recycling | Turn 1 | 2.81 | 1.93× | 2.83 | 1.90× | 2.85 | 2.03× | 2.78 | 2.01× | 2.82 | 1.97× |
| | | Turn 2 | 2.82 | 1.92× | 2.79 | 1.86× | 2.89 | 2.06× | 2.82 | 1.93× | 2.82 | 1.95× |
| | | Turn 3 | 2.81 | 1.97× | 2.80 | 1.89× | 2.89 | 2.02× | 2.80 | 1.87× | 2.81 | 1.94× |
| | | Turn 4 | 2.81 | 1.91× | 2.80 | 1.86× | 2.89 | 2.09× | 2.81 | 1.91× | 2.81 | 1.95× |
| | REST | Turn 1 | 1.35 | 1.01× | 1.39 | 1.02× | 1.40 | 1.09× | 1.33 | 1.02× | 1.37 | 1.04× |
| | | Turn 2 | 1.36 | 0.99× | 1.37 | 0.97× | 1.39 | 1.08× | 1.34 | 0.99× | 1.37 | 1.01× |
| | | Turn 3 | 1.38 | 1.01× | 1.38 | 0.99× | 1.41 | 1.09× | 1.32 | 0.99× | 1.37 | 1.02× |
| | | Turn 4 | 1.36 | 1.02× | 1.37 | 1.01× | 1.40 | 1.03× | 1.33 | 0.99× | 1.36 | 1.01× |
| | PIA | Turn 1 | 1.45 | 1.15× | 1.45 | 1.20× | 1.49 | 1.24× | 1.58 | 1.35× | 1.48 | 1.24× |
| | | Turn 2 | 1.56 | 1.17× | 1.55 | 1.20× | 1.58 | 1.36× | 1.66 | 1.37× | 1.58 | 1.28× |
| | | Turn 3 | 1.56 | 1.29× | 1.58 | 1.21× | 1.71 | 1.47× | 1.73 | 1.35× | 1.62 | 1.34× |
| | | Turn 4 | 1.56 | 1.17× | 1.61 | 1.27× | 1.72 | 1.49× | 1.76 | 1.39× | 1.63 | 1.34× |
| QW3-8B $(T=0.6)$ | AR | Turn 1 | 1.00 | 1.00× | 1.00 | 1.00× | 1.00 | 1.00× | 1.00 | 1.00× | 1.00 | 1.00× |
| | | Turn 2 | 1.00 | 1.00× | 1.00 | 1.00× | 1.00 | 1.00× | 1.00 | 1.00× | 1.00 | 1.00× |
| | | Turn 3 | 1.00 | 1.00× | 1.00 | 1.00× | 1.00 | 1.00× | 1.00 | 1.00× | 1.00 | 1.00× |
| | | Turn 4 | 1.00 | 1.00× | 1.00 | 1.00× | 1.00 | 1.00× | 1.00 | 1.00× | 1.00 | 1.00× |
| | EAGLE-3 | Turn 1 | 4.04 | 2.58× | 4.10 | 2.61× | 4.32 | 2.87× | 3.99 | 2.62× | 4.09 | 2.68× |
| | | Turn 2 | 4.11 | 2.65× | 4.15 | 2.66× | 4.27 | 2.84× | 4.02 | 2.62× | 4.12 | 2.70× |
| | | Turn 3 | 4.10 | 2.67× | 4.20 | 2.65× | 4.25 | 2.84× | 3.98 | 2.65× | 4.13 | 2.71× |
| | | Turn 4 | 4.12 | 2.65× | 4.22 | 2.67× | 4.34 | 2.84× | 4.02 | 2.67× | 4.16 | 2.72× |
| | SAM[EAGLE-3] | Turn 1 | 3.81 | 2.56× | 3.90 | 2.57× | 4.06 | 2.77× | 3.93 | 2.66× | 3.90 | 2.65× |
| | | Turn 2 | 3.81 | 2.68× | 3.91 | 2.75× | 4.16 | 2.93× | 3.88 | 2.72× | 3.91 | 2.78× |
| | | Turn 3 | 3.79 | 2.66× | 3.91 | 2.76× | 4.24 | 3.18× | 3.83 | 2.67× | 3.90 | 2.83× |
| | | Turn 4 | 3.84 | 2.79× | 3.91 | 2.83× | 4.28 | 3.21× | 3.85 | 2.70× | 3.93 | 2.90× |
| | SAM | Turn 1 | 1.81 | 1.69× | 1.82 | 1.66× | 1.77 | 1.59× | 1.93 | 1.79× | 1.84 | 1.68× |
| | | Turn 2 | 2.02 | 1.85× | 2.02 | 1.88× | 2.18 | 2.00× | 2.21 | 2.00× | 2.08 | 1.94× |
| | | Turn 3 | 2.15 | 1.95× | 2.14 | 2.00× | 2.39 | 2.31× | 2.31 | 2.07× | 2.21 | 2.10× |
| | | Turn 4 | 2.27 | 2.09× | 2.23 | 2.10× | 2.51 | 2.44× | 2.42 | 2.20× | 2.32 | 2.22× |
| | Recycling | Turn 1 | 2.91 | 2.02× | 2.88 | 2.01× | 2.87 | 2.08× | 2.90 | 2.02× | 2.90 | 2.04× |
| | | Turn 2 | 2.90 | 2.04× | 2.87 | 2.01× | 2.89 | 2.04× | 2.91 | 2.05× | 2.89 | 2.04× |
| | | Turn 3 | 2.89 | 2.04× | 2.88 | 1.98× | 2.91 | 2.06× | 2.91 | 2.06× | 2.89 | 2.04× |
| | | Turn 4 | 2.89 | 2.05× | 2.87 | 1.98× | 2.92 | 2.07× | 2.90 | 2.03× | 2.89 | 2.03× |
| | SpS | Turn 1 | 6.06 | 0.91× | 6.41 | 0.95× | 5.97 | 0.80× | 5.74 | 0.85× | 6.07 | 0.88× |
| | | Turn 2 | 6.26 | 0.95× | 6.23 | 0.96× | 5.92 | 0.78× | 5.84 | 0.85× | 6.09 | 0.88× |
| | | Turn 3 | 5.85 | 0.91× | 6.15 | 0.94× | 6.08 | 0.82× | 6.01 | 0.85× | 6.02 | 0.88× |
| | | Turn 4 | 6.04 | 0.94× | 6.11 | 0.95× | 6.07 | 0.81× | 5.90 | 0.83× | 6.03 | 0.88× |
| | REST | Turn 1 | 1.41 | 1.12× | 1.40 | 1.10× | 1.42 | 1.12× | 1.37 | 1.07× | 1.40 | 1.10× |
| | | Turn 2 | 1.40 | 1.11× | 1.40 | 1.10× | 1.43 | 1.12× | 1.37 | 1.07× | 1.40 | 1.10× |
| | | Turn 3 | 1.41 | 1.11× | 1.41 | 1.11× | 1.42 | 1.12× | 1.35 | 1.09× | 1.40 | 1.11× |
| | | Turn 4 | 1.41 | 1.14× | 1.40 | 1.10× | 1.43 | 1.10× | 1.35 | 1.06× | 1.40 | 1.10× |
| | PIA | Turn 1 | 1.55 | 1.32× | 1.57 | 1.32× | 1.52 | 1.30× | 1.64 | 1.40× | 1.57 | 1.33× |
| | | Turn 2 | 1.68 | 1.40× | 1.72 | 1.46× | 1.70 | 1.43× | 1.77 | 1.47× | 1.72 | 1.44× |
| | | Turn 3 | 1.72 | 1.39× | 1.72 | 1.40× | 1.82 | 1.54× | 1.79 | 1.48× | 1.75 | 1.46× |
| | | Turn 4 | 1.73 | 1.44× | 1.81 | 1.53× | 1.84 | 1.53× | 1.85 | 1.48× | 1.80 | 1.50× |

Table 12: Performance comparison of speculative decoding methods across turns under BoN framework with different temperature $T$ on QW3-4B and QW3-14B..

| Model | Method | Turn | AIME24 | | AIME25 | | MATH500 | | GPQA | | Overall | |
|---|---|---|---|---|---|---|---|---|---|---|---|---|
| | | | MAT | Speed | MAT | Speed | MAT | Speed | MAT | Speed | MAT | Speed |
| QW3-4B ($T = 0.6$) | AR | Turn 1 | 1.00 | 1.00× | 1.00 | 1.00× | 1.00 | 1.00× | 1.00 | 1.00× | 1.00 | 1.00× |
| | | Turn 2 | 1.00 | 1.00× | 1.00 | 1.00× | 1.00 | 1.00× | 1.00 | 1.00× | 1.00 | 1.00× |
| | | Turn 3 | 1.00 | 1.00× | 1.00 | 1.00× | 1.00 | 1.00× | 1.00 | 1.00× | 1.00 | 1.00× |
| | | Turn 4 | 1.00 | 1.00× | 1.00 | 1.00× | 1.00 | 1.00× | 1.00 | 1.00× | 1.00 | 1.00× |
| | EAGLE-3 | Turn 1 | 3.44 | 2.24× | 3.45 | 2.21× | 3.49 | 2.30× | 3.29 | 2.10× | 3.42 | 2.21× |
| | | Turn 2 | 3.43 | 2.20× | 3.50 | 2.26× | 3.52 | 2.30× | 3.31 | 2.11× | 3.44 | 2.22× |
| | | Turn 3 | 3.48 | 2.23× | 3.46 | 2.24× | 3.51 | 2.30× | 3.36 | 2.11× | 3.45 | 2.22× |
| | | Turn 4 | 3.45 | 2.27× | 3.50 | 2.22× | 3.57 | 2.32× | 3.35 | 2.14× | 3.46 | 2.24× |
| | SAM[EAGLE-3] | Turn 1 | 3.38 | 2.38× | 3.27 | 2.18× | 3.37 | 2.31× | 3.33 | 2.24× | 3.33 | 2.28× |
| | | Turn 2 | 3.33 | 2.29× | 3.34 | 2.31× | 3.55 | 2.58× | 3.38 | 2.36× | 3.37 | 2.39× |
| | | Turn 3 | 3.39 | 2.36× | 3.43 | 2.44× | 3.64 | 2.79× | 3.42 | 2.38× | 3.44 | 2.50× |
| | | Turn 4 | 3.39 | 2.46× | 3.44 | 2.43× | 3.79 | 2.86× | 3.41 | 2.52× | 3.47 | 2.58× |
| | SAM | Turn 1 | 1.85 | 1.72× | 1.87 | 1.68× | 1.81 | 1.64× | 2.04 | 1.83× | 1.89 | 1.72× |
| | | Turn 2 | 2.11 | 1.91× | 2.12 | 1.93× | 2.30 | 2.13× | 2.31 | 2.09× | 2.18 | 2.02× |
| | | Turn 3 | 2.25 | 2.03× | 2.25 | 2.04× | 2.37 | 2.28× | 2.47 | 2.14× | 2.31 | 2.13× |
| | | Turn 4 | 2.35 | 2.17× | 2.33 | 2.15× | 2.54 | 2.49× | 2.55 | 2.28× | 2.41 | 2.28× |
| | Recycling | Turn 1 | 2.92 | 2.16× | 2.92 | 2.14× | 2.93 | 2.31× | 2.94 | 2.20× | 2.92 | 2.20× |
| | | Turn 2 | 2.92 | 2.13× | 2.92 | 2.13× | 2.92 | 2.30× | 2.93 | 2.20× | 2.92 | 2.20× |
| | | Turn 3 | 2.93 | 2.09× | 2.92 | 2.14× | 2.92 | 2.31× | 3.03 | 2.19× | 2.95 | 2.19× |
| | | Turn 4 | 2.91 | 2.18× | 2.90 | 2.06× | 2.93 | 2.33× | 2.94 | 2.23× | 2.92 | 2.21× |
| | REST | Turn 1 | 1.41 | 1.14× | 1.42 | 1.10× | 1.43 | 1.16× | 1.38 | 1.04× | 1.41 | 1.11× |
| | | Turn 2 | 1.42 | 1.07× | 1.41 | 1.10× | 1.44 | 1.16× | 1.38 | 1.05× | 1.41 | 1.10× |
| | | Turn 3 | 1.42 | 1.09× | 1.41 | 1.11× | 1.44 | 1.16× | 1.38 | 1.02× | 1.41 | 1.09× |
| | | Turn 4 | 1.42 | 1.12× | 1.41 | 1.11× | 1.43 | 1.16× | 1.36 | 1.05× | 1.41 | 1.11× |
| | PIA | Turn 1 | 1.57 | 1.34× | 1.59 | 1.34× | 1.57 | 1.34× | 1.70 | 1.45× | 1.60 | 1.37× |
| | | Turn 2 | 1.71 | 1.41× | 1.75 | 1.47× | 1.75 | 1.52× | 1.84 | 1.51× | 1.76 | 1.48× |
| | | Turn 3 | 1.77 | 1.38× | 1.75 | 1.43× | 1.83 | 1.58× | 1.89 | 1.53× | 1.80 | 1.48× |
| | | Turn 4 | 1.77 | 1.51× | 1.78 | 1.43× | 1.86 | 1.61× | 1.89 | 1.57× | 1.81 | 1.54× |
| QW3-14B ($T = 0.6$) | AR | Turn 1 | 1.00 | 1.00× | 1.00 | 1.00× | 1.00 | 1.00× | 1.00 | 1.00× | 1.00 | 1.00× |
| | | Turn 2 | 1.00 | 1.00× | 1.00 | 1.00× | 1.00 | 1.00× | 1.00 | 1.00× | 1.00 | 1.00× |
| | | Turn 3 | 1.00 | 1.00× | 1.00 | 1.00× | 1.00 | 1.00× | 1.00 | 1.00× | 1.00 | 1.00× |
| | | Turn 4 | 1.00 | 1.00× | 1.00 | 1.00× | 1.00 | 1.00× | 1.00 | 1.00× | 1.00 | 1.00× |
| | EAGLE-3 | Turn 1 | 3.30 | 2.28× | 3.29 | 2.28× | 3.32 | 2.25× | 3.01 | 2.07× | 3.23 | 2.22× |
| | | Turn 2 | 3.28 | 2.28× | 3.29 | 2.27× | 3.34 | 2.26× | 3.05 | 2.07× | 3.24 | 2.22× |
| | | Turn 3 | 3.27 | 2.27× | 3.33 | 2.26× | 3.33 | 2.29× | 3.05 | 2.08× | 3.25 | 2.22× |
| | | Turn 4 | 3.28 | 2.26× | 3.31 | 2.27× | 3.33 | 2.29× | 3.04 | 2.05× | 3.24 | 2.22× |
| | SAM[EAGLE-3] | Turn 1 | 3.15 | 2.24× | 3.17 | 2.24× | 3.16 | 2.19× | 3.05 | 2.13× | 3.14 | 2.20× |
| | | Turn 2 | 3.19 | 2.30× | 3.19 | 2.31× | 3.32 | 2.46× | 3.10 | 2.21× | 3.19 | 2.32× |
| | | Turn 3 | 3.21 | 2.37× | 3.25 | 2.38× | 3.42 | 2.63× | 3.09 | 2.16× | 3.22 | 2.39× |
| | | Turn 4 | 3.27 | 2.42× | 3.28 | 2.44× | 3.51 | 2.76× | 3.18 | 2.29× | 3.29 | 2.48× |
| | SAM | Turn 1 | 1.75 | 1.62× | 1.75 | 1.63× | 1.72 | 1.52× | 1.81 | 1.66× | 1.76 | 1.61× |
| | | Turn 2 | 1.96 | 1.85× | 2.00 | 1.87× | 2.08 | 1.96× | 2.05 | 1.85× | 2.01 | 1.88× |
| | | Turn 3 | 2.11 | 1.98× | 2.11 | 1.99× | 2.24 | 2.15× | 2.17 | 1.92× | 2.14 | 2.01× |
| | | Turn 4 | 2.14 | 2.01× | 2.17 | 2.06× | 2.49 | 2.39× | 2.25 | 2.02× | 2.22 | 2.13× |
| | Recycling | Turn 1 | 2.85 | 2.12× | 2.85 | 2.12× | 2.88 | 2.18× | 2.84 | 2.13× | 2.85 | 2.14× |
| | | Turn 2 | 2.86 | 2.12× | 2.85 | 2.11× | 2.90 | 2.23× | 2.87 | 2.15× | 2.87 | 2.16× |
| | | Turn 3 | 2.86 | 2.13× | 2.87 | 2.08× | 2.88 | 2.19× | 2.89 | 2.16× | 2.87 | 2.14× |
| | | Turn 4 | 2.87 | 2.13× | 2.85 | 2.10× | 2.91 | 2.23× | 2.88 | 2.16× | 2.87 | 2.16× |
| | SpS | Turn 1 | 5.78 | 1.23× | 5.79 | 1.25× | 5.54 | 1.09× | 4.47 | 1.02× | 5.42 | 1.14× |
| | | Turn 2 | 5.76 | 1.21× | 5.84 | 1.24× | 5.64 | 1.09× | 5.02 | 1.04× | 5.60 | 1.14× |
| | | Turn 3 | 5.72 | 1.22× | 5.69 | 1.23× | 5.58 | 1.09× | 5.11 | 1.03× | 5.54 | 1.14× |
| | | Turn 4 | 5.79 | 1.23× | 5.72 | 1.25× | 5.72 | 1.10× | 4.90 | 1.04× | 5.55 | 1.15× |
| | REST | Turn 1 | 1.40 | 1.18× | 1.40 | 1.16× | 1.45 | 1.18× | 1.33 | 1.11× | 1.39 | 1.16× |
| | | Turn 2 | 1.38 | 1.14× | 1.40 | 1.16× | 1.41 | 1.17× | 1.35 | 1.11× | 1.39 | 1.15× |
| | | Turn 3 | 1.40 | 1.14× | 1.40 | 1.16× | 1.42 | 1.19× | 1.33 | 1.10× | 1.39 | 1.15× |
| | | Turn 4 | 1.40 | 1.17× | 1.40 | 1.18× | 1.41 | 1.18× | 1.33 | 1.11× | 1.39 | 1.16× |
| | PIA | Turn 1 | 1.51 | 1.34× | 1.54 | 1.34× | 1.53 | 1.31× | 1.59 | 1.37× | 1.54 | 1.34× |
| | | Turn 2 | 1.64 | 1.43× | 1.70 | 1.45× | 1.72 | 1.49× | 1.73 | 1.44× | 1.69 | 1.45× |
| | | Turn 3 | 1.70 | 1.46× | 1.72 | 1.45× | 1.76 | 1.53× | 1.81 | 1.50× | 1.74 | 1.49× |
| | | Turn 4 | 1.73 | 1.48× | 1.81 | 1.59× | 1.82 | 1.59× | 1.81 | 1.50× | 1.78 | 1.54× |

Table 13: Performance comparison of speculative decoding methods across turns under multi-round thinking framework with different temperature $T$ on QW3-4B and QW3-14B.

| Model | Bench / Method | Turn | AIME24 MAT | AIME24 Speed | AIME25 MAT | AIME25 Speed | MATH500 MAT | MATH500 Speed | GPQA MAT | GPQA Speed | Overall MAT | Overall Speed |
|---|---|---|---|---|---|---|---|---|---|---|---|---|
| QW3-4B ($T = 0.6$) | AR | Turn 1 | 1.00 | 1.00× | 1.00 | 1.00× | 1.00 | 1.00× | 1.00 | 1.00× | 1.00 | 1.00× |
| | | Turn 2 | 1.00 | 1.00× | 1.00 | 1.00× | 1.00 | 1.00× | 1.00 | 1.00× | 1.00 | 1.00× |
| | EAGLE-3 | Turn 1 | 3.39 | 2.21× | 3.42 | 2.20× | 3.48 | 2.32× | 3.27 | 2.15× | 3.38 | 2.22× |
| | | Turn 2 | 3.57 | 2.44× | 3.59 | 2.38× | 3.64 | 2.50× | 3.41 | 2.39× | 3.56 | 2.43× |
| | SAM[EAGLE-3] | Turn 1 | 3.30 | 2.25× | 3.32 | 2.20× | 3.33 | 2.36× | 3.40 | 2.30× | 3.33 | 2.28× |
| | | Turn 2 | 3.58 | 2.74× | 3.54 | 2.60× | 3.90 | 3.12× | 3.66 | 2.75× | 3.62 | 2.81× |
| | SAM | Turn 1 | 1.88 | 1.79× | 1.85 | 1.72× | 1.83 | 1.67× | 2.10 | 1.92× | 1.91 | 1.77× |
| | | Turn 2 | 2.25 | 2.35× | 2.13 | 2.17× | 2.26 | 2.31× | 2.42 | 2.33× | 2.24 | 2.29× |
| | Recycling | Turn 1 | 2.92 | 2.17× | 2.89 | 2.10× | 2.91 | 2.35× | 2.93 | 2.21× | 2.91 | 2.21× |
| | | Turn 2 | 2.92 | 2.29× | 2.93 | 2.18× | 2.93 | 2.46× | 2.96 | 2.44× | 2.93 | 2.35× |
| | REST | Turn 1 | 1.41 | 1.13× | 1.42 | 1.10× | 1.45 | 1.15× | 1.37 | 1.04× | 1.41 | 1.11× |
| | | Turn 2 | 1.41 | 1.11× | 1.41 | 1.09× | 1.41 | 1.15× | 1.32 | 1.07× | 1.39 | 1.11× |
| | PIA | Turn 1 | 1.59 | 1.35× | 1.60 | 1.31× | 1.55 | 1.29× | 1.67 | 1.38× | 1.60 | 1.33× |
| | | Turn 2 | 1.98 | 1.77× | 1.93 | 1.65× | 2.09 | 1.86× | 2.17 | 1.83× | 2.00 | 1.78× |
| QW3-14B ($T = 0.6$) | AR | Turn 1 | 1.00 | 1.00× | 1.00 | 1.00× | 1.00 | 1.00× | 1.00 | 1.00× | 1.00 | 1.00× |
| | | Turn 2 | 1.00 | 1.00× | 1.00 | 1.00× | 1.00 | 1.00× | 1.00 | 1.00× | 1.00 | 1.00× |
| | EAGLE-3 | Turn 1 | 3.24 | 2.23× | 3.34 | 2.25× | 3.29 | 2.21× | 3.01 | 2.06× | 3.23 | 2.19× |
| | | Turn 2 | 3.34 | 2.34× | 3.45 | 2.47× | 3.51 | 2.40× | 3.16 | 2.18× | 3.37 | 2.34× |
| | SAM[EAGLE-3] | Turn 1 | 3.11 | 2.16× | 3.19 | 2.21× | 3.20 | 2.17× | 3.03 | 2.07× | 3.13 | 2.15× |
| | | Turn 2 | 3.29 | 2.45× | 3.37 | 2.48× | 3.54 | 2.62× | 3.40 | 2.50× | 3.36 | 2.52× |
| | SAM | Turn 1 | 1.74 | 1.62× | 1.77 | 1.64× | 1.67 | 1.49× | 1.80 | 1.65× | 1.75 | 1.59× |
| | | Turn 2 | 2.02 | 1.95× | 1.99 | 1.94× | 2.28 | 2.21× | 2.14 | 1.98× | 2.06 | 2.02× |
| | Recycling | Turn 1 | 2.86 | 2.10× | 2.85 | 2.08× | 2.86 | 2.15× | 2.84 | 2.11× | 2.85 | 2.11× |
| | | Turn 2 | 2.86 | 2.11× | 2.87 | 2.12× | 2.94 | 2.24× | 2.87 | 2.20× | 2.88 | 2.17× |
| | SpS | Turn 1 | 5.78 | 1.21× | 5.79 | 1.23× | 5.54 | 1.09× | 4.47 | 1.01× | 5.42 | 1.13× |
| | | Turn 2 | 5.78 | 1.25× | 6.21 | 1.29× | 6.04 | 1.13× | 4.96 | 1.05× | 5.87 | 1.17× |
| | REST | Turn 1 | 1.41 | 1.17× | 1.40 | 1.17× | 1.43 | 1.17× | 1.34 | 1.12× | 1.39 | 1.16× |
| | | Turn 2 | 1.41 | 1.17× | 1.41 | 1.17× | 1.41 | 1.17× | 1.31 | 1.09× | 1.39 | 1.15× |
| | PIA | Turn 1 | 1.51 | 1.34× | 1.56 | 1.37× | 1.52 | 1.31× | 1.57 | 1.36× | 1.54 | 1.34× |
| | | Turn 2 | 1.84 | 1.68× | 1.90 | 1.67× | 2.02 | 1.85× | 2.07 | 1.83× | 1.92 | 1.76× |

• **Visualization of inference acceleration process**. To demonstrate the effectiveness of different speculative decoding approaches in test-time scaling scenarios, we present the responses of DSL-8B with various speculative decoding methods (EAGLE-3, SAM, SAM[EAGLE-3]) under multi-turn thinking framework. In our visualization, red underlined text indicates tokens successfully predicted by the draft model, while blue underlined text represents tokens generated directly by the target model. The analysis reveals distinct characteristics and limitations of each approach. EAGLE-3's draft generation is fundamentally constrained by its maximum tree depth, which inherently limits its capacity for long-sequence modeling. While this constraint enables efficient tree-based speculation for shorter sequences, it becomes a significant bottleneck when dealing with extended reasoning chains typical in test-time scaling applications. In contrast, SAM demonstrates superior capability in generating longer draft sequences in a single speculation round, significantly exceeding EAGLE-3's maximum generation length. However, this extended generation capacity comes at the cost of reduced MAT, resulting in substantially more actual decoding steps compared to EAGLE-3. This trade-off between draft length and acceptance efficiency often diminishes the overall acceleration benefits. The hybrid approach SAM[EAGLE-3] strategically combines the strengths of both methodologies, achieving an optimal balance between long-sequence modeling capability and high acceptance rates. This hybrid method inherits SAM's extended draft generation capacity while maintaining EAGLE-3's superior acceptance efficiency, making it particularly well-suited for test-time scaling scenarios.

```
Responses of DSL-8B with EAGLE-3 under Multi-turn Thinking (Target, Draft)
▷ Turn 1:
Okay, so I need to figure out the profit for the math club from their bake sale. Hmm,
profit is usually calculated as total revenue minus total costs, right? So, I should
probably start by calculating how much money they made from selling the cookies, cup-
cakes, and brownies, and then subtract the cost of baking them.\n\nFirst, let me note
down all the given information:\n\n- They sold 54 cookies at three for $1 each.\n- They
sold 20 cupcakes at $2 each.\n- They sold 35 brownies at $1 each.\n- The total cost to
bake all these items was $15.\n\nAlright, so I need to find the total revenue and then
subtract the total cost to get the profit.\n\nLet me break it down item by item.\n\nStart-
ing with the cookies: 54 cookies sold at three for $1. So, how much did they make from
```

cookies?\n\nWell, if three cookies cost \$1, then one cookie would cost 1/3 of a dollar, which is approximately \$0.333. But maybe it's better to calculate it in whole dollars first.\n\nSo, 54 cookies divided by 3 cookies per dollar is 18 dollars. So, they made \$18 from cookies.\n\nWait, let me verify that: 3 cookies for \$1, so 54 divided by 3 is indeed 18. So, cookies contribute \$18 to the revenue.\n\nNext, cupcakes: 20 cupcakes sold at \$2 each. So, 20 times \$2 is \$40. That seems straightforward.\n\nSo, cupcakes bring in \$40.\n\nThen, brownies: 35 brownies sold at \$1 each. So, 35 times \$1 is \$35. So, brownies add \$35 to the revenue.\n\nNow, let me add up all the revenues:\n\nCookies: \$18\n\nCupcakes: \$40\n\nBrownies: \$35\n\nTotal revenue = 18 + 40 + 35.\n\nLet me compute that: 18 + 40 is 58, and 58 + 35 is 93. So, total revenue is \$93.\n\nNow, the total cost to bake these items was \$15. So, profit is total revenue minus total cost.\n\nSo, profit = 93 - 15.\n\n93 minus 15 is 78. So, the profit is \$78.\n\nWait, let me double-check my calculations to make sure I didn't make a mistake.\n\nStarting with cookies: 54 divided by 3 is 18. That seems right.\n\nCupcakes: 20 times 2 is 40. That's correct.\n\nBrownies: 35 times 1 is 35. That's straightforward.\n\nTotal revenue: 18 + 40 is 58, plus 35 is 93. Yes, that adds up.\n\nTotal cost is given as \$15, so subtracting that from 93 gives 78. Hmm, that seems like a lot, but let me think if I did everything correctly.\n\nWait, is the cost per item or is the total cost \$15? The problem says it cost them \$15 to bake these items. So, that's the total cost, not per item. So, yes, subtracting \$15 from \$93 is correct.\n\nSo, profit is \$78.\n\nWait, but let me think again: 54 cookies at 3 for \$1 is 18, 20 cupcakes at \$2 is 40, 35 brownies at \$1 is 35. So, 18 + 40 is 58, plus 35 is 93. So, total revenue is \$93.\n\nTotal cost is \$15, so profit is 93 - 15, which is 78. So, yes, that seems correct.\n\nI don't think I made a mistake here. So, the profit is \$78.\n\n**Final Answer**\nThe math club's profit was \boxed78 dollars.\n</think>\n\nTo determine the profit for the math club from their bake sale, we need to calculate the total revenue from selling cookies, cupcakes, and brownies, and then subtract the total cost of baking these items.\n\n1. **Revenue from cookies**:\n  - 54 cookies sold at 3 for \$1.\n  - Number of sets sold: \( \frac543 = 18 \)\n  - Revenue from cookies: \( 18 \times 1 = 18 \) dollars.\n\n2. **Revenue from cupcakes**:\n  - 20 cupcakes sold at \$2 each.\n  - Revenue from cupcakes: \( 20 \times 2 = 40 \) dollars.\n\n3. **Revenue from brownies**:\n  - 35 brownies sold at \$1 each.\n  - Revenue from brownies: \( 35 \times 1 = 35 \) dollars.\n\n4. **Total revenue**:\n  - Total revenue = Revenue from cookies + Revenue from cupcakes + Revenue from brownies\n  - Total revenue = \( 18 + 40 + 35 = 93 \) dollars.\n\n5. **Total cost**:\n  - The total cost to bake these items was \$15.\n\n6. **Profit**:\n  - Profit = Total revenue - Total cost\n  - Profit = \( 93 - 15 = 78 \) dollars.\n\nThus, the profit for the math club was \(\boxed78\) dollars.

▷ **Turn 2:**

First, I need to calculate the revenue from selling cookies. They sold 54 cookies at 3 for \$1. So, the number of sets sold is 54 divided by 3, which equals 18. Multiplying 18 by \$1 gives a revenue of \$18 from cookies.\n\nNext, I'll calculate the revenue from cupcakes. They sold 20 cupcakes at \$2 each. Multiplying 20 by \$2 results in a revenue of \$40 from cupcakes.\n\nThen, I'll calculate the revenue from brownies. They sold 35 brownies at \$1 each. Multiplying 35 by \$1 gives a revenue of \$35 from brownies.\n\nAdding up all the revenues: \$18 (cookies) + \$40 (cupcakes) + \$35 (brownies) equals a total revenue of \$93.\n\nThe total cost to bake all the items was \$15. To find the profit, I'll subtract the total cost from the total revenue: \$93 - \$15 equals a profit of \$78.\n\nTherefore, the math club made a profit of \(\boxed78\) dollars.\n</think>\n\nTo determine the profit for the math club from their bake sale, we need to calculate the total revenue from selling cookies, cupcakes, and brownies, and then subtract the total cost of baking these items.\n\n1. **Revenue from cookies**:\n  - 54 cookies sold at 3 for \$1.\n  - Number of sets sold: \( \frac543 = 18 \)\n  - Revenue from cookies: \( 18 \times 1 = 18 \) dollars.\n\n2. **Revenue from cupcakes**:\n  - 20 cupcakes sold at \$2 each.\n  - Revenue from cupcakes: \( 20 \times 2 = 40 \) dollars.\n\n3. **Revenue from brownies**:\n  - 35 brownies sold at \$1 each.\n  - Revenue from brownies: \( 35 \times 1 = 35 \) dollars.\n\n4. **Total revenue**:\n  - Total revenue = Revenue from cookies + Revenue from cupcakes + Revenue from brownies\n  - Total revenue = \( 18 + 40 + 35 = 93 \) dollars.\n\n5. **Total cost**:\n  - The total cost to bake these items was \$15.\n\n6. **Profit**:\n  - Profit = Total revenue - Total cost\n  - Profit = \( 93 - 15 = 78 \) dollars.\n\nThus, the profit for the math club was \(\boxed78\) dollars.

**Responses of DSL-8B with SAM under Multi-turn Thinking (Target, Draft)**
▷ **Turn 1:**
Okay, so I need to figure out the profit for the math club from their bake sale. Hmm, profit is usually calculated as total revenue minus total costs, right? So, I should probably start by calculating how much money they made from selling the cookies, cupcakes, and brownies, and then subtract the cost of baking them.\n\nFirst, let me note down all the given information:\n\n- They sold 54 cookies at three for \$1 each.\n- They sold 20 cupcakes at \$2 each.\n- They sold 35 brownies at \$1 each.\n- The total cost to bake all these items was \$15.\n\nAlright, so I need to find the total revenue and then subtract the total cost to get the profit.\n\nLet me break it down item by item.\n\nStarting with the cookies: 54 cookies sold at three for \$1. So, how much did they make from cookies?\n\nWell, if three cookies cost \$1, then one cookie would cost 1/3 of a dollar, which is approximately \$0.333. But maybe it's better to calculate it in whole dol-

lars first.\n\nSo, 54 cookies divided by 3 cookies per dollar is 18 dollars. So, they made $18 from cookies.\n\nWait, let me verify that: 3 cookies for $1, so 54 divided by 3 is indeed 18. So, cookies contribute $18 to the revenue.\n\nNext, cupcakes: 20 cupcakes sold at $2 each. So, 20 times $2 is $40. That seems straightforward.\n\nSo, cupcakes bring in $40.\n\nThen, brownies: 35 brownies sold at $1 each. So, 35 times $1 is $35. So, brownies add $35 to the revenue.\n\nNow, let me add up all the revenues:\n\nCookies: $18\n\nCupcakes: $40\n\nBrownies: $35\n\nTotal revenue = 18 + 40 + 35.\n\nLet me compute that: 18 + 40 is 58, and 58 + 35 is 93. So, total revenue is $93.\n\nNow, the total cost to bake these items was $15. So, profit is total revenue minus total cost.\n\nSo, profit = 93 - 15.\n\n93 minus 15 is 78. So, the profit is $78.\n\nWait, let me double-check my calculations to make sure I didn't make any mistakes.\n\nStarting with cookies: 54 divided by 3 is 18. That seems right.\n\nCupcakes: 20 times 2 is 40. That's correct.\n\nBrownies: 35 times 1 is 35. That's straightforward.\n\nTotal revenue: 18 + 40 is 58, plus 35 is 93. That adds up.\n\nTotal cost is given as $15, so subtracting that from 93 gives 78. Hmm, that seems high, but let me think.\n\nWait, is the cost per item or total cost? The problem says it cost them $15 to bake these items. So, that's the total cost, not per item. So, yes, subtracting $15 from total revenue is correct.\n\nSo, profit is $78.\n\nWait, but let me think again: 54 cookies at 3 for $1 is 18 dollars. 20 cupcakes at 2 each is 40. 35 brownies at 1 each is 35. So, 18 + 40 is 58, plus 35 is 93. So, total revenue is 93. Total cost is 15. So, 93 - 15 is 78. So, yes, that seems correct.\n\nAlternatively, maybe I can calculate the cost per item and see if that makes sense.\n\nWait, but the problem says it cost them $15 to bake these items. So, that's the total cost, not per item. So, I don't need to calculate cost per item. So, that's why I can directly subtract the total cost from total revenue.\n\nSo, I think my calculation is correct.\n\nTherefore, the profit is $78.\n\n**Final Answer**\nThe math club's profit was \boxed78 dollars.\n</think>\n\nTo determine the profit for the math club from their bake sale, we need to calculate the total revenue from selling cookies, cupcakes, and brownies, and then subtract the total cost of baking these items.\n\n1. **Calculate revenue from cookies:**\n  - 54 cookies sold at 3 for $1.\n  - Revenue from cookies: \( \frac543 = 18 \) dollars.\n2. **Calculate revenue from cupcakes:**\n  - 20 cupcakes sold at $2 each.\n  - Revenue from cupcakes: \( 20 \times 2 = 40 \) dollars.\n3. **Calculate revenue from brownies:**\n  - 35 brownies sold at $1 each.\n  - Revenue from brownies: \( 35 \times 1 = 35 \) dollars.\n4. **Calculate total revenue:**\n  - Total revenue: \( 18 + 40 + 35 = 93 \) dollars.\n5. **Calculate profit:**\n  - Total cost to bake the items: $15.\n  - Profit: \( 93 - 15 = 78 \) dollars.\n\nThus, the profit for the math club was \(\boxed78\) dollars.

▷ **Turn 2:**
First, I need to calculate the revenue from selling cookies. They sold 54 cookies at 3 for $1. So, the number of sets sold is 54 divided by 3, which equals 18. Therefore, the revenue from cookies is 18 dollars.\n\nNext, I'll calculate the revenue from cupcakes. They sold 20 cupcakes at $2 each. Multiplying 20 by 2 gives a revenue of 40 dollars from cupcakes.\n\nThen, I'll determine the revenue from brownies. They sold 35 brownies at $1 each. So, the revenue from brownies is 35 dollars.\n\nAdding up all the revenues: 18 dollars (cookies) + 40 dollars (cupcakes) + 35 dollars (brownies) equals a total revenue of 93 dollars.\n\nFinally, to find the profit, I'll subtract the total baking cost of 15 dollars from the total revenue. This gives a profit of 93 minus 15, which equals 78 dollars.\n\nTherefore, the math club's profit from the bake sale is \(\boxed78\) dollars.\n</think>\n\nTo determine the profit for the math club from their bake sale, we need to calculate the total revenue from selling cookies, cupcakes, and brownies, and then subtract the total cost of baking these items.\n\n1. **Calculate revenue from cookies:**\n  - 54 cookies sold at 3 for \$1.\n  - Revenue from cookies: \( \frac543 = 18 \) dollars.\n\n2. **Calculate revenue from cupcakes:**\n  - 20 cupcakes sold at \$2 each.\n  - Revenue from cupcakes: \( 20 \times 2 = 40 \) dollars.\n\n3. **Calculate revenue from brownies:**\n  - 35 brownies sold at \$1 each.\n  - Revenue from brownies: \( 35 \times 1 = 35 \) dollars.\n\n4. **Calculate total revenue:**\n  - Total revenue: \( 18 + 40 + 35 = 93 \) dollars.\n\n5. **Calculate profit:**\n  - Total cost to bake the items: \$15.\n  - Profit: \( 93 - 15 = 78 \) dollars.\n\nThus, the profit for the math club was \(\boxed78\) dollars.

**Responses of DSL-8B with SAM[EAGLE-3] under Multi-turn Thinking (Target, Draft)**
▷ **Turn 1:**
Okay, so I need to figure out the profit for the math club from their bake sale. Hmm, profit is usually calculated as total revenue minus total costs, right? So, I should probably start by calculating how much money they made from selling the cookies, cupcakes, and brownies, and then subtract the cost of baking them.\n\nFirst, let me note down all the given information:\n\n- They sold 54 cookies at three for $1 each.\n- They sold 20 cupcakes at $2 each.\n- They sold 35 brownies at $1 each.\n- The total cost to bake all these items was $15.\n\nAlright, so I need to find the total revenue and then subtract the total cost to get the profit.\n\nLet me break it down item by item.\n\nStarting with the cookies: 54 cookies sold at three for $1. So, how much did they make from cookies?\n\nWell, if three cookies cost $1, then one cookie would cost 1/3 of a dollar, which is approximately $0.333. But maybe it's better to calculate it in whole dollars first.\n\nSo, 54 cookies divided by 3 cookies per dollar is 18 dollars. So, they made

$18 from cookies.\n\nWait, let me verify that: 3 cookies for $1, so 54 divided by 3 is indeed 18. So, cookies contribute $18 to the revenue.\n\nNext, cupcakes: 20 cupcakes sold at $2 each. So, 20 times $2 is $40. That seems straightforward.\n\nSo, cupcakes bring in $40.\n\nThen, brownies: 35 brownies sold at $1 each. So, 35 times $1 is $35. So, brownies add $35 to the revenue.\n\nNow, let me add up all the revenues:\n\nCookies: $18\n\nCupcakes: $40\n\nBrownies: $35\n\nTotal revenue = 18 + 40 + 35.\n\nLet me compute that: 18 + 40 is 58, and 58 + 35 is 93. So, total revenue is $93.\n\nNow, the total cost to bake these items was $15. So, profit is total revenue minus total cost.\n\nSo, profit = 93 − 15.\n\n93 minus 15 is 78. So, the profit is $78.\n\nWait, let me double-check my calculations to make sure I didn't make a mistake.\n\nStarting with cookies: 54 divided by 3 is 18. That seems right.\n\nCupcakes: 20 times 2 is 40. That's correct.\n\nBrownies: 35 times 1 is 35. That's straightforward.\n\nTotal revenue: 18 + 40 is 58, plus 35 is 93. Yes, that adds up.\n\nTotal cost is given as $15, so subtracting that from 93 gives 78. Hmm, that seems like a lot, but let me think if I did everything correctly.\n\nWait, is the cost per item or is the total cost $15? The problem says it cost them $15 to bake these items. So, that's the total cost, not per item. So, yes, subtracting $15 from $93 is correct.\n\nSo, profit is $78.\n\nWait, but let me think again. Is there another way to compute this? Maybe by calculating the cost per item and then subtracting from the revenue per item?\n\nLet me try that approach to cross-verify.\n\nFor cookies: 54 cookies sold at 3 for $1. So, each cookie costs 1/3 of a dollar, which is approximately $0.333. So, total cost for cookies would be 54 times (1/3) which is 18 dollars. So, same as before.\n\nCupcakes: 20 cupcakes at $2 each. So, total cost is 20 times 2, which is $40.\n\nBrownies: 35 brownies at $1 each. So, total cost is 35 times 1, which is $35.\n\nTotal cost: 18 + 40 + 35 is 93. Wait, but the problem says the total cost was $15. That doesn't make sense. Wait, hold on, that can't be.\n\nWait, hold on, maybe I misread the problem. Let me check again.\n\n"A math club is having a bake sale as a fundraiser to raise money for an upcoming trip. They sell 54 cookies at three for $1, and 20 cupcakes at $2 each, and 35 brownies at $1 each. If it cost the math club $15 to bake these items, what was their profit?"\n\nWait, so the total cost to bake these items was $15, not the cost per item. So, that means the total cost is $15, regardless of how many items they sold. So, the total cost is a fixed $15, not per item.\n\nSo, in that case, the total revenue is $93, as I calculated before, and total cost is $15, so profit is 93 − 15 = $78.\n\nWait, but if the total cost is $15, that seems really low for baking 54 cookies, 20 cupcakes, and 35 brownies. Maybe I should check if the cost is per item or total.\n\nWait, the problem says, "it cost the math club $15 to bake these items." So, that's the total cost, not per item. So, that's correct.\n\nSo, the total cost is $15, so profit is total revenue minus total cost, which is 93 − 15 = 78.\n\nSo, the profit is $78.\n\nWait, but let me think again. If the total cost is $15, but the total revenue is $93, that seems like a significant profit. Is that realistic? Well, maybe, considering they sold a lot of items.\n\nWait, 54 cookies at three for a dollar is 18 dollars, 20 cupcakes at 2 each is 40, and 35 brownies at 1 each is 35. So, 18 + 40 + 35 is 93. So, that's correct.\n\nTotal cost is 15, so profit is 93 − 15 = 78.\n\nSo, I think that's correct.\n\nAlternatively, maybe the problem is expecting me to calculate the cost per item and then subtract from the revenue per item, but that would be a different approach.\n\nWait, let me try that.\n\nIf I calculate the cost per cookie, cupcake, and brownie, and then subtract from the revenue per item, but the problem says the total cost is $15, so that might not be the right approach.\n\nWait, perhaps the problem is that the cost is per item, but the problem says "it cost the math club $15 to bake these items." So, that's the total cost, not per item.\n\nSo, in that case, the total cost is $15, so profit is total revenue minus total cost.\n\nSo, 93 − 15 is 78.\n\nTherefore, the profit is $78.\n\nWait, but let me think again. Maybe I should calculate the cost per item and then see if the total cost is 15.\n\nWait, if the total cost is $15, then the cost per item would be total cost divided by total items.\n\nTotal items: 54 cookies + 20 cupcakes + 35 brownies.\n\nSo, 54 + 20 is 74, plus 35 is 109 items in total.\n\nSo, total cost is $15, so cost per item is 15 / 109, which is approximately $0.137.\n\nBut that seems too low per item, but maybe that's the case.\n\nAlternatively, maybe the cost is per type of item.\n\nWait, the problem doesn't specify whether the $15 is the total cost or per type. It just says "it cost the math club $15 to bake these items." So, that's the total cost.\n\nSo, in that case, the total cost is $15, so profit is total revenue minus total cost, which is 93 − 15 = 78.\n\nSo, I think that's correct.\n\nTherefore, the profit is $78.\n\n**Final Answer**\nThe math club's profit was \boxed78 dollars.\n</think>\n\nTo determine the profit for the math club from their bake sale, we need to calculate the total revenue from selling cookies, cupcakes, and brownies, and then subtract the total cost of baking these items.\n\n1. **Calculate the revenue from cookies:**\n  − 54 cookies sold at 3 for $1.\n  − Revenue from cookies: \( \frac543 \times 1 = 18 \) dollars.\n\n2. **Calculate the revenue from cupcakes:**\n  − 20 cupcakes sold at $2 each.\n  − Revenue from cupcakes: \( 20 \times 2 = 40 \) dollars.\n\n3. **Calculate the revenue from brownies:**\n  − 35 brownies sold at $1 each.\n  − Revenue from brownies: \( 35 \times 1 = 35 \) dollars.\n\n4. **Calculate the total revenue:**\n  − Total revenue: \( 18 + 40 + 35 = 93 \) dollars.\n\n5. **Calculate the profit:**\n  − Total cost to bake the items is $15.\n  − Profit: \( 93 − 15 = 78 \) dollars.\n\nThus, the profit for the math club was \(\boxed78\) dollars.

▷ **Turn 2:**
First, I need to calculate the revenue from selling cookies. They sold 54 cookies at 3 for $1. So, the number of sets sold is 54 divided by 3, which equals 18. Therefore, the revenue from cookies is 18 times $1, totaling $18.\n\nNext, I'll calculate the revenue

```
from cupcakes. They sold 20 cupcakes at $2 each. So, the revenue from cupcakes is 20 mul-
tiplied by $2, which equals $40.\n\nThen, I'll calculate the revenue from brownies. They
sold 35 brownies at $1 each. So, the revenue from brownies is 35 multiplied by $1, total-
ing $35.\n\nAdding up all the revenues: $18 (cookies) + $40 (cupcakes) + $35 (brownies)
equals a total revenue of $93.\n\nFinally, to find the profit, I'll subtract the total
cost of baking the items, which is $15, from the total revenue. So, $93 minus $15 equals
a profit of $78.\n\nTherefore, the math club's profit from the bake sale is \(\boxed78\)
dollars.\n</think>\n\nTo determine the profit for the math club from their bake sale, we
need to calculate the total revenue from selling cookies, cupcakes, and brownies, and
then subtract the total cost of baking these items.\n\n1. **Calculate the revenue from
cookies:**\n  - 54 cookies sold at 3 for \$1.\n  - Revenue from cookies: \( \frac543
\times 1 = 18 \) dollars.\n\n2. **Calculate the revenue from cupcakes:**\n  - 20 cupcakes
sold at \$2 each.\n  - Revenue from cupcakes: \( 20 \times 2 = 40 \) dollars.\n\n3. **Cal-
culate the revenue from brownies:**\n  - 35 brownies sold at \$1 each.\n  - Revenue from
brownies: \( 35 \times 1 = 35 \) dollars.\n\n4. **Calculate the total revenue:**\n  - To-
tal revenue: \( 18 + 40 + 35 = 93 \) dollars.\n\n5. **Calculate the profit:**\n  - To-
tal cost to bake the items is \$15.\n  - Profit: \( 93 - 15 = 78 \) dollars.\n\nThus, the
profit for the math club was \(\boxed78\) dollars.
```

## A.4 POTENTIAL EXTENSION TO OTHER REASONING FRAMEWORKS

This work primarily investigates two mainstream test-time scaling frameworks BoN and multi-round thinking. We argue that other complex reasoning frameworks, such as ToT and MCTS-based search algorithms, could also benefit from speculative decoding. Conceptually, these frameworks can be regarded as localized, step-level test-time scaling methods. Instead of generating multiple complete responses as in BoN or multi-round thinking, these frameworks expand and evaluate several reasoning paths at each intermediate step. This localized exploration inevitably introduces substantial overlap and redundancy among candidate steps, a phenomenon closely related to the computational inefficiency examined in this benchmark. However, leveraging speculative decoding to address this redundancy in search-based frameworks presents unique technical challenges. A key distinction makes the direct application of existing speculative techniques to ToT or MCTS-based reasoning non-trivial. Unlike the relatively linear generation process in BoN or multi-round thinking, these search-based frameworks involve dynamic operations such as state evaluation, pruning, and backtracking. Existing speculative decoding methods, whether N-gram-based or advanced variants using suffix automata (e.g., SAM) or adjacency matrices (e.g., Recycling), are designed for monotonic forward generation with a single contiguous prefix. Training-based methods such as EAGLE-3 share the same limitation. Therefore, adapting them to search-based frameworks would require modifying their underlying write and lookup logic, for example by maintaining separate speculative modules (such as the individual SAM) for each search step, or extending data structures such as suffix automata and adjacency matrices to handle deletion operations, since certain steps may be pruned during the search. In summary, adapting speculative decoding to complex reasoning frameworks is a promising direction for exploiting redundancy, but it requires non-trivial framework-specific modifications that fall beyond the scope of this benchmark. We leave this adaptation for future work.

## A.5 DISCUSSION ON HYBRID STRATEGIES

The simple hybrid strategy benchmarked in this paper demonstrates remarkable effectiveness. If such a straightforward combination already yields a substantial performance gain, it strongly indicates the presence of extensive, underutilized redundancy within current architectures. We certainly agree that developing more dynamic hybrid strategies is a vital future direction. While a comprehensive exploration extends somewhat beyond the scope of this work, we suggest several promising directions, including: (i) **Context-Aware Switching**, where the strategy adapts dynamically based on the semantic context of the generation task, such as prioritizing N-gram methods for structured outputs like code while employing training-based draft models for open-ended natural language generation; (ii) **Phase-Specific Optimization**, where the strategy is tailored to different stages of the generation process, for example by utilizing N-gram speculation during a summarization phase where the text is highly likely to reiterate content from the preceding reasoning steps; and (iii) **Uncertainty-Based Toggling**, where the strategy is adjusted on-the-fly according to the model's predictive uncertainty,

specifically by leveraging N-gram speculation when low uncertainty suggests formulaic structures, while switching to training-based methods for open-ended generation.

## A.6 THE USE OF LLMS

In the preparation of this manuscript, LLMs are used solely for language polishing purposes. Specifically, the LLM assisted in improving the clarity, grammar, and fluency of the text during the final editing stage. They do not contribute to the design of the research presented in this paper. All ideas, analyses, and conclusions are the sole work of the authors.

