# OpenReview forum: "Scaling Up, Speeding Up: A Benchmark of Speculative Decoding for Efficient LLM Test-Time Scaling"
_ICLR.cc/2026/Conference — ICLR 2026 Poster_

### Official Review · Reviewer_Hr8t · 2025-10-31

**Soundness:** 3
**Presentation:** 3
**Contribution:** 2
**Rating:** 4
**Confidence:** 3

**Summary:**

The paper presents comprehensive benchmark experiments evaluating three types of speculative decoding methods -- model-based, training-based, and N-gram-based -- in the context of LLM test-time scaling. The empirical results reveal several notable findings: (a) training-based methods exhibit robustness to sampling temperature, (b) N-gram-based methods are effective at identifying repeated patterns at the token level, and (c) hybrid approaches can leverage the strengths of both. These findings offer insights into the behavior of different approaches for reasoning models and may inspire further research on more effective methods for test-time scaling.

**Strengths:**

1. Empirically evaluates an important setting for the use of speculative decoding, i.e., test-time scaling for reasoning models. This setting can particularly benefit from more efficient inference, and understanding what works and why is highly valuable.

2. Evaluates a wide range of speculative decoding methods including model-based, training-based, N-gram-based, and hybrid methods.

3. Reports several notable findings, suggesting that hybrid approaches may be particularly promising.

**Weaknesses:**

1. The datasets used for the benchmark experiments are mostly focused on math reasoning. Evaluating how the findings extend to other types of data, e.g., coding, general reasoning, social reasoning, would provide more valuable insights into the methods.

2. Limited model scales. Results for models of scales other than 8B would strengthen the findings.

3. While the paper reports multiple notable findings regarding how different types of methods perform on the benchmarks, technical novelty (e.g., proposing a better working hybrid method) is rather limited.

**Questions:**

Q. Have the authors experimented with models of other scales, other types of reasoning benchmarks, etc.?

Q. How large is the variance for the performance numbers, e.g., reported in Tables 3 and 4.

---

> ### Author Response · Authors · 2025-11-21
> **Response to The Reviewer [1 / 3]**
>
> * **[Weakness 1][Generalization to Non-Mathematical Reasoning Tasks]** Thanks for the valuable comment! Following your suggestion, we have extended our evaluation to non-mathematical reasoning tasks, including code generation (LiveCodeBench) and summarization (CNN/Daily Mail). The results are as follows:
>     |LiveCodeBench|DSL-8B (T=0)|DSL-8B (T=0.6)|QW3-8B (T=0)|QW3-8B (T=0.6)|
>     |:---|:---:|:---:|:---:|:---:|
>     |Method|MAT / Speed|MAT / Speed|MAT / Speed|MAT / Speed|
>     |EAGLE-3|2.12 / 1.64x|2.00 / 1.33x|4.37 / 2.85x|4.02 / 2.58x|
>     |SAM[EAGLE-3]|4.17 / 3.34x|2.30 / 1.80x|4.46 / 3.27x|3.70 / 2.54x|
>     |SAM|2.90 / 2.79x|1.72 / 1.44x|2.19 / 1.92x|1.77 / 1.58x|
>     |Recycling|2.91 / 2.02x|2.69 / 1.71x|2.90 / 1.98x|2.83 / 1.92x|
>     |PLD|2.24 / 1.81x|-|1.94 / 1.57x|-|
>     |SpS|-|-|5.93 / 0.83x|5.16 / 0.85x|
>     |REST|1.33 / 1.11x|1.37 / 1.02x|1.35 / 1.14x|1.37 / 1.09x|
>     |Lookahead|2.15 / 1.57x|-|2.08 / 1.53x|-|
>     |PIA|2.40 / 1.71x|1.63 / 1.04x|2.04 / 1.46x|1.72 / 1.32x|
>     |AR|1.00 / 1.00x|1.00 / 1.00x|1.00 / 1.00x|1.00 / 1.00x|
>
>     |CNN/Daily Mail|DSL-8B (T=0)|DSL-8B (T=0.6)|QW3-8B (T=0)|QW3-8B (T=0.6)|
>     |:---|:---:|:---:|:---:|:---:|
>     |Method|MAT / Speed|MAT / Speed|MAT / Speed|MAT / Speed|
>     |EAGLE-3|4.76 / 2.83x|4.60 / 2.65x|4.11 / 2.63x|4.01 / 2.53x|
>     |SAM[EAGLE-3]|4.92 / 3.05x|4.70 / 2.79x|4.31 / 2.77x|4.17 / 2.61x|
>     |SAM|1.53 / 1.45x|1.39 / 1.33x|1.55 / 1.45x|1.52 / 1.42x|
>     |Recycling|2.57 / 1.81x|2.50 / 1.71x|2.73 / 1.95x|2.70 / 1.86x|
>     |PLD|1.47 / 1.32|-|1.46 / 1.30x|-|
>     |SpS|-|-|3.57 / 0.51|3.44 / 0.54
>     |REST|1.28 / 1.04x|1.29 / 1.00x|1.28 / 1.05x|1.28 / 1.01x|
>     |Lookahead|1.59 / 1.29|-|1.61 / 1.30x|-|
>     |PIA|1.61 / 1.42x|1.48 / 1.30x|1.58 / 1.39x|1.55 / 1.36x|
>     |AR|1.00 / 1.00x|1.00 / 1.00x|1.00 / 1.00x|1.00 / 1.00x|
>
>    We observe that on the code and summarization datasets, SAM[EAGLE-3] consistently achieves the highest speedup/MAT across nearly all scenarios. Moreover, the N-gram-based methods can even outperform the training-based method in some cases. For example, both SAM and Recycling achieve higher acceleration ratios than EAGLE-3 when applied to DeepSeek-R1-Distill-Llama-8B on the LiveCodeBench. These observations are consistent with the findings reported in our manuscript. This indicates that redundancy is **not specific to mathematical reasoning but a universal phenomenon** in generation tasks. The consistent superiority of hybrid methods across domains further validates our benchmark's core insights. We have also included these results and the related discussion in the revised manuscript (please refer to Appendix A.3, page 19 of the revised manuscript).
>
>
> * **[Weakness 2][Model Scale Analysis]** Thanks for the insightful comment! We fully agree that investigating behavior across different model scales is crucial for establishing the generalizability of our findings. We would like to point out that **we have actually conducted an analysis of speculative decoding across models with different scales in Appendix A.3 (Table 6)**. In addition to evaluating Qwen3-8B models, we further investigate the effectiveness of speculative decoding methods across different model scales, including the smaller Qwen3-4B and larger Qwen3-14B models. Key findings include:
>   - SAM[EAGLE-3] achieves the best performance across all model sizes, demonstrating the potential of hybrid speculative decoding methods (this validates Takeaway 4). This cross-scale consistency indicates that the N‑gram-based method retains its ability to capture redundancy effectively across model sizes.
>   - EAGLE-3 shows slight performance degradation when accelerating 4B/14B models compared to the 8B scenario. This stems from the fact that the EAGLE-3 draft models for 4B/14B come from a different team (AngelSlim), highlighting how training-based speculative decoding methods are highly dependent on the draft model's training quality (this validates Takeaway 1).
>   - SpS exhibits a more pronounced speedup ratio as the size gap between the target and draft models increases, suggesting that a larger capacity difference enables more effective speculation.
>
>   More details can be found in Appendix A.3 (page 18 of the revised manuscript). We hope these results address your concern regarding model scale variations.

---

> ### Author Response · Authors · 2025-11-21
> **Response to The Reviewer [2 / 3]**
>
> * **[Weakness 3][Novelty & Contribution]** Thanks for the comment! We appreciate your concern and would like to clarify the following points:
>   - **The Motivation of our Benchmark** As a benchmark, we acknowledge that our contributions differ from method-proposing works. We aim to systematically evaluate existing speculative decoding methods in the context of test-time scaling to reveal their strengths, weaknesses, and hidden challenges. In this context, **if such a straightforward combination already yields a substantial performance gain, it strongly indicates the presence of extensive, underutilized redundancy**. This finding reinforces our benchmark’s central insight that, in the context of test-time scaling, there remains abundant redundancy that has yet to be effectively leveraged.
>   - **A Simple Hybrid Strategy Already Works** First, we emphasize that the simple hybrid strategy benchmarked in this paper demonstrates remarkable effectiveness, as evidenced by the results in Tables 3 and 4 of our manuscript. Furthermore, our analysis of the strategy’s potential in Appendix A.3 (Figure 3) reveals that, with a well-chosen static threshold, the simple hybrid method can achieve a MAT exceeding 8, which is nearly double that of the SOTA EAGLE-3. Although dynamic thresholding could yield higher MAT, it introduces a crucial trade-off between complexity and efficiency. **Therefore, we posit that this simple, low-overhead hybrid strategy is a well-suited baseline to inspire future work.** Nevertheless, despite the potential risk of increased model complexity, we have, as suggested, incorporated several possible hybrid directions into the revised manuscript, including: (i) Context-Aware Switching, where the strategy adapts dynamically based on the semantic context of the generation task; (ii) Phase-Specific Optimization, where the strategy is tailored to different stages of the generation process; and (iii) Uncertainty-Based Toggling, where the strategy is adjusted on-the-fly according to the model’s predictive uncertainty. Further elaboration on these three potential directions can be found in the revised manuscript, and we remain open to further discussion on this point.
>   - **Novelty & Contribution** Our work addresses three critical gaps in the field: **(i) Filling a critical but overlooked gap in the application context.** Test-time scaling presents a fundamentally different computational landscape: high repetition in reasoning traces (Table 1), extended sequences exceeding thousands of tokens, and multiple iterative reasoning paths. Despite its growing importance, no prior work has systematically studied speculative decoding in this setting. **(ii) Revealing counterintuitive findings that challenge conventional wisdom.** Our benchmark uncovers several unexpected phenomena, including but not limited to: simple n-gram methods can rival sophisticated trained models in certain scenarios; training-based methods exhibit unexpected fragility in test-time scaling paradigms; and hybrid approaches show promise yet remain significantly underoptimized. **(iii) Contributing standardized evaluation protocols.** Our benchmark systematically evaluates 9 SOTA methods across 3 categories, 4 reasoning tasks (AIME24/25, MATH-500, GPQA), and 2 test-time scaling paradigms, with ablations over key parameters to facilitate reproducible research.
>
>   Following your helpful suggestion, we have revised the manuscript accordingly (please refer to Appendix A.5, page 29 of the revised manuscript). We welcome any further discussion on this point!
>
> * **[Question 1][Model Scale Analysis & Other Types of Reasoning Tasks]** Thanks for the comment! We have conducted the corresponding experiments. Please refer to our response to Weakness 2 for the model scale analysis, and to Weakness 1 for results on other types of reasoning tasks.

---

> ### Author Response · Authors · 2025-11-21
> **Response to The Reviewer [3 / 3]**
>
> * **[Question 2][Error Analysis]** Thanks for the constructive comment! Due to time and computational constraints, we conduct error analysis on a representative subset of methods (EAGLE-3, SAM[EAGLE-3, and Recycling) by running them three times. We respectfully note that a complete evaluation across all methods and datasets would require over 2,500 A6000 GPU hours, which is challenging within the review timeline. The results from our partial analysis demonstrate that the impact of randomness is minimal, with **standard deviations consistently below 0.1** across tested configurations (Qwen3-8B, Multi-round Thinking). We have included these error bars in the revised manuscript (please refer to Appendix A.3, page 20 of the revised manuscript).
>
>     | Qwen3-8B (T=0) | AIME24 | AIME25 | MATH500 | GPQA | Overall |
>     |-|-|-|-|-|-|
>     | Method | MAT / Speed | MAT / Speed | MAT / Speed | MAT / Speed | MAT / Speed |
>     | EAGLE-3 | 4.31±0.00 / 2.77±0.03 | 4.39±0.02 / 2.87±0.01 | 4.50±0.00 / 3.01±0.03 | 4.39±0.00 / 2.89±0.02 | 4.38±0.00 / 2.88±0.02 |
>     | SAM[EAGLE-3] | 4.32±0.07 / 3.00±0.07 | 4.46±0.01 / 3.15±0.03 | 4.91±0.08 / 3.43±0.04 | 5.80±0.09 / 4.10±0.06 | 4.72±0.03 / 3.43±0.05 |
>     | Recycling | 2.99±0.03 / 2.03±0.04 | 2.96±0.02 / 2.07±0.06 | 2.99±0.01 / 2.15±0.05 | 3.06±0.01 / 2.14±0.04 | 3.00±0.01 / 2.10±0.05 |
>     | AR | 1.00±0.00 / 1.00±0.00 | 1.00±0.00 / 1.00±0.00 | 1.00±0.00 / 1.00±0.00 | 1.00±0.00 / 1.00±0.00 | 1.00±0.00 / 1.00±0.00 |
>
>     | Qwen3-8B (T=0.6) | AIME24 | AIME25 | MATH500 | GPQA | Overall |
>     |-|-|-|-|-|-|
>     | Method | MAT / Speed | MAT / Speed | MAT / Speed | MAT / Speed | MAT / Speed |
>     | EAGLE-3 | 4.12±0.01 / 2.60±0.01 | 4.21±0.02 / 2.71±0.01 | 4.34±0.02 / 2.85±0.02 | 4.02±0.02 / 2.60±0.07 | 4.16±0.01 / 2.70±0.03 |
>     | SAM[EAGLE-3] | 3.89±0.04 / 2.61±0.06 | 3.94±0.01 / 2.70±0.03 | 4.18±0.01 / 2.90±0.03 | 3.99±0.03 / 2.72±0.09 | 3.97±0.02 / 2.74±0.05 |
>     | Recycling | 2.89±0.01 / 1.95±0.05 | 2.90±0.02 / 1.97±0.06 | 2.90±0.01 / 2.05±0.06 | 2.92±0.01 / 2.00±0.07 | 2.90±0.00 / 2.00±0.06 |
>     | AR | 1.00±0.00 / 1.00±0.00 | 1.00±0.00 / 1.00±0.00 | 1.00±0.00 / 1.00±0.00 | 1.00±0.00 / 1.00±0.00 | 1.00±0.00 / 1.00±0.00 |

---

> ### Author Response · Authors · 2025-11-26
> **Looking forward to your feedback!**
>
> Dear Reviewer Hr8t,
>
> Thank you once again for your constructive feedback! We truly appreciate your recognition of the importance of our problem setting and the comprehensiveness of our evaluation. Guided by your suggestions, we have made our best effort to conduct additional experiments and provide clarifications in our response. We hope that our answers have clarified the issues raised and that you might consider re-evaluating our paper. We remain available to answer any further questions!
>
> Best regards,
> Authors

---

### Official Review · Reviewer_eh5y · 2025-10-31

**Soundness:** 2
**Presentation:** 3
**Contribution:** 3
**Rating:** 4
**Confidence:** 3

**Summary:**

The paper presents the first benchmark for evaluating speculative decoding methods in test-time scaling of LLMs, offering standardized protocols across paradigms like Best-of-N sampling and multi-round thinking. Experiments show that simple N-gram-based methods efficiently handle repetitive reasoning patterns, suggesting their integration with model- or training-based approaches can make test-time scaling both faster and more effective.

**Strengths:**

1. The paper tackes an important problem of evaluating speculative decoding approaches in test-time scaling scenarios, and provide a formal benchmark.
2. The authors perform extensive evaluations ranging in nine different speculative decoding methods, four datasets, and two LLMs.
3. The authors provide a comprehensive analysis of the experiment results, along with clear takeaways for the readers.

**Weaknesses:**

1. As far as the reviewer understands, the reasoning traces were sampled sequentially in best-of-N scenarios, using batch size 1 (L264). However in realistic scenarios, these traces would be sampled in parallel (i.e. using batch decoding). This suggests a critical gap between the benchmark performance and real-world benefits.

2. The benchmark does not consider test-time tree search (e.g. beam search or DVTS [1]), which is another major test-time scaling strategy. (Note: This weakness is not critical, as the reviewer understands that it might add too much complexity to the benchmark.)

[1] Beeching et. al., Scaling Test Time Compute with Open Models

**Questions:**

Would the Best-of-N speedups be different if the reasoning traces were samples in a batch? (e.g. The amount of overlap would be much smaller for N-gram based models)

---

> ### Author Response · Authors · 2025-11-21
> **Response to The Reviewer [1 / 3]**
>
> * **[Weakness 1][Batched BoN Setup]** Thanks for the valuable comment! Please refer to our response to Question 1 for a detailed discussion on this point.
> * **[Weakness 2]** .Thanks for the comment! We agree this is an interesting and forward-looking direction, and we appreciate the reviewer’s understanding of the added complexity.
>     - **Technical Challenges** A key distinction makes the direct application of existing speculative techniques to tree search-based reasoning non-trivial. Unlike the relatively linear generation process in BoN or multi-round thinking, these search-based frameworks involve dynamic operations such as state evaluation, pruning, and backtracking. Existing speculative decoding methods, whether N‑gram-based or advanced variants using suffix automata (e.g., SAM) or adjacency matrices (e.g., Recycling), are designed for monotonic forward generation with a single contiguous prefix. Training‑based methods such as Eagle‑3 share the same limitation. Therefore, adapting them to search-based frameworks would require modifying their underlying write and lookup logic, for example, by maintaining separate speculative modules (such as individual SAM) for each search step, or extending data structures such as suffix automata and adjacency matrices to handle deletion operations, since certain steps may be pruned during the search.
>     - **Tree Search Frameworks also have Redundancy** Conceptually, the frameworks mentioned in the comment can be regarded as localized, step-level test-time scaling methods. Instead of generating multiple complete responses as in BoN or multi-round thinking, these frameworks expand and evaluate several reasoning paths at each intermediate step. This localized exploration inevitably introduces substantial overlap and redundancy among candidate steps, a phenomenon closely related to the computational inefficiency examined in this benchmark. Therefore, the findings and insights from our benchmark remain relevant to understanding similar redundancy patterns in tree search frameworks. This is indeed a highly promising direction for future work; however, as the reviewer rightly noted, fully supporting such dynamic search procedures would substantially increase the complexity and slightly beyond the scope of this paper. We leave it for future exploration.
>
>   We have added a discussion of this point in the revised manuscript (please refer to Appendix A.4, page 29 of the revised manuscript).

---

> ### Author Response · Authors · 2025-11-21
> **Response to The Reviewer [2 / 3]**
>
> * **[Question 1][Batched BoN Setup][Part-1]** Thanks for the valuable comment!  We would like to elaborate on this issue from three complementary perspectives:
>   - **Constraint from Reference Implementations** Our work primarily serves as a benchmark, and to ensure objectivity and fairness, we intentionally avoid altering the logic of the evaluated methods’ original implementations. However, it is worth noting that many existing codes are not well-suited for batched inference. For example:
>     - The official implementation of EAGLE-3 does not support batched inference. While we are aware that certain third-party frameworks, such as vLLM and SGLang, support batched inference for EAGLE-3, they do not cover other baselines, preventing a unified comparison in a consistent environment.
>     - The SAM method’s suffix automaton does not support batched writing or lookup.
>     - Recycling does not support batched adjacency matrix construction.
>
>   Given the architectural constraints, implementing varying batch sizes would require significant modifications to the original codebases. To ensure strict fidelity to the original implementations and guarantee a fair baseline comparison, we prioritized maintaining the current setup. While we agree that this direction is valuable, we believe it would be best addressed in future work to avoid introducing confounding variables in the current benchmark.
>
>   - **Generalizability to Batched Settings** We maintain that our single-batch results serve as a reliable proxy for performance in moderate-batch regimes due to the memory-bound nature of LLM decoding. Speculative decoding improves hardware utilization by verifying multiple tokens in a single forward pass, thereby reducing the frequency of memory accesses and overall latency. From a roofline model perspective [1], while extremely large batch sizes may eventually shift the bottleneck to computation where verification overhead could negate gains, increasing the batch size within the memory-bound region before reaching the compute-bound consistently yields inference acceleration. Crucially, for N-gram-based methods, the cost of draft generation is negligible, making the speedup highly positively correlated with the Mean Accepted Tokens (MAT). Consequently, the high MAT observed in our single-batch experiments serves as a strong indicator of potential speedups in batched scenarios. We focused on single-batch inference to isolate these intrinsic algorithmic capabilities from the uneven system-level engineering [2] required to optimize batched execution across different baselines.
>     - [1] LLM Inference Unveiled: Survey and Roofline Model Insights, Yuan et al., 2024.
>     - [2] Efficient Speculative Decoding for Llama at Scale: Challenges and Solutions, Meta, 2025.
>   ```
>   A friendly reminder to check the next response for [Question 1][Batched BoN Setup][Part-2].
>   ```

---

> ### Author Response · Authors · 2025-11-21
> **Response to The Reviewer [3 / 3]**
>
> * **[Question 1][Batched BoN Setup][Part-2]**
>   - **Effort on Batch Try** Despite the challenges discussed above, we devote considerable effort these days to implementing preliminary batched versions of EAGLE-3 and SAM for exploration under the BON scenario. For SAM, to enable batched inference, we modify the suffix automaton’s query mechanism: instead of maintaining a single shared automaton, we construct one for each reasoning path, and draft tokens across branches are cross-looked-up for matches. While this design makes batching possible, it inevitably increases the query complexity beyond O(1), representing a necessary compromise under the current framework. We test BON with N = 4 and N = 8 on a subset of the four reasoning tasks using DeepSeek-R1-Distill-Llama-8B. The results are as follows (where SAM-Single is the original implementation using a single automaton per reasoning path, and SAM-Cross is the modified version that supports cross–branch token look-ups):
>
>     |Method|AIME24|AIME25|MATH500|GPQA|
>     |-|-|-|-|-|
>     |Batch Size = 4|MAT / Speed|MAT / Speed|MAT / Speed|MAT / Speed|
>     |EAGLE-3|3.30 / 1.52x|3.34 / 1.56x|3.10 / 1.46x|3.08 / 1.41x|
>     |SAM-Single|1.81 / 1.33x|1.87 / 1.44x|2.91 / 2.72x|2.01 / 1.58x|
>     |SAM-Cross|1.86 / 1.34x|2.11 / 1.62x|3.19 / 2.89x|2.21 / 1.73x|
>
>     |Method|AIME24|AIME25|MATH500|GPQA|
>     |-|-|-|-|-|
>     |Batch Size = 8|MAT / Speed|MAT / Speed|MAT / Speed|MAT / Speed|
>     |EAGLE-3|3.34 / 1.17x|3.37 / 1.18x|3.13 / 1.11x|3.06 / 1.03x|
>     |SAM-Single|1.75 / 1.06x|1.81 / 1.18x|2.52 / 2.24x|1.92 / 1.36x|
>     |SAM-Cross|1.82 / 1.12x|1.92 / 1.20x|3.12 / 2.61x|1.99 / 1.34x|
>
>     The results suggest that increasing the batch size indeed tends to trigger computation-bound behavior. For instance, EAGLE-3 achieves an average speedup of around 1.5x at batch size 4, which is notably lower than its MAT and, in some cases, even slightly behind the N-gram-based method SAM. This observation is consistent with our earlier findings in the benchmark, indicating that redundancy plays a crucial role in test-time scaling acceleration. In practice, when the batch size increases, the speculative size typically needs to be reduced to avoid becoming compute-bound; the observed slowdown is therefore likely due to the current testing configuration, which did not carefully calibrate this trade-off. While we anticipate improved speedups with further tuning, we leave this extensive optimization for future work due to time limitations. In addition, the parallelization of BON inherently reduces overlapping redundancy across different inference paths, which further explains the diminishing speedup. However, this does not contradict our benchmark insights. On the contrary, it reinforces our conclusion that redundancy should be preserved and fully utilized in speculative decoding. Our benchmark, therefore, substantiates a key practical insight that BoN should be applied along the sequential dimension, while the batch dimension should be reserved for handling different queries to maximize throughput (as leaving the available redundancy unexploited would be an unacceptable waste), ensuring that redundancy is maintained and contributes to acceleration.
>
>   More details can be found in Appendix A.3 (page 20 of the revised manuscript). We hope these additional results are helpful in addressing your concern.

---

> ### Author Response · Authors · 2025-11-26
> **Looking forward to your feedback!**
>
> Dear Reviewer eh5y,
>
> Thank you once again for your valuable comments! We truly appreciate that you acknowledged the importance of our problem setting and our evaluation efforts. In response to your comments, we have diligently worked to incorporate new experiments and clarify the points you raised to the best of our ability. We hope these updates help clarify the issues and that you will consider re-evaluating our paper. Please let us know if you have any further questions or remaining concerns!
>
> Best regards,
> Authors

---

### Official Review · Reviewer_mU4f · 2025-10-31

**Soundness:** 3
**Presentation:** 3
**Contribution:** 3
**Rating:** 6
**Confidence:** 3

**Summary:**

The paper discusses different methods of speculative decoding and tests them across reasoning benchmarks to measure MAT/wall clock speed up.
The paper is a little limited in scope and could have been a bit more thorough. If the authors wouldn't mind running this on some larger benchmarks with different scaled models, I believe this would make a very nice paper. I'm still content with the paper, so I recommend acceptance.

**Strengths:**

1. It's a nice paper with useful results.
2. The Takeaways are a pretty good analysis of what's going on with each speculative decoding method.
3. Presentation is pretty nice.
4. Practical results.

**Weaknesses:**

1. It's not really a benchmark. It's just running the different methods on common benchmarks. Not a huge deal since it's still good science.
2. WOuld have been nice to scale these across difficult model scales. The results may have been different.
3. Datasets are tiny, like 120 total samples. Could have done this on more deata.
4. Error bars could have been nice if you could rerun this multiple times to see what happens.
5. Varying batch size would have been nice.

**Questions:**

N/A

---

> ### Author Response · Authors · 2025-11-21
> **Response to The Reviewer [1 / 4]**
>
> * **[Weakness 1][Assesment on Benchmark]** Thanks for the thoughtful comment! We completely understand your concern and would like to clarify what we mean by “benchmark” in our work and how it differs from existing ones.
>   - **A New Benchmark Setting Focused on Test-Time Scaling** Prior works mainly evaluated speculative decoding under single-turn inference or standard QA scenarios. In contrast, our study systematically investigates multi-round thinking and BON-style test-time scaling settings, scenarios that had not been benchmarked or standardized before. These settings exhibit distinct characteristics such as repetitive and extended reasoning traces, diverse reasoning patterns, and significantly longer sequence lengths. We thus position our work as a new benchmark suite specifically designed for this underexplored domain.
>   - **Uncovering Counterintuitive Findings** Our systematic evaluation reveals several surprising patterns: simple N-gram methods can match or even outperform sophisticated trained models in certain contexts; training-based approaches show unexpected fragility across diverse reasoning scenarios; and hybrid methods, while promising, remain significantly underoptimized. These findings provide practical insights and guidance for both practitioners and researchers.
>   - **Establishing Standardized Evaluation Protocols** We evaluate 9 SOTA methods across 3 categories, 4 reasoning tasks, and 2 representative test-time scaling paradigms, supported by comprehensive ablations. This benchmark promotes reproducible research and fair comparison in this emerging area. We believe that rigorous empirical evaluation plays an essential role in advancing understanding and driving future progress in speculative decoding.
>
>   Overall, we truly appreciate your fair assessment and are open to any further discussion!
>
> * **[Weakness 2][Model Scale Analysis]** Thanks for the valuable comment! We completely agree that examining the behavior across different model scales is important for understanding the generalizability of our findings. We would like to point out that **we have actually conducted an analysis of speculative decoding across models with different scales in Appendix A.3 (Table 6)**. In addition to evaluating Qwen3-8B models, we further investigate the effectiveness of speculative decoding methods across different model scales, including the smaller Qwen3-4B and larger Qwen3-14B models. Key findings include:
>   - SAM[EAGLE-3] achieves the best performance across all model sizes, demonstrating the potential of hybrid speculative decoding methods (this validates Takeaway 4). This cross-scale consistency indicates that the N‑gram-based method retains its ability to capture redundancy effectively across model sizes.
>   - EAGLE-3 shows slight performance degradation when accelerating 4B/14B models compared to the 8B scenario. This stems from the fact that the EAGLE-3 draft models for 4B/14B come from a different team (AngelSlim), highlighting how training-based speculative decoding methods are highly dependent on the draft model's training quality (this validates Takeaway 1).
>   - SpS exhibits a more pronounced speedup ratio as the size gap between the target and draft models increases, suggesting that a larger capacity difference enables more effective speculation.
>
>   More details can be found in Appendix A.3 (page 18 of the revised manuscript). We hope these results address your concern regarding model scale variations.

---

> ### Author Response · Authors · 2025-11-21
> **Response to The Reviewer [2 / 4]**
>
> * **[Weakness 3][Dataset Size]** Thanks for the insightful comment! We appreciate the opportunity to clarify the following two key points:
>   - **Dataset Expansion** Following your suggestion, we have extended our dataset with 30 code generation samples (LiveCodeBench) and 30 text summarization samples (CNN/Daily Mail), as shown in the table below:
>     |LiveCodeBench|DSL-8B (T=0)|DSL-8B (T=0.6)|QW3-8B (T=0)|QW3-8B (T=0.6)|
>     |:---|:---:|:---:|:---:|:---:|
>     |Method|MAT / Speed|MAT / Speed|MAT / Speed|MAT / Speed|
>     |EAGLE-3|2.12 / 1.64x|2.00 / 1.33x|4.37 / 2.85x|4.02 / 2.58x|
>     |SAM[EAGLE-3]|4.17 / 3.34x|2.30 / 1.80x|4.46 / 3.27x|3.70 / 2.54x|
>     |SAM|2.90 / 2.79x|1.72 / 1.44x|2.19 / 1.92x|1.77 / 1.58x|
>     |Recycling|2.91 / 2.02x|2.69 / 1.71x|2.90 / 1.98x|2.83 / 1.92x|
>     |PLD|2.24 / 1.81x|-|1.94 / 1.57x|-|
>     |SpS|-|-|5.93 / 0.83x|5.16 / 0.85x|
>     |REST|1.33 / 1.11x|1.37 / 1.02x|1.35 / 1.14x|1.37 / 1.09x|
>     |Lookahead|2.15 / 1.57x|-|2.08 / 1.53x|-|
>     |PIA|2.40 / 1.71x|1.63 / 1.04x|2.04 / 1.46x|1.72 / 1.32x|
>     |AR|1.00 / 1.00x|1.00 / 1.00x|1.00 / 1.00x|1.00 / 1.00x|
>
>     |CNN/Daily Mail|DSL-8B (T=0)|DSL-8B (T=0.6)|QW3-8B (T=0)|QW3-8B (T=0.6)|
>     |:---|:---:|:---:|:---:|:---:|
>     |Method|MAT / Speed|MAT / Speed|MAT / Speed|MAT / Speed|
>     |EAGLE-3|4.76 / 2.83x|4.60 / 2.65x|4.11 / 2.63x|4.01 / 2.53x|
>     |SAM[EAGLE-3]|4.92 / 3.05x|4.70 / 2.79x|4.31 / 2.77x|4.17 / 2.61x|
>     |SAM|1.53 / 1.45x|1.39 / 1.33x|1.55 / 1.45x|1.52 / 1.42x|
>     |Recycling|2.57 / 1.81x|2.50 / 1.71x|2.73 / 1.95x|2.70 / 1.86x|
>     |PLD|1.47 / 1.32|-|1.46 / 1.30x|-|
>     |SpS|-|-|3.57 / 0.51|3.44 / 0.54
>     |REST|1.28 / 1.04x|1.29 / 1.00x|1.28 / 1.05x|1.28 / 1.01x|
>     |Lookahead|1.59 / 1.29|-|1.61 / 1.30x|-|
>     |PIA|1.61 / 1.42x|1.48 / 1.30x|1.58 / 1.39x|1.55 / 1.36x|
>     |AR|1.00 / 1.00x|1.00 / 1.00x|1.00 / 1.00x|1.00 / 1.00x|
>
>     We observe that on the code and summarization datasets, SAM[EAGLE-3] consistently achieves the highest speedup across nearly all scenarios. Moreover, the N-gram-based methods can even outperform the training-based method in some cases. For example, both SAM and Recycling achieve higher acceleration ratios than EAGLE-3 when applied to DeepSeek-R1-Distill-Llama-8B on the LiveCodeBench. These observations are consistent with the findings reported in our manuscript. This indicates that the model’s reasoning process across different types of data exhibits redundancy, which is **a universal phenomenon** that can be effectively exploited to enhance speculative decoding. We have also included these results and the related discussion in the revised manuscript (please refer to Appendix A.3, page 19 of the revised manuscript).
>
>   - **Robustness of Acceleration Measurements** The measured speedup ratios are highly stable, as they are obtained from averages over hundreds of thousands of generated tokens, resulting in extremely small variance. Additionally, the patterns within each dataset category are relatively consistent, so increasing the sample size within a category has minimal impact on speedup ratios. To verify this, we test subsets of 10, 20, and 30 samples from each reasoning task; the results are as follows:
>     |T=0|AIME25|AIME25|AIME25|MATH500|MATH500|MATH500|
>     |:---|:---:|:---:|:---:|:---:|:---:|:---:|
>     |Method|#1-10|#1-20|#1-30|#1-10|#1-20|#1-30|
>     |EAGLE-3|2.89x|2.84x|2.87x|3.04x|3.04x|3.04x|
>     |SAM[EAGLE-3]|3.23x|3.18x|3.18x|3.43x|3.57x|3.47x|
>     |SAM|2.12x|2.09x|2.07x|1.95x|1.95x|1.95x|
>     |Recycling|2.14x|2.14x|2.13x|2.19x|2.19x|2.20x|
>     |PLD|1.68x|1.65x|1.66x|1.58x|1.63x|1.63x|
>     |SpS|0.96x|0.95x|0.96x|0.75x|0.77x|0.77x|
>     |REST|1.13x|1.14x|1.14x|1.17x|1.17x|1.16x|
>     |Lookahead|1.57x|1.56x|1.54x|1.58x|1.57x|1.58x|
>     |PIA|1.81x|1.73x|1.68x|1.50x|1.53x|1.63x|
>     |AR|1.00x|1.00x|1.00x|1.00x|1.00x|1.00x|
>
>
>     |T=0.6|AIME25|AIME25|AIME25|MATH500|MATH500|MATH500|
>     |:---|:---:|:---:|:---:|:---:|:---:|:---:|
>     |Method|#1-10|#1-20|#1-30|#1-10|#1-20|#1-30|
>     |EAGLE-3|2.77x|2.75x|2.71x|2.90x|2.87x|2.87x|
>     |SAM[EAGLE-3]|2.74x|2.73x|2.72x|2.95x|2.89x|2.93x|
>     |SAM|1.79x|1.79x|1.78x|1.72x|1.74x|1.74x|
>     |Recycling|2.06x|2.05x|2.04x|2.11x|2.10x|2.11x|
>     |SpS|0.92x|0.94x|0.95x|0.78x|0.80x|0.80x|
>     |REST|1.10x|1.11x|1.10x|1.12x|1.12x|1.12x|
>     |PIA|1.48x|1.45x|1.45x|1.42x|1.43x|1.45x|
>     |AR|1.00x|1.00x|1.00x|1.00x|1.00x|1.00x|
>
>     The results show remarkable stability, demonstrating that our measurements are robust even with smaller sample counts, which we hope addresses your concern regarding dataset size and measurement reliability.

---

> ### Author Response · Authors · 2025-11-21
> **Response to The Reviewer [3 / 4]**
>
> * **[Weakness 4][Error Analysis]** Thanks for the constructive comment! Due to time and computational constraints, we conduct error analysis on a representative subset of methods by running them three times. We respectfully note that a complete evaluation across all methods and datasets would require over 2,500 A6000 GPU hours, which is challenging within the review timeline. The results from our partial analysis demonstrate that the impact of randomness is minimal, with **standard deviations consistently below 0.1** across tested configurations (Qwen3-8B, Multi-round Thinking). We have included these error bars in the revised manuscript (please refer to Appendix A.3, page 20 of the revised manuscript).
>
>     | Qwen3-8B (T=0) | AIME24 | AIME25 | MATH500 | GPQA | Overall |
>     |-|-|-|-|-|-|
>     | Method | MAT / Speed | MAT / Speed | MAT / Speed | MAT / Speed | MAT / Speed |
>     | EAGLE-3 | 4.31±0.00 / 2.77±0.03 | 4.39±0.02 / 2.87±0.01 | 4.50±0.00 / 3.01±0.03 | 4.39±0.00 / 2.89±0.02 | 4.38±0.00 / 2.88±0.02 |
>     | SAM[EAGLE-3] | 4.32±0.07 / 3.00±0.07 | 4.46±0.01 / 3.15±0.03 | 4.91±0.08 / 3.43±0.04 | 5.80±0.09 / 4.10±0.06 | 4.72±0.03 / 3.43±0.05 |
>     | Recycling | 2.99±0.03 / 2.03±0.04 | 2.96±0.02 / 2.07±0.06 | 2.99±0.01 / 2.15±0.05 | 3.06±0.01 / 2.14±0.04 | 3.00±0.01 / 2.10±0.05 |
>     | AR | 1.00±0.00 / 1.00±0.00 | 1.00±0.00 / 1.00±0.00 | 1.00±0.00 / 1.00±0.00 | 1.00±0.00 / 1.00±0.00 | 1.00±0.00 / 1.00±0.00 |
>
>     | Qwen3-8B (T=0.6) | AIME24 | AIME25 | MATH500 | GPQA | Overall |
>     |-|-|-|-|-|-|
>     | Method | MAT / Speed | MAT / Speed | MAT / Speed | MAT / Speed | MAT / Speed |
>     | EAGLE-3 | 4.12±0.01 / 2.60±0.01 | 4.21±0.02 / 2.71±0.01 | 4.34±0.02 / 2.85±0.02 | 4.02±0.02 / 2.60±0.07 | 4.16±0.01 / 2.70±0.03 |
>     | SAM[EAGLE-3] | 3.89±0.04 / 2.61±0.06 | 3.94±0.01 / 2.70±0.03 | 4.18±0.01 / 2.90±0.03 | 3.99±0.03 / 2.72±0.09 | 3.97±0.02 / 2.74±0.05 |
>     | Recycling | 2.89±0.01 / 1.95±0.05 | 2.90±0.02 / 1.97±0.06 | 2.90±0.01 / 2.05±0.06 | 2.92±0.01 / 2.00±0.07 | 2.90±0.00 / 2.00±0.06 |
>     | AR | 1.00±0.00 / 1.00±0.00 | 1.00±0.00 / 1.00±0.00 | 1.00±0.00 / 1.00±0.00 | 1.00±0.00 / 1.00±0.00 | 1.00±0.00 / 1.00±0.00 |
>
> * **[Weakness 5][Varying Batch Size][Part-1]** Thanks for the insightful comment! We would like to address this point from three perspectives:
>   - **Constraint from Reference Implementations** Our work primarily serves as a benchmark, and to ensure objectivity and fairness, we intentionally avoid altering the logic of the evaluated methods’ original implementations. However, it is worth noting that many existing codes are not well-suited for batched inference. For example:
>     - The official implementation of EAGLE-3 does not support batched inference. While we are aware that certain third-party frameworks, such as vLLM and SGLang, support batched inference for EAGLE-3, they do not cover other baselines, preventing a unified comparison in a consistent environment.
>     - The SAM method’s suffix automaton does not support batched writing or lookup.
>     - Recycling does not support batched adjacency matrix construction.
>
>     Given the architectural constraints, implementing varying batch sizes would require significant modifications to the original codebases. To ensure strict fidelity to the original implementations and guarantee a fair baseline comparison, we prioritized maintaining the current setup. While we agree that this direction is valuable, we believe it would be best addressed in future work to avoid introducing confounding variables in the current benchmark.
>   ```
>   A friendly reminder to check the next response for [Weakness 5][Varying Batch Size][Part-2].
>   ```

---

> ### Author Response · Authors · 2025-11-21
> **Response to The Reviewer [4 / 4]**
>
> * **[Weakness 5][Varying Batch Size][Part-2]**
>   - **Generalizability to Batched Settings** We maintain that our single-batch results serve as a reliable proxy for performance in moderate-batch regimes due to the memory-bound nature of LLM decoding. Speculative decoding improves hardware utilization by verifying multiple tokens in a single forward pass, thereby reducing the frequency of memory accesses and overall latency. From a roofline model perspective [1], while extremely large batch sizes may eventually shift the bottleneck to computation where verification overhead could negate gains, increasing the batch size within the memory-bound region before reaching the compute-bound consistently yields inference acceleration. Crucially, for N-gram-based methods, the cost of draft generation is negligible, making the speedup highly positively correlated with the Mean Accepted Tokens (MAT). Consequently, the high MAT observed in our single-batch experiments serves as a strong indicator of potential speedups in batched scenarios. We focused on single-batch inference to isolate these intrinsic algorithmic capabilities from the uneven system-level engineering [2] required to optimize batched execution across different baselines.
>     - [1] LLM Inference Unveiled: Survey and Roofline Model Insights, Yuan et al., 2024.
>     - [2] Efficient Speculative Decoding for Llama at Scale: Challenges and Solutions, Meta, 2025.
>
>   - **Effort on Batch Try** Despite the challenges discussed above, we devote considerable effort these days to implementing preliminary batched versions of EAGLE-3 and SAM for exploration under the BON scenario. For SAM, to enable batched inference, we modify the suffix automaton’s query mechanism: instead of maintaining a single shared automaton, we construct one for each reasoning path, and draft tokens across branches are cross-looked-up for matches. While this design makes batching possible, it inevitably increases the query complexity beyond O(1), representing a necessary compromise under the current framework. We test BON with N = 4 and N = 8 on a subset of the four reasoning tasks using DeepSeek-R1-Distill-Llama-8B. The results are as follows (where SAM-Single is the original implementation using a single automaton per reasoning path, and SAM-Cross is the modified version that supports cross–branch token look-ups):
>
>     |Method|AIME24|AIME25|MATH500|GPQA|
>     |-|-|-|-|-|
>     |Batch Size = 4|MAT / Speed|MAT / Speed|MAT / Speed|MAT / Speed|
>     |EAGLE-3|3.30 / 1.52x|3.34 / 1.56x|3.10 / 1.46x|3.08 / 1.41x|
>     |SAM-Single|1.81 / 1.33x|1.87 / 1.44x|2.91 / 2.72x|2.01 / 1.58x|
>     |SAM-Cross|1.86 / 1.34x|2.11 / 1.62x|3.19 / 2.89x|2.21 / 1.73x|
>
>     |Method|AIME24|AIME25|MATH500|GPQA|
>     |-|-|-|-|-|
>     |Batch Size = 8|MAT/Speed|MAT/Speed|MAT/Speed|MAT/ Speed|
>     |EAGLE-3|3.34/ 1.17x|3.37 / 1.18x|3.13 / 1.11x|3.06 / 1.03x|
>     |SAM-Single|1.75 / 1.06x|1.81 / 1.18x|2.52 / 2.24x|1.92 / 1.36x|
>     |SAM-Cross|1.82 / 1.12x|1.92 / 1.20x|3.12 / 2.61x|1.99 / 1.34x|
>
>     The results suggest that increasing the batch size indeed tends to trigger computation-bound behavior. For instance, EAGLE-3 achieves an average speedup of around 1.5x at batch size 4, which is notably lower than its MAT and, in some cases, even slightly behind the N-gram-based method SAM. This observation is consistent with our earlier findings in the benchmark, indicating that redundancy plays a crucial role in test-time scaling acceleration. In practice, when the batch size increases, the speculative size typically needs to be reduced to avoid becoming compute-bound; the observed slowdown is therefore likely due to the current testing configuration, which did not carefully calibrate this trade-off. While we anticipate improved speedups with further tuning, we leave this extensive optimization for future work due to time limitations. In addition, the parallelization of BON inherently reduces overlapping redundancy across different inference paths, which further explains the diminishing speedup. However, this does not contradict our benchmark insights. On the contrary, it reinforces our conclusion that redundancy should be preserved and fully utilized in speculative decoding. Our benchmark, therefore, substantiates a key practical insight that BoN should be applied along the sequential dimension, while the batch dimension should be reserved for handling different queries to maximize throughput (as leaving the available redundancy unexploited would be an unacceptable waste), ensuring that redundancy is maintained and contributes to acceleration.
>
>   More details can be found in Appendix A.3 (page 20 of the revised manuscript). We hope the new results clarify the issue.

---

> ### Author Response · Authors · 2025-11-26
> **Looking forward to your feedback!**
>
> Dear Reviewer mU4f,
>
> Thank you again for the time and effort you’ve dedicated to reviewing our work! As the discussion phase is drawing to a close, we would be very grateful if you could review our responses and the extensive additional experiments we have provided. We hope that our answers have addressed your concerns and that you might consider re-evaluating our paper. We welcome further discussion if there are any unclear points!
>
> Best regards,
> Authors

---

### Official Review · Reviewer_7EPL · 2025-11-01

**Soundness:** 3
**Presentation:** 3
**Contribution:** 2
**Rating:** 6
**Confidence:** 2

**Summary:**

This is a benchmark paper that targets on accelerating test time scaling method by speculative decoding. It compares model-based, training-based, and N-gram approaches.
The key finding is that simple N-gram-based methods (like SAM) are highly effective at capturing this redundancy. A hybrid method, SAM[EAGLE-3], achieved the best overall speedup by combining N-gram's repetition capture with a training-based method's semantic prediction. N-gram methods also show "progressive acceleration," getting faster across multiple reasoning turns , though they are sensitive to temperature

**Strengths:**

- The problem this paper targets seems relevant, and the method is intuitive and motivated.

- There are many interesting findings and the experiments seem to be solid. For example, the most interesting finding is that simple N-gram-based methods, particularly SAM, are highly effective at capturing the redundancy in Best of N or Multi Turn answers. The findings are comprehensive.

**Weaknesses:**

- The one concern I have right now is the novelty of the method. Maybe this is because this is my first time reviewing on the dataset and benchmark field.

- Whether other complex reasoning frameworks such as Tree-of-thought or MCTS-based search algorithms could also benefits from speculative decoding?

- Also, would the approach generalized to other field such as code generation and summarization?

- The paper correctly identifies that hybrid methods (like SAM[EAGLE-3]) are a promising direction and achieve the best performance. However, it also admits that "current hybrid strategies remain heuristic" and that the potential of N-gram matching is "underexploited" due to "suboptimal integration strategies". The paper benchmarks one such simple strategy but does not propose or experiment with more dynamic or refined strategies to truly unlock this potential.

**Questions:**

See weakness.

---

> ### Author Response · Authors · 2025-11-21
> **Response to The Reviewer [1 / 3]**
>
> * **[Weakness 1][Novelty & Contribution]** Thanks for the comment! We acknowledge that, as a benchmark paper, our contributions differ from method-proposing works. We aim to systematically evaluate and contextualize existing speculative decoding methods in the context of test-time scaling, with the goal of identifying their strengths, weaknesses, and overlooked challenges, in line with the broader objectives of benchmark research. Specifically, we believe our work offers three unique contributions that are critical to the community:
>   - **(i) The First Systematic Study on Test-Time Scaling Efficiency** Test-time scaling presents a fundamentally different computational landscape: high repetition in reasoning traces (Table 1), extended sequences exceeding thousands of tokens, and multiple iterative reasoning paths. Despite its growing importance, no prior work has systematically studied speculative decoding in this setting. Prior works have mainly evaluated short-context chat or QA benchmarks, which fail to capture the distinctive characteristics of test-time scaling.
>   - **(ii) Counter-intuitive Findings** Our benchmark uncovers several unexpected phenomena, including but not limited to: simple N-gram methods can rival sophisticated trained models in certain scenarios; training-based methods exhibit unexpected fragility in test-time scaling paradigms; and hybrid approaches show promise yet remain significantly underoptimized.
>   - **(iii) Standardized Protocol** Our benchmark systematically evaluates 9 SOTA methods across 3 categories, 4 reasoning tasks (AIME24/25, MATH-500, GPQA), and 2 test-time scaling paradigms, with ablations over key parameters to facilitate reproducible research.
>
>   We hope these clarifications address your concerns and demonstrate the value of our contributions to this emerging area. We remain open to further discussion if needed!
>
> * **[Weakness 2][Potential Extension to Other Reasoning Frameworks]** Thanks for raising this forward-looking question regarding the applicability of speculative decoding to complex reasoning frameworks. The answer is **YES**, although the adaptation is not straightforward.
>   - **Common Source of Redundancy** Conceptually, the frameworks mentioned in the comment can be regarded as localized, step-level test-time scaling methods. Instead of generating multiple complete responses as in BoN or multi-round thinking, these frameworks expand and evaluate several reasoning paths at each intermediate step. This localized exploration inevitably introduces substantial overlap and redundancy among candidate steps, a phenomenon closely related to the computational inefficiency examined in this benchmark.
>   - **Technical Challenges: Non-trivial Adaptation** A key distinction makes the direct application of existing speculative techniques to ToT or MCTS-based reasoning non-trivial. Unlike the relatively linear generation process in BoN or multi-round thinking, these search-based frameworks involve dynamic operations such as state evaluation, pruning, and backtracking. Existing speculative decoding methods, whether N‑gram-based or advanced variants using suffix automata (e.g., SAM) or adjacency matrices (e.g., Recycling), are designed for monotonic forward generation with a single contiguous prefix. Training‑based methods such as Eagle‑3 share the same limitation. Therefore, adapting them to search-based frameworks would require modifying their underlying write and lookup logic, for example, by maintaining separate speculative modules (such as the individual SAM) for each search step, or extending data structures such as suffix automata and adjacency matrices to handle deletion operations, since certain steps may be pruned during the search.
>
>   In summary, adapting speculative decoding to complex reasoning frameworks is a promising direction for exploiting redundancy, but it requires non-trivial changes that may compromise the objectivity of benchmarking. We leave this for future work and have added a discussion of this point in the revised manuscript (please refer to Appendix A.4, page 29 of the revised manuscript).

---

> ### Author Response · Authors · 2025-11-21
> **Response to The Reviewer [2 / 3]**
>
> * **[Weakness 3][Generalization to Other Domains]** Thanks for the constructive comment! Following your suggestion, we have extended our evaluation to additional domains, including code generation (LiveCodeBench) and summarization (CNN/Daily Mail). The results are as follows:
>     |LiveCodeBench|DSL-8B (T=0)|DSL-8B (T=0.6)|QW3-8B (T=0)|QW3-8B (T=0.6)|
>     |:---|:---:|:---:|:---:|:---:|
>     |Method|MAT / Speed|MAT / Speed|MAT / Speed|MAT / Speed|
>     |EAGLE-3|2.12 / 1.64x|2.00 / 1.33x|4.37 / 2.85x|4.02 / 2.58x|
>     |SAM[EAGLE-3]|4.17 / 3.34x|2.30 / 1.80x|4.46 / 3.27x|3.70 / 2.54x|
>     |SAM|2.90 / 2.79x|1.72 / 1.44x|2.19 / 1.92x|1.77 / 1.58x|
>     |Recycling|2.91 / 2.02x|2.69 / 1.71x|2.90 / 1.98x|2.83 / 1.92x|
>     |PLD|2.24 / 1.81x|-|1.94 / 1.57x|-|
>     |SpS|-|-|5.93 / 0.83x|5.16 / 0.85x|
>     |REST|1.33 / 1.11x|1.37 / 1.02x|1.35 / 1.14x|1.37 / 1.09x|
>     |Lookahead|2.15 / 1.57x|-|2.08 / 1.53x|-|
>     |PIA|2.40 / 1.71x|1.63 / 1.04x|2.04 / 1.46x|1.72 / 1.32x|
>     |AR|1.00 / 1.00x|1.00 / 1.00x|1.00 / 1.00x|1.00 / 1.00x|
>
>     |CNN/Daily Mail|DSL-8B (T=0)|DSL-8B (T=0.6)|QW3-8B (T=0)|QW3-8B (T=0.6)|
>     |:---|:---:|:---:|:---:|:---:|
>     |Method|MAT / Speed|MAT / Speed|MAT / Speed|MAT / Speed|
>     |EAGLE-3|4.76 / 2.83x|4.60 / 2.65x|4.11 / 2.63x|4.01 / 2.53x|
>     |SAM[EAGLE-3]|4.92 / 3.05x|4.70 / 2.79x|4.31 / 2.77x|4.17 / 2.61x|
>     |SAM|1.53 / 1.45x|1.39 / 1.33x|1.55 / 1.45x|1.52 / 1.42x|
>     |Recycling|2.57 / 1.81x|2.50 / 1.71x|2.73 / 1.95x|2.70 / 1.86x|
>     |PLD|1.47 / 1.32|-|1.46 / 1.30x|-|
>     |SpS|-|-|3.57 / 0.51|3.44 / 0.54
>     |REST|1.28 / 1.04x|1.29 / 1.00x|1.28 / 1.05x|1.28 / 1.01x|
>     |Lookahead|1.59 / 1.29|-|1.61 / 1.30x|-|
>     |PIA|1.61 / 1.42x|1.48 / 1.30x|1.58 / 1.39x|1.55 / 1.36x|
>     |AR|1.00 / 1.00x|1.00 / 1.00x|1.00 / 1.00x|1.00 / 1.00x|
>
>   We observe that on the code and summarization datasets, SAM[EAGLE-3] consistently achieves the highest speedup across nearly all scenarios. Moreover, the N-gram-based methods can even outperform the training-based method in some cases. For example, both SAM and Recycling achieve higher acceleration ratios than EAGLE-3 when applied to DeepSeek-R1-Distill-Llama-8B on the LiveCodeBench. These observations are consistent with the findings reported in our manuscript. This indicates that redundancy is **not specific to mathematical reasoning but a universal phenomenon** in generation tasks. The consistent superiority of hybrid methods across domains further validates our benchmark's core insights. We have also included these results and the related discussion in the revised manuscript (please refer to Appendix A.3, page 19 of the revised manuscript).

---

> ### Author Response · Authors · 2025-11-21
> **Response to The Reviewer [3 / 3]**
>
> * **[Weakness 4][Discussion on Hybrid Strategies]** Thanks for the insightful comment! We certainly agree that developing more dynamic hybrid strategies is a vital future direction. That said, we feel it is necessary to provide the following clarifications:
>   - **The Motivation of our Benchmark** As a benchmark, we acknowledge that our contributions differ from method-proposing works. We aim to systematically evaluate existing speculative decoding methods in the context of test-time scaling to reveal their strengths, weaknesses, and hidden challenges. In this context, **if such a straightforward combination already yields a substantial performance gain, it strongly indicates the presence of extensive, underutilized redundancy**. This finding reinforces our benchmark’s central insight that, in the context of test-time scaling, there remains abundant redundancy that has yet to be effectively leveraged.
>   - **A Simple Hybrid Strategy Already Works** First, we emphasize that the simple hybrid strategy benchmarked in this paper demonstrates remarkable effectiveness, as evidenced by the results in Tables 3 and 4 of our manuscript. Furthermore, our analysis of the strategy’s potential in Appendix A.3 (Figure 3) reveals that, with a well-chosen static threshold, the simple hybrid method can achieve a MAT exceeding 8, which is nearly double that of the SOTA EAGLE-3. Although dynamic thresholding could yield higher MAT, it introduces a crucial trade-off between complexity and efficiency. **Therefore, we posit that this simple, low-overhead hybrid strategy is a well-suited baseline to inspire future work.** Nevertheless, despite the potential risk of increased model complexity, we have, as suggested, incorporated several possible hybrid directions into the revised manuscript, including: (i) Context-Aware Switching, where the strategy adapts dynamically based on the semantic context of the generation task; (ii) Phase-Specific Optimization, where the strategy is tailored to different stages of the generation process; and (iii) Uncertainty-Based Toggling, where the strategy is adjusted on-the-fly according to the model’s predictive uncertainty. Further elaboration on these three potential directions can be found in the revised manuscript (please refer to Appendix A.5, page 29 of the revised manuscript), and we remain open to further discussion on this point.
>
>   Following your helpful suggestion, we have revised the manuscript accordingly (please refer to Appendix A.5, page 29 of the revised manuscript). We welcome any further discussion on this point!

---

> ### Author Response · Authors · 2025-11-26
> **Looking forward to your feedback!**
>
> Dear Reviewer 7EPL,
>
> Thank you again for your valuable feedback! We have completed the additional experiments and updated the manuscript based on your suggestions. As the discussion phase is drawing to a close, we kindly ask if our responses and revisions have satisfactorily addressed your concerns. We look forward to hearing from you!
>
> Best,
> Authors

---

### Meta-Review · Area_Chair_H1Tp · 2026-01-14

**Summary:**

The paper studies the performance of various speculative decoding (SD) methods on test-time scaling methods. The paper provides extensive experiments on various reasoning tasks and parameter settings using popular test-time scaling paradigms including batched scaling and sequential scaling. Various insights regarding the choice of drafter (among model-based, n-gram based, and hybrid), the effect of temperature, the effect of drafter training and the effect of scaling have been discussed.

During the rebuttal phase, the authors addressed reviewers's concerns regarding the generalization of the observations across different tasks and model families,  statistical significance of the observations by providing more experimental results, which I believe are sufficient to address these concerns.

Two concerns that still stand is regarding the batched implementation of the scaling algorithms, which is critical for batched scaling methods such as best-of-N, and technical novelty. While these two concerns limit the paper's contribution, I think the paper still provides valuable insights and useful benchmarking results to the speculative decoding literature, where such results are strongly needed, especially regarding test-time scaling methods.

Hence I would recommend the paper for acceptance. The authors should open source the code that can be used to reproduce the results, include the newly added experiments in the revised version, add discussion on the batched implementation, and incorporate other changes suggested by the reviewer.

**Reviewer Concerns:**

See above discussions.

**Reviewer Scores:**

I don't expect reviewer Hr8t and eh5y might increase their scores. Other reviewers will likely keep their scores.

---

### Decision · Program_Chairs · 2026-01-26

Accept (Poster)